# FracAug: Fractional Augmentation boost Graph-level Anomaly Detection under Limited Supervision

## Abstract

Graph-level anomaly detection (GAD) is critical in diverse domains such as drug discovery, yet high labeling costs and dataset imbalance hamper the performance of Graph Neural Networks (GNNs). To address these issues, we propose FracAug, an innovative plug-in augmentation framework that enhances GNNs by generating semantically consistent graph variants and pseudo-labeling with mutual verification. Unlike previous heuristic methods, FracAug learns semantics within given graphs and synthesizes fractional variants, guided by a novel weighted distance-aware margin loss. This captures multi-scale topology to generate diverse, semantic-preserving graphs unaffected by data imbalance. Then, FracAug utilizes predictions from both original and augmented graphs to pseudo-label unlabeled data, iteratively expanding the training set. As a model-agnostic module compatible with various GNNs, FracAug demonstrates remarkable universality and efficacy: experiments across 14 GNNs on 12 real-world datasets show consistent gains, boosting average AUROC, AUPRC, and F1-score by up to 5.72%, 7.23%, and 4.18%, respectively.

## 1 Introduction

Graph-structured data is pivotal in real applications ranging from drug discovery to anomaly identification among proteins (Zhang et al., 2022). While Graph Neural Networks (GNNs) excel at modeling topological and feature-based patterns through message-passing, their effectiveness in Graph-level Anomaly Detection (GAD)—distinguishing anomalous graphs from normal ones—is hindered by two key challenges: limited supervision and extreme class imbalance, as demonstrated in Section 5. Specifically, anomalies represent rare instances, exacerbating data imbalance and restricting the availability of labeled training samples (Chen et al., 2024; Dong et al., 2024). While data augmentation techniques have revolutionized computer vision (Zhang et al., 2023) by generating synthetic labels through rotations or crops, their adaptation to graph domains presents unique challenges. Unlike images, graphs inhabit non-Euclidean space where seemingly minor structural modifications (e.g., edge removal) risk distorting semantic properties and violating the label-invariant assumption—a critical constraint in GAD's challenging setting of limited supervision and inherent class imbalance.

Existing graph-level augmentation methods, such as MAA (Yoo et al., 2022), often employ heuristic modifications without considering data properties, leading to compromised semantics or insufficient diversity in GAD tasks. Consequently, their direct application may underperform vanilla GAD models. We attribute this gap to three key issues: (1) the absence of semantic-preserving augmentation strategies, (2) inadequate handling of imbalance, and (3) ineffective utilization of unlabeled data.

To address these challenges, we introduce FracAug, a novel plug-in augmentation framework that generates semantic-preserving graph variants and pseudo-labels for unlabeled graphs to train GNNs for GAD. Our key innovation leverages the fractional power of adjacency matrices, which encodes multi-scale topological relationships. By computing polynomials of various fractional graphs, guided by weighted distance-aware margin loss, FracAug introduces controlled structural variations while ensuring semantic consistency with the original graph's label, independent of the underlying data distribution. Afterward, a given GNN will produce predictions for both original and synthetic samples, enabling FracAug to employ a mutual verification mechanism for pseudo-labeling unlabeled graphs,

thereby iteratively expanding the training set. This approach not only mitigates supervision scarcity but also enhances model robustness against class imbalance.

In summary, our contributions are as follows:

- We present FracAug, the first augmentation framework designed for GAD that maintains effectiveness under the dual constraints of limited supervision and imbalanced distribution
- FracAug operates as a model-agnostic plug-in augmentation framework compatible with 14 GNNs without architectural modifications, facilitating seamless integration into existing models.
- Extensive experiments on 12 real-world datasets demonstrate that FracAug enhances performance across diverse GNNs, significantly outperforming existing graph augmentation approaches.

## 2 RELATED WORK

**Graph Classification.** Generalized GNNs, such as GCN (Kipf & Welling, 2017), GraphSAGE (Hamilton et al., 2017), GAT (Velickovic et al., 2018), and GIN (Xu et al., 2019), excel at learning graph representations through neighborhood aggregation. Recent advances include LRGNN (Wei et al., 2023), which captures long-range dependencies with stacking GNNs, and GRDL (Wang & Fan, 2024), which achieves state-of-the-art (SOTA) performance by learning representation distributions of graphs. However, these representative GNNs are not specifically designed for GAD tasks. While they can capture certain topological or feature-based patterns, their performance degrades under data imbalance and limited supervision.

**Graph-level Anomaly Detection.** Recognizing the challenges underlying GAD tasks, researchers have introduced specialized approaches to address them. For instance, iGAD (Zhang et al., 2022) introduces dual-discriminative kernels guided by a point mutual information-based loss function to better capture graph anomalies. Later, by mapping anomalies and normal graphs to separate areas based on adjusted candidate nodes, GmapAD (Ma et al., 2023a) shows advanced performance. Moreover, RQGNN (Dong et al., 2024) leverages the Rayleigh Quotient to detect graph anomalies effectively within spectral space. Recently, UniGAD (Lin et al., 2024) combines different levels of graph anomaly detection to capture comprehensive information to enhance the detection accuracy. Although these specialized frameworks show promising performance in addressing data imbalance challenges, they still present inferior results due to the limited supervision issue.

**Graph-level Augmentation.** To address the scarcity of labeled examples, researchers also develop diverse augmentation techniques for graph-level tasks. For example, MAA (Yoo et al., 2022) proposes two separate methods, NodeSam and SubMix, to generate synthetic samples by heuristic structure modification. Besides, GLA (Yue et al., 2022) generates the latent representations as the augmented graphs during the training phase. Subsequently, GMixup (Han et al., 2022) and FGWMixup (Ma et al., 2023b) interpolate graphs or features linearly to mix normal and anomalous samples for producing novel samples. Nevertheless, they fail to produce semantic-preserving samples when dealing with imbalanced data with limited supervision, resulting in unsatisfactory performance.

In contrast, FracAug diverges by leveraging the fractional power of adjacency matrices, a mathematically grounded operation that preserves semantics within graphs while introducing multi-scale structural variations. The incorporation of weighted distance-aware margin loss further enables FracAug to adapt to the imbalanced scenario. Furthermore, its pseudo-labeling mechanism explicitly addresses the limited supervision constraint. As a plug-in module, FracAug overcomes above limitations in GNNs and graph-level augmentation methods without modifying GNN architectures, enabling given GNN models to learn discriminative features for GAD tasks even with sparse labels.

## 3 PRELIMINARIES

**Notation.** Let $G = (\boldsymbol{A}, \boldsymbol{X})$ denote an undirected graph with $n$ nodes and $m$ edges, where $\boldsymbol{A} \in \mathbb{R}^{n \times n}$ is the adjacency matrix and $\boldsymbol{X} \in \mathbb{R}^{n \times F}$ is the node feature matrix. $\boldsymbol{A}_{ij} = 1$ if an edge exists between node $i$ and $j$, and $\boldsymbol{A}_{ij} = 0$ otherwise. $\boldsymbol{D}$ is the diagonal degree matrix of $\boldsymbol{A}$, and the normalized adjacency matrix can be defined as $\tilde{\boldsymbol{A}} = \boldsymbol{D}^{-\frac{1}{2}} \boldsymbol{A} \boldsymbol{D}^{-\frac{1}{2}}$ correspondingly. For a given matrix $\boldsymbol{M}$, $\boldsymbol{M}^\alpha$ stands for the $\alpha$-th power of matrix $\boldsymbol{M}$, where $\alpha \geq 0$. When $\boldsymbol{M}$ can be eigendecomposed,

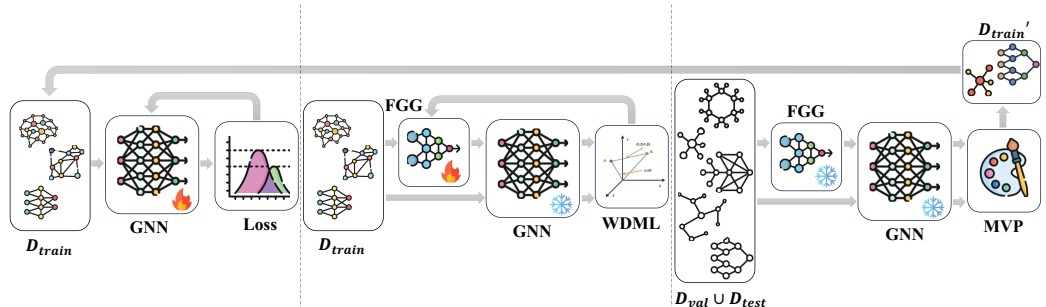

Figure 1: Overview of FracAug.

$M^\alpha = U\Lambda^\alpha V$, where $U, V \in \mathbb{C}^{n \times n}$ are unitary matrices and $\Lambda \in \mathbb{R}^{n \times n}$ is a diagonal matrix composed of the eigenvalues of $M$.

**Continuous Semantic Space.** Given a graph $G$ with adjacency matrix $A$ and a graph signal $x \in \mathbb{R}^F$, the semantic space of $G$ is defined as a subspace $\mathcal{S} \subseteq \mathbb{R}^F$ generated by the set of vectors obtained through the application of powers of $A$ to $x$. Unlike previous approaches that rely on discrete semantics constrained by integer powers, i.e., $\{A^t x | t \in \mathbb{N}\}$, our continuous semantic space formulation, $\mathcal{S} = span\{A^t x | t \in \mathbb{N}\}$, captures the underlying continuous semantic manifold of the graph, which enables us to synthesize novel semantic-preserving graph instances, shown in Section 4.

**Graph-level Anomaly Detection.** In this work, we focus on enhancing GAD performance under limited supervision. Given a training set with $k$ labeled samples, $\mathcal{D}_{train} = \{(G_1, y_1), (G_2, y_2), ..., (G_k, y_k)\}$, the goal of GAD is to train a model that classifies unseen graphs as normal or anomalous. In real deployment, there are two main challenges in GAD. Firstly, $\mathcal{D}_{all} = \mathcal{D}_{train} \cup \mathcal{D}_{val} \cup \mathcal{D}_{test}$ contains $N_0$ normal graphs and $N_1$ anomalous graphs, where $N_0 \gg N_1$, leading to severe imbalanced problem. Secondly, only limited labeled graphs are accessible during training, i.e., $k \ll N_0 + N_1$, resulting in the limited supervision issue. Therefore, the key to enhancing the ability of GNNs on real-world GAD tasks is to address these two challenges simultaneously.

**Graph-level Augmentation.** Graph-level augmentation has been proven effective in improving the performance of GNNs on graph-level tasks. Graph generation and pseudo-labeling are the most common ways to conduct graph-level augmentation:

- Graph generation: This strategy maps the graph $G \in \mathcal{D}_{train}$ to a new graph $G'$, i.e., $(G, y) \mapsto (G', y)$. The generated graph should have a semantic meaning similar to that of the original $G$.
- Pseudo-labeling: This approach leverages a GNN trained on $\mathcal{D}_{train}$ to classify samples from $\mathcal{D}_{val} \cup \mathcal{D}_{test}$ and assign pseudo-labels to samples with high confidence under a certain criterion.

By combining graph generation and pseudo-labeling techniques while tackling the imbalanced issue, FracAug effectively boosts GNN performance for GAD under limited supervision.

## 4 METHOD

### 4.1 OVERVIEW

Our proposed FracAug consists of three key components: (1) **Fractional Graph Generator (FGG)** in Section 4.2 captures the inherent semantics of graphs, enabling the synthesis of fractional variants that maintain semantic consistency with originals, as we demonstrate theoretically. (2) **Weighted Distance-Aware Margin Loss (WDML)** in Section 4.3 addresses data imbalance to guide FGG, employing distance-based margins to position synthetic graphs near original counterparts while ensuring distinctiveness. (3) **Mutual Verification Pseudo-Labeler (MVP)** in Section 4.4 minimizes pseudo-labeling errors through mutual verification of predictions from original and synthetic graphs, facilitating reliable and iterative training set expansion.

Figure 1 illustrates the pipeline of our FracAug. Initially, we warm up a given GNN to establish a preliminary semantic understanding of the GAD task. Then, we freeze the GNN parameters and utilize its outputs to train the FGG with WDML. The trained FGG then generates fractional graph variants, and the GNN predicts on both original and synthetic graphs to pseudo-label data within the validation and test sets using MVP, which are subsequently incorporated into the original training set.

Finally, we train the GNN using the new training set and continue the above process until both the GNN and our FracAug framework reach reasonable capability.

## 4.2 Fractional Graph Generator

**Flexible Eigengraph Combinations.** The fractional power of the adjacency matrix, $\boldsymbol{A}^\alpha$, where $\alpha \geq 0$, serves as the mathematical foundation of our framework due to its unique properties. Unlike integer powers of $\boldsymbol{A}$, which only capture discrete-step neighborhood aggregations, fractional powers enable continuous interpolation of graph structures, providing fine-grained control over topological variations. Crucially, $\boldsymbol{A}^\alpha$ can be expressed as a combination of eigengraphs derived from eigendecompositions. For an undirected graph $G$ with symmetric adjacency matrix $\boldsymbol{A}$, we can decompose $\boldsymbol{A}^\alpha$ as:

$$\boldsymbol{A}^\alpha = \boldsymbol{U} \boldsymbol{\Lambda}^\alpha \boldsymbol{U}^T = \sum_{i=1}^n \lambda_i^\alpha \boldsymbol{u}_i \boldsymbol{u}_i^T,$$

where $\boldsymbol{\Lambda}$ is the diagonal eigenvalue matrix containing $\{\lambda_i\}_{i=1}^n$ in a descending order, $\boldsymbol{U}$ is the eigenvector matrix formed by $\{\boldsymbol{u}_i\}_{i=1}^n$, and $\boldsymbol{u}_i \boldsymbol{u}_i^T$ is the $i$-th eigengraph. It reveals two key advantages:

- Multi-scale Structure Adaptation: Fractional powers enable tunable control over spectral components via $\alpha$, where lower values ($\alpha < 1$) emphasize homophilic graph signals (low-frequency eigengraphs), while higher values ($\alpha > 1$) accentuate heterophilic graph signals (high-frequency eigengraphs) (Yan et al., 2023). This adaptive reweighting preserves the hierarchical topology while generating augmented graphs, signaling structural anomalies for detection.
- Semantic-preserving Combination: By combining eigengraphs, FracAug preserves semantic-critical structures (targeting spectral deviations linked to anomalies (Dong et al., 2024)), ensuring that generated graphs retain the original semantics.

**Semantic Preservation.** Prior studies, such as GIN (Xu et al., 2019), rely on integer powers of adjacency matrices, limiting them to discrete semantic preservation. In contrast, we prove that for any $\alpha \geq 0$, $\boldsymbol{A}^\alpha \boldsymbol{x}$ resides in the original semantic space. Moreover, we further derive a theoretical boundary to quantify differences between the original and fractional graphs, detailed in Appendix A.

**Theorem 1.** *Given a polynomial function $p(\cdot; \boldsymbol{\theta})$ parameterized by $\boldsymbol{\theta}$, for any $\alpha \geq 0$, there exists $\boldsymbol{\theta}^*$ such that $\boldsymbol{A}^\alpha \approx p(\boldsymbol{A}; \boldsymbol{\theta}^*) = \sum_{t=0}^T \boldsymbol{\theta}_t^* \boldsymbol{A}^t, T \in \mathbb{N}$. With proper parameter $\boldsymbol{\theta}^*$, the difference of them is bounded by $\beta e^{-\gamma T}$, where $\beta, \gamma > 0$ depend on the eigenvalues of $\boldsymbol{A}$. Since $\boldsymbol{A}^\alpha$ can be represented as a polynomial combination of $\{\boldsymbol{A}^t\}_{t \in \mathbb{N}}$, $\boldsymbol{A}^\alpha \boldsymbol{x}$ lies in $\mathcal{S}$ of the original graph as $T \to \infty$.*

Theorem 1 ensures that fractional graphs preserve the original semantic space while encoding multi-scale semantics. The continuous parameter $\alpha$ spans all possible semantic variations within this space, enabling rich and comprehensive augmentation. Besides, previous approaches such as MAA (Yoo et al., 2022) leverage heuristic perturbation techniques for generating synthetic graphs, which may result in useful substitutes near the semantic space of the original graph. Theorem 2 formally bridges these perturbation-based approaches with our fractional graph augmentation, revealing their shared theoretical foundations. The complete proof is provided in Appendix A.

**Theorem 2.** *Let the structural perturbation on a graph be a perturbation matrix $\boldsymbol{P}$ added to the original graph, so that any graph generated by a structural perturbation method can be expressed as $\boldsymbol{A} + \boldsymbol{P}$. Then, we can derive $||\boldsymbol{A}^\alpha - (\boldsymbol{A} + \boldsymbol{P})|| \leq c||\boldsymbol{P}|| + \max_i |\lambda_i - \lambda_i^\alpha|$, where $c$ depends on $\alpha$ and the spectral gap of $\boldsymbol{P}$, and $\lambda_i$ is the $i$-th eigenvalue of $\boldsymbol{A} + \boldsymbol{P}$. Thus, by choosing an appropriate $\alpha$, $\boldsymbol{A}^\alpha$ can approximate any graph generated by structural perturbation methods.*

Based on Theorem 2, we observe that for a suitably chosen $\alpha$, the fractional graph can approximate any sample generated by perturbation-based framework, demonstrating its generalization capability.

**Fractional Graph Generation.** Building on the above analysis, we conclude that fractional graphs offer powerful augmentation capabilities. However, directly deriving the fractional power of the adjacency matrix can be computationally prohibitive and may yield invalid results for non-semi-definite adjacency matrix. Thus, a transformation function $h(\cdot)$ is applied to the adjacency matrix to ensure valid fractional powers while preserving structural integrity (Yan et al., 2023). Specifically, instead of adding self-loops before normalization of the adjacency matrix, we introduce them after

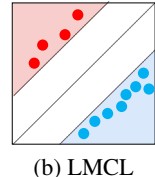 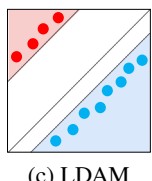 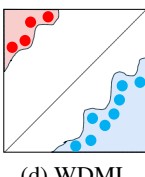

(a) Softmax        (b) LMCL        (c) LDAM        (d) WDML

Figure 2: Decision boundaries of different margin losses.

normalization and rescale the matrix, so the resulting adjacency matrix can be defined as:

$$\hat{\boldsymbol{A}} = h(\tilde{\boldsymbol{A}}) = \frac{1}{2}(\boldsymbol{I} + \tilde{\boldsymbol{A}}),$$

which is still a normalized adjacency matrix. Since all eigenvalues of $\tilde{\boldsymbol{A}}$ lie within $[-1, 1]$, the corresponding eigenvalues of $\hat{\boldsymbol{A}}$ fall within $[0, 1]$. Therefore, $h(\cdot)$ transforms $\boldsymbol{A}$ into a positive semi-definite matrix $\hat{\boldsymbol{A}}$, which allows the design of FGG.

Moreover, to mitigate computational costs for large graphs, we precompute the eigendecomposition (EVD) using the Arnoldi method (Lehoucq et al., 1998), retaining only the top-$k_l$ largest and top-$k_s$ smallest eigenpairs. Denote $\hat{\boldsymbol{\Lambda}}_{k_l}, \hat{\boldsymbol{\Lambda}}_{k_s}$ as diagonal matrices of the top-$k_l$ largest and top-$k_s$ smallest eigenvalues, $\boldsymbol{U}_{k_l} = \boldsymbol{U}[:, 0 : k_l], \boldsymbol{U}_{k_s} = \boldsymbol{U}[:, n-k_s : n]$ as the corresponding matrices of eigenvectors, and the generated graph of $G(\boldsymbol{A}, \boldsymbol{X})$ as $G'(\boldsymbol{A}', \boldsymbol{X})$, FGG can be formulated as:

$$g(\boldsymbol{A}, k, H) = \sum_{h=1}^{H} \omega_h \boldsymbol{U}_k \hat{\boldsymbol{\Lambda}}_k^{\alpha_h} \boldsymbol{U}_k^T,$$

$$\boldsymbol{A}' = \text{FGG}(\boldsymbol{A}, k_l, k_s, H_l, H_s) = \omega g(\boldsymbol{A}, k_l, H_l) + (1 - \omega) g(\boldsymbol{A}, k_s, H_s),$$

where $\sum_{h=1}^{H} \omega_h = 1$, $\omega$ are learnable coefficients, and $\alpha_h$ is $h$-th learnable fractional power of the matrix. By combining multiple fractional graphs with tunable weights, our generated graphs can capture comprehensive information while preserving semantics. Although the anomalous properties are well-preserved in graphs from FGG based on previous analysis, inherent data imbalance risks biasing FGG training. To counteract this, in Section 4.3, we design WDML to guide FGG training.

### 4.3 WEIGHTED DISTANCE-AWARE MARGIN LOSS

**Revisiting Margin Loss.** To better separate the semantic spaces of different graphs and enable FGG to generate high-quality fractional graphs robust to class imbalance, we introduce a novel margin loss function. Before illustrating the details of WDML, we first reexamine representative margin losses, with comprehensive empirical validation provided in Appendix G. Formally, margin loss based on cross-entropy can be defined as:

$$L = -\frac{1}{N} \sum_{i=1}^{N} \log \frac{e^{\boldsymbol{s}_{\boldsymbol{y}_i} - m}}{e^{\boldsymbol{s}_{\boldsymbol{y}_i} - m} + \sum_{j=1, j \neq \boldsymbol{y}_i}^{C} e^{\boldsymbol{s}_j}}, \tag{1}$$

where $N$ is the number of the samples, $\boldsymbol{s}$ represents the normalized logits predicted by a given GNN, $\boldsymbol{y}_i$ is the ground truth label of $i$-th sample, $m$ is the margin that determines the decision boundary, and $C$ is the number of classes.

As shown in Figure 2 (a), when setting margin $m$ to 0, the margin loss is degraded to a cross-entropy loss, which lacks explicit mechanisms to separate classes in complex scenarios. Another margin loss is LMCL (Wang et al., 2018) with $m$ as a hyperparameter. Figure 2 (b) describes the result of setting $m > 0$, enforcing better inter-class separation. However, this uniform margin shifts the decision boundaries of different classes by the same value, which fails to detect the unique class-specific properties. Afterward, LDAM (Cao et al., 2019) in Figure 2 (c) tackles the issues by setting class-specific margin $m_c$ for $c$-th class so that the decision boundaries can accommodate scenarios where classes require distinct margins. Existing margin losses typically employ fixed margins, which prove suboptimal for GAD where sample-specific semantic variations exist. To address this, we propose WDML, which assigns dynamic margins based on the intrinsic distance of each synthetic sample and its original graph with a weight according to its class. Figure 2 (d) describes the adaptive decision boundary of WDML.

**Margin Loss Based on Sample-Specific Distance.** For the $i$-th training graph $G_i$ and its counterpart $G_i'$ generated by FGG, we extract graph-level embeddings $\boldsymbol{o}_i$ and $\boldsymbol{o}_i'$ via a given GNN. Then, our distance-aware margin can be defined as:

$$m_i = \frac{1 - \cos(\boldsymbol{o}_i, \boldsymbol{o}_i')}{2}, \tag{2}$$

where $\cos$ represents cosine similarity. Substituting $m$ in Equation 1 with the sample-specific margin $m_i$ yields a distance-aware margin loss. By computing angular distances in Equation 2, this loss shifts the semantic space away from decision boundaries by a margin $m_i$, ensuring generated samples retain the original label with high confidence. To further address class imbalance, WDML incorporates weights based on class frequency:

$$L_{\text{WDML}} = -\sum_{i=1}^{N} \frac{1}{N_{\boldsymbol{y}_i}} \log \frac{e^{\boldsymbol{s}_{\boldsymbol{y}_i} - m_i}}{e^{\boldsymbol{s}_{\boldsymbol{y}_i} - m_i} + \sum_{j=1, j\neq \boldsymbol{y}_i}^{C} e^{\boldsymbol{s}_j}},$$

where $N_{\boldsymbol{y}_i}$ is the number of samples in class $\boldsymbol{y}_i$. With the assistance of WDML, FGG can generate fractional graphs effectively without being biased by the imbalanced distribution of labels. To further boost the performance by data augmentation, we design MVP to combine graph generation and pseudo-labeling techniques, whose details will be elaborated in Section 4.4.

### 4.4 MUTUAL VERIFICATION PSEUDO-LABELER

**Insight on Mutual Verification.** Prior pseudo-labeling methods for related tasks, such as ConsisGAD (Chen et al., 2024), only rely on confidences from original samples, prone to high errors under low supervision (Dong et al., 2025). Therefore, we first investigate how mutual verification mitigates the error rates compared to single-view methods, theoretically. The proof is detailed in the Appendix A.

**Proposition 1.** *For a given GNN, assume its prediction error rates for original graphs and corresponding fractional graphs are both $\delta$. The correlation coefficient between the errors is denoted as $\rho$. Then, when mutual verification is used, compared to single-view methods, the reduction factor of the error rate and its variance can be up to $\delta + \rho\delta(1 - \delta)$ and $\rho$, respectively.*

Proposition 1 demonstrates that the mutual verification mechanism leverages semantic consistency between original graphs and their fractional counterparts to enhance pseudo-labeling reliability. Building on this, we design MVP based on the agreement between predicted labels for the original and synthetic samples, as detailed below.

**High-Quality Pseudo-Label Prediction.** Based on the above analysis, MVP assigns a pseudo-label $\hat{y}_i$ to the $i$-th sample in the validation or test set if and only if:

$$\hat{y}_i = \begin{cases} 0, & p_i \leq \tau_n \wedge p_i' \leq \tau_n, \\ 1, & p_i \geq \tau_a \wedge p_i' \geq \tau_a, \end{cases}$$

where $p_i, p_i'$ represent the anomaly probabilities of the $i$-th original sample and its fractional counterpart, respectively, and $\tau_n, \tau_a$ denote the confidence thresholds of a sample being normal/anomalous. For any given GNN, we iteratively incorporate high-confidence pseudo-labeled samples from the validation and test sets into the training set, further mitigating the limited supervision issue.

The core innovation of our mutual verification framework lies in leveraging the semantic consistency between original and fractional graphs to generate high-confidence pseudo-labels. This mechanism addresses the scarcity of labeled anomalies by iteratively expanding the training set with reliable samples, guided by theoretical guarantees of robustness.

In summary, our proposed FracAug combines FGG, WDML, and MVP to generate fractional graphs and pseudo-label samples to boost the performance of GNNs on GAD tasks under limited supervision. The experiments in Section 5 further validate our theoretical analysis in Section 4.

## 5 EXPERIMENTS

### 5.1 EXPERIMENTAL SETUP

**Datasets.** We evaluate FracAug on 12 real-world datasets, including MCF-7, MOLT-4, PC-3, SW-620, NCI-H23, OVCAR-8, P388, SF-295, SN12C, UACC257, PROTEINS_full and DBLP_v1.

Table 1: Average AUROC, AUPRC, and F1-score on 6 datasets with multiple runs, using graph classification models as baselines, where the white columns represent vanilla models and the "+FA" represent models augmented by FracAug.

| Datasets | Metrics | GCN | +FA | SAGE | +FA | GAT | +FA | GIN | +FA | LRGNN | +FA | GRDL | +FA |
|---|---|---|---|---|---|---|---|---|---|---|---|---|---|
| P388 | AUROC | 0.5171 | **0.5896** | 0.5820 | **0.6277** | 0.4964 | **0.5758** | 0.5565 | **0.5913** | 0.5546 | **0.6316** | 0.5500 | **0.5852** |
|  | AUPRC | 0.3488 | **0.3926** | 0.3569 | 0.3540 | 0.2045 | **0.2134** | 0.2850 | **0.3309** | 0.2880 | **0.2953** | 0.2318 | **0.2859** |
|  | F1-score | 0.3428 | **0.3886** | 0.4138 | **0.4741** | 0.4478 | **0.5481** | 0.4468 | **0.4491** | 0.4430 | **0.5496** | 0.4808 | **0.4814** |
| SF-295 | AUROC | 0.5730 | **0.5813** | 0.5858 | **0.6057** | 0.5960 | **0.6171** | 0.5844 | **0.6076** | 0.5903 | **0.6185** | 0.6156 | **0.6349** |
|  | AUPRC | 0.3290 | **0.3308** | 0.3161 | **0.3222** | 0.2652 | **0.2708** | 0.2766 | **0.2832** | **0.3000** | 0.2972 | 0.2796 | **0.3115** |
|  | F1-score | 0.4199 | **0.4279** | 0.4463 | **0.4652** | 0.5065 | **0.5389** | 0.4803 | **0.5047** | 0.4669 | **0.5068** | **0.5221** | 0.5173 |
| SN12C | AUROC | 0.5624 | **0.5818** | 0.5705 | **0.6030** | 0.5863 | **0.6020** | 0.5995 | **0.6079** | 0.5973 | **0.6104** | 0.6061 | **0.6211** |
|  | AUPRC | 0.2812 | **0.2981** | 0.2859 | **0.3133** | 0.2468 | **0.2585** | 0.2696 | **0.2746** | 0.2729 | **0.2888** | 0.2803 | **0.2875** |
|  | F1-score | 0.4463 | **0.4546** | 0.4514 | **0.4670** | 0.5058 | **0.5191** | 0.5030 | **0.5110** | 0.4978 | **0.5012** | 0.5026 | **0.5183** |
| UACC257 | AUROC | 0.5660 | **0.5831** | 0.6006 | **0.6132** | 0.5890 | **0.6174** | 0.5877 | **0.6015** | 0.6020 | **0.6189** | 0.6155 | **0.6340** |
|  | AUPRC | 0.3334 | **0.3509** | **0.3360** | 0.3337 | **0.3493** | 0.3389 | 0.2480 | **0.2598** | 0.3047 | **0.3215** | 0.2942 | **0.3051** |
|  | F1-score | 0.3921 | **0.3954** | 0.4289 | **0.4456** | 0.4031 | **0.4455** | 0.4906 | **0.4983** | 0.4585 | **0.4631** | 0.4843 | **0.4990** |
| PROTEINS_full | AUROC | 0.6186 | **0.6259** | 0.5942 | **0.6310** | 0.6157 | **0.6836** | 0.5799 | **0.6174** | 0.6434 | **0.6503** | 0.5895 | **0.5987** |
|  | AUPRC | 0.6325 | **0.6404** | 0.6086 | **0.6516** | 0.6350 | **0.7005** | 0.6259 | **0.6358** | 0.6603 | **0.6722** | 0.6015 | **0.6077** |
|  | F1-score | 0.6199 | **0.6273** | 0.5909 | **0.6289** | 0.6158 | **0.6859** | 0.5679 | **0.6175** | 0.6431 | **0.6469** | 0.5856 | **0.5962** |
| DBLP_v1 | AUROC | 0.7866 | **0.7973** | 0.6218 | **0.6825** | 0.6119 | **0.6885** | 0.6231 | **0.8044** | 0.7922 | **0.8006** | 0.8089 | **0.8222** |
|  | AUPRC | 0.8462 | **0.8515** | 0.7133 | **0.7769** | 0.7507 | **0.7796** | 0.7201 | **0.8626** | 0.8485 | **0.8537** | 0.8671 | **0.8716** |
|  | F1-score | 0.7854 | **0.7974** | 0.6161 | **0.6805** | 0.5782 | **0.6868** | 0.5996 | **0.8028** | 0.7919 | **0.8007** | 0.8071 | **0.8220** |

Table 2: Average AUROC, AUPRC, and F1-score on 6 datasets with multiple runs, using GAD models as baselines, where the white columns represent vanilla models and the "+FA" represent models augmented by FracAug.

| Datasets | Metrics | iGAD | +FA | GmapAD | +FA | RQGNN | +FA | UniGAD | +FA |
|---|---|---|---|---|---|---|---|---|---|
| P388 | AUROC | 0.5143 | **0.5300** | 0.4782 | **0.5057** | 0.5952 | **0.6108** | 0.5104 | **0.5167** |
|  | AUPRC | 0.1923 | **0.1939** | 0.2478 | **0.2599** | 0.2484 | **0.2650** | 0.1679 | **0.1748** |
|  | F1-score | 0.4669 | **0.4843** | 0.3894 | **0.4099** | 0.5879 | **0.5883** | 0.4781 | **0.4812** |
| SF-295 | AUROC | 0.5811 | **0.5815** | 0.5414 | **0.5535** | 0.5582 | **0.5902** | 0.5439 | **0.5730** |
|  | AUPRC | 0.2666 | **0.2821** | 0.3066 | **0.3070** | 0.2141 | **0.2342** | **0.3000** | 0.2846 |
|  | F1-score | **0.4836** | 0.4705 | 0.4030 | **0.4167** | 0.5719 | **0.5847** | 0.4117 | **0.4582** |
| SN12C | AUROC | 0.5522 | **0.5537** | 0.5343 | **0.5441** | 0.5597 | **0.6038** | 0.5433 | **0.5497** |
|  | AUPRC | 0.1817 | **0.1858** | 0.3262 | **0.3315** | 0.1927 | **0.2442** | **0.2028** | 0.1961 |
|  | F1-score | **0.5151** | 0.5144 | 0.3754 | **0.3818** | 0.5648 | **0.5826** | 0.4851 | **0.4986** |
| UACC257 | AUROC | 0.5697 | **0.5748** | 0.5394 | **0.5597** | 0.5528 | **0.5692** | 0.5710 | **0.5832** |
|  | AUPRC | 0.1906 | **0.1970** | 0.2927 | **0.3004** | 0.1601 | **0.1885** | 0.2510 | **0.2642** |
|  | F1-score | 0.5201 | **0.5224** | 0.3986 | **0.4144** | 0.5522 | **0.5678** | 0.4676 | **0.4710** |
| PROTEINS_full | AUROC | 0.5976 | **0.6206** | 0.5041 | **0.6289** | 0.5641 | **0.6365** | 0.6173 | **0.6212** |
|  | AUPRC | 0.6200 | **0.6333** | 0.5169 | **0.6436** | 0.5673 | **0.6563** | 0.6295 | **0.6338** |
|  | F1-score | 0.5960 | **0.6211** | 0.5020 | **0.6299** | 0.5600 | **0.6310** | 0.6178 | **0.6223** |
| DBLP_v1 | AUROC | 0.7755 | **0.7909** | 0.4975 | **0.5045** | 0.8065 | **0.8082** | 0.7601 | **0.7965** |
|  | AUPRC | 0.8377 | **0.8473** | 0.6242 | **0.6548** | 0.8584 | **0.8598** | 0.8346 | **0.8509** |
|  | F1-score | 0.7749 | **0.7910** | 0.4968 | **0.5021** | 0.8060 | **0.8079** | 0.7549 | **0.7966** |

These datasets are obtained from TUDataset[1], and their detailed statistics are listed in Appendix B. We randomly divide each dataset into 1%/1%/98% for $\mathcal{D}_{train}$/$\mathcal{D}_{val}$/$\mathcal{D}_{test}$ to simulate the limited supervision scenario in real applications. Due to the limited space, we present results of MCF-7, MOLT-4, PC-3, SW-620, NCI-H23, and OVCAR-8 in Appendix J.

**Baselines.** We integrate our FracAug with 10 distinct GNNs, including generalized graph classification models and specialized GAD models, to demonstrate its broad applicability. Besides, to further confirm the usefulness of FracAug, we compare FracAug against 4 SOTA graph-level augmentation frameworks based on their original vanilla models.

- Graph Classification: GCN (Kipf & Welling, 2017), GraphSAGE (Hamilton et al., 2017), GAT (Velickovic et al., 2018), GIN (Xu et al., 2019), LRGNN (Wei et al., 2023), and GRDL (Wang & Fan, 2024).
- Graph-level Anomaly Detection: iGAD (Zhang et al., 2022), GmapAD (Ma et al., 2023a), RQGNN (Dong et al., 2024), and UniGAD (Lin et al., 2024).

[1]https://chrsmrrs.github.io/datasets/docs/datasets/

Table 3: Average AUROC, AUPRC, and F1-score on 6 datasets with multiple runs, using graph-level augmentation models as baselines, where the white columns represent vanilla models and their own augmentation method, while the "+FA" represent vanilla models augmented by FracAug.

| Datasets | Metrics | MAAv | NodeSam | SubMix | +FA | GLAv | GLA | +FA | GMixupv | GMixup | +FA | FGWMixupv | FGWMixup | +FA |
|---|---|---|---|---|---|---|---|---|---|---|---|---|---|---|
| P388 | AUROC | 0.5500 | 0.5069 | 0.5057 | **0.5720** | 0.5622 | 0.5816 | **0.6057** | 0.5469 | 0.5265 | **0.5647** | 0.5480 | 0.5409 | **0.5729** |
| | AUPRC | 0.1958 | 0.1985 | 0.1127 | **0.2229** | 0.2157 | 0.2264 | **0.2632** | 0.1694 | 0.1536 | **0.1957** | 0.2056 | 0.1951 | **0.2362** |
| | F1-score | 0.5520 | 0.5000 | 0.4987 | **0.5746** | 0.5372 | 0.5766 | **0.5925** | 0.5315 | 0.5078 | **0.5373** | 0.5421 | 0.5282 | **0.5798** |
| SF-295 | AUROC | 0.5649 | 0.5579 | 0.5292 | **0.5753** | 0.5954 | 0.6060 | **0.6197** | 0.5665 | 0.5687 | **0.6040** | 0.5893 | 0.5981 | **0.6459** |
| | AUPRC | 0.2114 | 0.1939 | 0.2218 | **0.2252** | 0.2316 | 0.2451 | **0.2648** | 0.2004 | 0.2061 | **0.2509** | 0.2331 | 0.2585 | **0.3017** |
| | F1-score | 0.5736 | 0.5644 | 0.5406 | **0.5820** | 0.5643 | 0.5702 | **0.5855** | 0.5245 | 0.5205 | **0.5371** | 0.5300 | 0.5161 | **0.5623** |
| SN12C | AUROC | 0.5509 | 0.5639 | 0.5160 | **0.5795** | 0.5715 | 0.6003 | **0.6141** | 0.5713 | 0.5336 | **0.5984** | 0.5831 | 0.5693 | **0.6314** |
| | AUPRC | 0.1726 | 0.1845 | 0.1966 | **0.2164** | 0.2048 | 0.2344 | **0.2528** | 0.2163 | 0.2047 | **0.2524** | 0.2382 | 0.2252 | **0.2929** |
| | F1-score | 0.5538 | 0.5405 | 0.5190 | **0.5770** | 0.5644 | 0.5590 | **0.5655** | 0.5136 | 0.4920 | **0.5195** | 0.5085 | 0.5002 | **0.5326** |
| UACC257 | AUROC | 0.5623 | 0.5023 | 0.5211 | **0.5947** | 0.6159 | 0.6198 | **0.6327** | 0.5853 | 0.5843 | **0.6209** | 0.5535 | 0.5632 | **0.6368** |
| | AUPRC | 0.1805 | 0.0559 | 0.0942 | **0.2210** | 0.2755 | 0.2745 | **0.2978** | 0.2365 | 0.2285 | **0.2963** | 0.1534 | 0.2172 | **0.2718** |
| | F1-score | 0.5541 | 0.5005 | 0.5214 | **0.5784** | 0.5033 | **0.5131** | 0.5050 | 0.4993 | **0.5042** | 0.4898 | 0.5470 | 0.4830 | **0.5571** |
| PROTEINS_full | AUROC | 0.6009 | 0.6083 | 0.4998 | **0.6217** | 0.5652 | 0.5325 | **0.6249** | 0.5411 | 0.5132 | **0.6097** | 0.5078 | 0.5259 | **0.6082** |
| | AUPRC | 0.6183 | 0.6294 | 0.6186 | **0.6366** | 0.6053 | 0.5343 | **0.6476** | 0.5810 | 0.5057 | **0.6244** | 0.5300 | **0.6934** | 0.6333 |
| | F1-score | 0.6015 | 0.6039 | 0.4316 | **0.6214** | 0.5577 | 0.5291 | **0.6227** | 0.5348 | 0.5025 | **0.6102** | 0.4909 | 0.3809 | **0.6031** |
| DBLP_v1 | AUROC | 0.6446 | 0.6608 | 0.6205 | **0.6822** | 0.7040 | 0.6402 | **0.7222** | 0.7939 | 0.7885 | **0.7994** | 0.7865 | 0.7772 | **0.7989** |
| | AUPRC | 0.7689 | 0.7868 | **0.7947** | 0.7816 | 0.7882 | 0.7450 | **0.8085** | 0.8503 | 0.8471 | **0.8563** | 0.8461 | 0.8377 | **0.8549** |
| | F1-score | 0.6252 | 0.6408 | 0.6291 | **0.6778** | 0.7029 | 0.6147 | **0.7177** | 0.7937 | 0.7878 | **0.7985** | 0.7856 | 0.7772 | **0.7981** |

- Graph-level Augmentation: MAA (Yoo et al., 2022), GLA (Yue et al., 2022), GMixup (Han et al., 2022), and FGWMixup (Ma et al., 2023b).

**Experimental Settings.** To ensure fair evaluation, we standardize evaluations by: (1) sourcing all baseline code from GitHub and replacing loss functions with weighted version to mitigate class imbalance; (2) using authors' recommended hyperparameters for baselines, while optimizing FracAug's hyperparameters via grid search to maximize the summed AUROC/AUPRC/F1-score on validation sets. Complete configurations are detailed in Appendix F.

## 5.2 EXPERIMENTAL RESULTS

Note that we conduct all the experiments in a semi-supervised setting, where we use both the validation and the test sets for pseudo-labeling. We first evaluate the performance of FracAug on 6 graph classification models and 4 GAD models. Tables 1 and 2 report the AUROC, AUPRC, and F1-score on 6 datasets. Besides, we also compare FracAug with 4 graph-level augmentation methods on their vanilla models, as shown in Table 3. The best performance of each model is highlighted in boldface. To sum up, FracAug effectively boosts the performance of GNNs and outperforms almost all baselines on these real-world datasets. Next, we provide our detailed observations.

**Augmentation for Graph Classification Models.** We analyze 4 generalized GNNs (GCN, Graph-SAGE, GAT, and GIN) and 2 recent models (LRGNN and GRDL) under limited supervision conditions. While generalized GNNs, due to architectural simplicity, struggle to capture nuanced anomaly patterns in GAD tasks, FracAug boosts their performance across most datasets as shown in Table 1, validating its augmentation efficacy. Surprisingly, LRGNN and GRDL initially underperform simpler GNNs in some cases, likely hindered by label scarcity, but regain competitiveness when integrated with FracAug, highlighting FracAug's adaptability to advanced architectures.

**Augmentation for Graph-level Anomaly Detection Models.** Specialized GAD models (iGAD, GmapAD, RQGNN, and UniGAD) exploit task-specific properties but falter under limited supervision due to insufficient generalization capability. Notably, these task-specific architectures may underperform even basic GNNs in low-label regimes as presented in Table 2, emphasizing FracAug's effectiveness. Our framework universally elevates their performance by compensating for supervision scarcity, validating its versatility across model paradigms.

**Comparison with Graph-level Augmentation Frameworks.** To further prove the effectiveness of FracAug, we compare it against leading graph-level augmentation frameworks, including MAA, GLA, GMixup, and FGWMixup. In Table 3, we denote their corresponding vanilla models as MAAv, GLAv, GMixupv, and FGWMixupv, respectively. Such a setting will preserve the ability of those augmentation frameworks. Nevertheless, as we can see, the augmentation methods fail to generalize effectively to GAD tasks under limited supervision—the performance of the vanilla models may drop after the augmentation. In contrast, our FracAug can boost all vanilla models across real-world datasets, which demonstrates the usefulness of FracAug.

Table 4: Ablation study.

| Datasets | Metrics | GIN | +FA | w/o largest | w/o smallest | w/o WDML | w/o MVP |
|---|---|---|---|---|---|---|---|
| P388 | AUROC | 0.5565 | 0.5913 | 0.5620 | 0.5708 | 0.5730 | 0.5599 |
|  | AUPRC | 0.2850 | 0.3309 | 0.2917 | 0.3050 | 0.3074 | 0.3014 |
|  | F1-score | 0.4468 | 0.4491 | 0.4482 | 0.4470 | 0.4470 | 0.4371 |
| SF-295 | AUROC | 0.5844 | 0.6076 | 0.5971 | 0.5949 | 0.5920 | 0.5996 |
|  | AUPRC | 0.2766 | 0.2832 | 0.2723 | 0.2716 | 0.2653 | 0.2790 |
|  | F1-score | 0.4803 | 0.5047 | 0.5001 | 0.4975 | 0.4994 | 0.4973 |
| SN12C | AUROC | 0.5995 | 0.6079 | 0.5993 | 0.5963 | 0.5910 | 0.5990 |
|  | AUPRC | 0.2696 | 0.2746 | 0.2685 | 0.2666 | 0.2632 | 0.2675 |
|  | F1-score | 0.5030 | 0.5110 | 0.5041 | 0.5013 | 0.4971 | 0.5046 |
| UACC257 | AUROC | 0.5877 | 0.6015 | 0.5939 | 0.5854 | 0.5938 | 0.5914 |
|  | AUPRC | 0.2480 | 0.2598 | 0.2517 | 0.2527 | 0.2585 | 0.2586 |
|  | F1-score | 0.4906 | 0.4983 | 0.4956 | 0.4835 | 0.4890 | 0.4858 |
| PROTEINS_full | AUROC | 0.5799 | 0.6174 | 0.5986 | 0.5842 | 0.5990 | 0.5854 |
|  | AUPRC | 0.6259 | 0.6358 | 0.6111 | 0.5988 | 0.6205 | 0.6042 |
|  | F1-score | 0.5679 | 0.6175 | 0.5995 | 0.5848 | 0.5976 | 0.5857 |
| DBLP_v1 | AUROC | 0.6231 | 0.8044 | 0.7615 | 0.7828 | 0.7762 | 0.7628 |
|  | AUPRC | 0.7201 | 0.8626 | 0.8301 | 0.8411 | 0.8374 | 0.8294 |
|  | F1-score | 0.5996 | 0.8028 | 0.7594 | 0.7829 | 0.7760 | 0.7619 |

Table 5: Varying $k_l$-$k_s$-$H_l$-$H_s$ on different datasets based on GIN.

| Datasets | PROTEINS_full | | | DBLP_v1 | | |
|---|---|---|---|---|---|---|
| $k_l$-$k_s$-$H_l$-$H_s$ | AUROC | AUPRC | F1-score | AUROC | AUPRC | F1-score |
| 3-3-3-3 | 0.6174 | 0.6298 | 0.6187 | 0.7950 | 0.8514 | 0.7947 |
| 3-3-3-4 | 0.6141 | 0.6327 | 0.6142 | 0.7972 | 0.8572 | 0.7955 |
| 3-3-4-3 | 0.6103 | 0.6291 | 0.6104 | 0.7995 | 0.8546 | 0.7992 |
| 3-3-4-4 | 0.6174 | 0.6358 | 0.6175 | 0.7925 | 0.8486 | 0.7925 |
| 3-4-3-3 | 0.6174 | 0.6298 | 0.6187 | 0.7950 | 0.8514 | 0.7947 |
| 3-4-3-4 | 0.6082 | 0.6304 | 0.6072 | 0.7972 | 0.8572 | 0.7955 |
| 3-4-4-3 | 0.6103 | 0.6291 | 0.6104 | 0.7995 | 0.8546 | 0.7992 |
| 3-4-4-4 | 0.6094 | 0.6210 | 0.6104 | 0.7982 | 0.8524 | 0.7982 |
| 4-3-3-3 | 0.6174 | 0.6298 | 0.6187 | 0.7885 | 0.8509 | 0.7867 |
| 4-3-3-4 | 0.6124 | 0.6284 | 0.6133 | 0.7972 | 0.8538 | 0.7967 |
| 4-3-4-3 | 0.6161 | 0.6295 | 0.6174 | 0.8044 | 0.8626 | 0.8028 |
| 4-3-4-4 | 0.6138 | 0.6316 | 0.6142 | 0.8007 | 0.8570 | 0.8000 |
| 4-4-3-3 | 0.6174 | 0.6298 | 0.6187 | 0.7972 | 0.8579 | 0.7953 |
| 4-4-3-4 | 0.6161 | 0.6295 | 0.6174 | 0.7994 | 0.8560 | 0.7987 |
| 4-4-4-3 | 0.6108 | 0.6334 | 0.6095 | 0.8015 | 0.8617 | 0.7996 |
| 4-4-4-4 | 0.6094 | 0.6210 | 0.6104 | 0.8008 | 0.8577 | 0.7999 |

## 5.3 ABLATION STUDY

To examine the effectiveness of each component in FracAug, we conduct an ablation study on 6 datasets based on GIN (ablation study on the other 6 datasets can be found in Appendix L), which is shown in Table 4. Specifically, the "+FA" represents the performance of GIN with FracAug, where w/o largest and w/o smallest denote removing the fractional graphs generated by top-$k_l$ largest and top-$k_s$ smallest eigenvalues, respectively, w/o WDML means replacing our proposed WDML with weighted cross-entropy loss, and w/o MVP pseudo-labels samples in $\mathcal{D}_{val} \cup \mathcal{D}_{test}$ using only the predicted probability of synthetic graphs. As shown in Table 4, FracAug consistently outperforms its four variants, which demonstrates the benefits of these components.

## 5.4 HYPERPARAMETER ANALYSIS

Table 5 presents a systematic exploration of FracAug's performance, measured in AUROC, AUPRC, and F1-score, when varying $k_l, k_s, H_l, H_s$ between 3 and 4. Specifically, $k_l, k_s$ denote the top-$k_l$

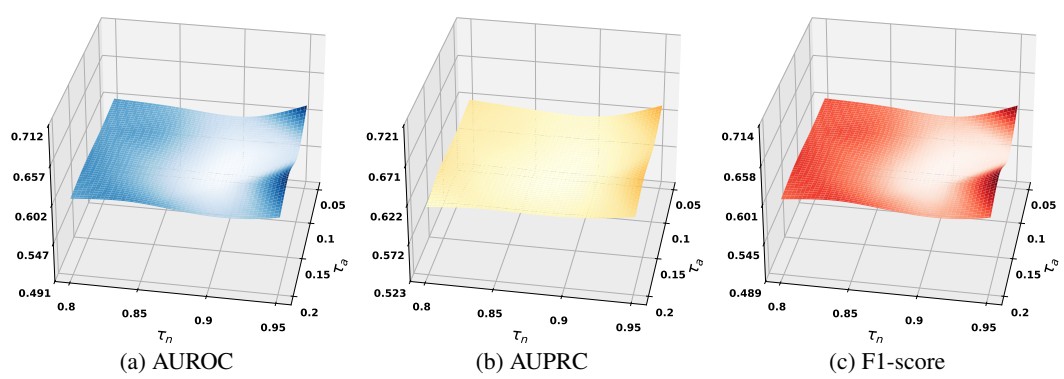

Figure 3: Varying $\tau_a$ and $\tau_n$ for PROTEINS_full based on GIN.

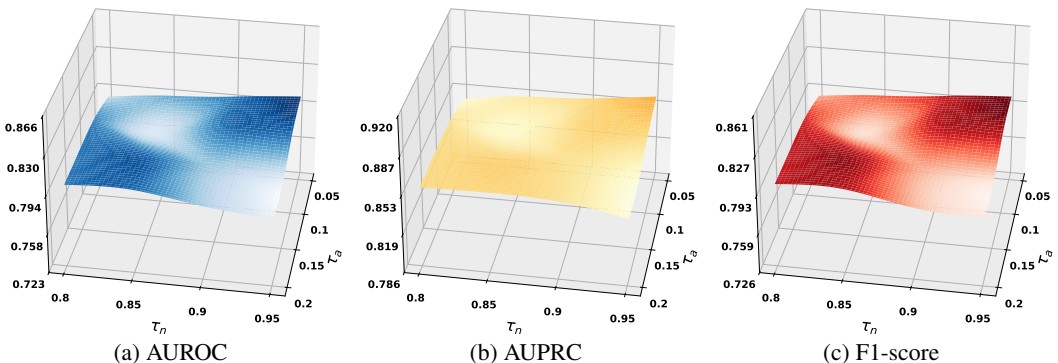

Figure 4: Varying $\tau_a$ and $\tau_n$ for DBLP_v1 based on GIN.

largest and top-$k_s$ eigenvalues generated by EVD, while $H_l, H_s$ denote the number of learnable fractional powers of the matrix for the largest and smallest eigenvalues. By sweeping each of these four parameters, we generate a compact grid of 16 configurations. As detailed in Appendix F, each evaluation metric prefers a slightly different quadruple of $(k_l, k_s, H_l, H_s)$, but more importantly, Table 5 reveals that the detection performance barely wavers across all measured combinations. This robustness not only validates the spectral augmentation strategy of FracAug but also suggests that practitioners can avoid laborious hyperparameter tuning without sacrificing performance.

In a parallel study, Figures 3 and 4 examine the sensitivity of FracAug to the pseudo-labeling thresholds $\tau_n$ and $\tau_a$. Here, $\tau_n$ specifies the percentile above which a node is considered "normal", and $\tau_a$ the percentile below which it is flagged as "anomalous". By varying $\tau_n$ from $0.8$ to $0.95$ and $\tau_a$ from $0.05$ to $0.2$, we again explore 16 threshold pairs on each dataset, logging the resulting AUROC, AUPRC, and F1-score for every pair. Remarkably, all three metrics remain essentially flat throughout this entire range, indicating that FracAug's pseudo-labeling module is forgiving of moderate threshold choices. Leveraging this stability, we adopt $\tau_n = 0.05$ and $\tau_a = 0.95$ as our default across all datasets, thereby avoiding extensive threshold tuning without affecting performance.

## 6 CONCLUSION

In this paper, we investigate the efficacy of leveraging fractional graph variants for data augmentation in GAD under limited supervision scenarios. Based on the analysis, we design a model-agnostic plug-in augmentation framework, FracAug, which includes three key components: FGG, WDML, and MVP. FGG with WDML captures semantics from original samples and then generates semantic-preserving fractional graphs during model training, unaffected by the imbalanced data distribution, while MVP employs mutual verification to enhance pseudo-labeling reliability, iteratively expanding the training set. Comprehensive experiments demonstrate that FracAug not only effectively improves the performance of any given GNN but also significantly outperforms other graph-level augmentation methods, demonstrating the effectiveness of our method.

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

## A    PROOFS

**Proof of Theorem 1.** To derive the approximation of $\boldsymbol{A}^\alpha$ and the corresponding error bound, we first consider a function for real numbers, i.e., $f(x) = x^\alpha$ defined on an interval $[a, b] \subset (0, +\infty)$. To satisfy the requirement of Chebyshev series approximation, we map $[a, b]$ to the standard Chebyshev interval $[-1, 1]$ via the linear transformation:

$$x = \frac{2}{b-a}(x' - \frac{b+a}{2}),$$

where $x' \in [a, b]$ maps to $x \in [-1, 1]$. Then we further define:

$$\tilde{f}(x) = (\frac{(b-a)x + (b+a)}{2})^\alpha,$$

which can be approximated using Chebyshev series approximation as:

$$\tilde{f}(x) \approx p_T(x) = \sum_{t=0}^{T} c_t P_t(x),$$

where $P_t(x)$ is the $t$-th Chebyshev polynomial, and the $c_t$ are the Chebyshev coefficients. Specifically, we can find the coefficients $c_t$ through the application of an inner product:

$$\int_{-1}^{+1} \frac{P_m(x)\tilde{f}(x)}{\sqrt{1-x^2}}dx = \sum_{t=0}^{\infty} c_t \int_{-1}^{+1} \frac{P_m(x)P_t(x)}{\sqrt{1-x^2}}dx.$$

On the interval $[-1, 1]$, we have:

$$\int_{-1}^{+1} \frac{P_m(x)P_t(x)}{\sqrt{1-x^2}}dx = \begin{cases} 0, & m \neq t, \\ \pi, & m = t = 0, \\ \frac{\pi}{2}, & m = t \neq 0, \end{cases}$$

so we can derive:

$$c_t = \begin{cases} \dfrac{1}{\pi} \displaystyle\int_{-1}^{+1} \frac{P_t(x)\tilde{f}(x)}{\sqrt{1-x^2}}dx, & t = 0, \\ \dfrac{2}{\pi} \displaystyle\int_{-1}^{+1} \frac{P_t(x)\tilde{f}(x)}{\sqrt{1-x^2}}dx, & t \neq 0. \end{cases}$$

Afterward, to obtain the error bound of the approximation, we leverage the following Theorem:

**Theorem 3.** *(Theorems 8.1 and 8.2 from previous work (Trefethen, 2019)) Let a function $f(x)$ analytic in $[-1, 1]$ be analytically continuable to open Bernstein ellipse $\boldsymbol{E}_\rho$, where it satisfies $|f(x)| \leq M$ for some $M$, then for each $t \geq 0$, its Chebyshev approximation $p_T(x)$ satisfies $||f(x) - p_T(x)|| \leq \frac{4M\rho^{-T}}{\rho - 1}$, where $\rho$ depends on the distance from $[-1, 1]$ to the nearest singularity of $f(x)$.*

The function $f(x) = x^\alpha$ has a branch point at $x = 0$. For $[a, b] \subset (0, +\infty)$, the mapped function $\tilde{f}(x)$ is analytic in a Bernstein ellipse $\boldsymbol{E}_\rho$, excluding $x = 0$. Therefore, $\tilde{f}(x)$ satisfies Theorem 3, so we can have:

$$||\tilde{f}(x) - p_T(x)|| \leq \frac{4M\rho^{-T}}{\rho - 1} = \beta e^{-\gamma T},$$

where $\beta = \frac{4M}{\rho - 1}$ and $\gamma = \ln \rho$.

Similarly, we can directly apply the function to $\boldsymbol{A}$ with eigenvalues in $[\lambda_{min}, \lambda_{max}] \subset (0, +\infty)$, then we can conclude:

$$||\boldsymbol{A}^\alpha - p_T(\boldsymbol{A})|| \leq \beta e^{-\gamma T},$$

where $\beta, \gamma$ is derived from $[\lambda_{min}, \lambda_{max}]$. $\qquad\square$

**Proof of Theorem 2.** Using the Dunford-Taylor integral, for a contour $\Gamma$ enclosing the spectra of $\boldsymbol{A}$ and $\boldsymbol{A} + \boldsymbol{P}$, we have:

$$\boldsymbol{A}^\alpha = \frac{1}{2\pi i} \int_\Gamma x^\alpha (x\boldsymbol{I} - \boldsymbol{A})^{-1} dx,$$

$$(\boldsymbol{A} + \boldsymbol{P})^\alpha = \frac{1}{2\pi i} \int_\Gamma x^\alpha (x\boldsymbol{I} - (\boldsymbol{A} + \boldsymbol{P}))^{-1} dx.$$

Then we subtract the two integrals:

$$\boldsymbol{A}^\alpha - (\boldsymbol{A} + \boldsymbol{P})^\alpha = \frac{1}{2\pi i} \int_\Gamma x^\alpha [(x\boldsymbol{I} - \boldsymbol{A})^{-1} - (x\boldsymbol{I} - (\boldsymbol{A} + \boldsymbol{P}))^{-1}] dx.$$

After applying the resolvent identity, we can have:

$$(x\boldsymbol{I} - \boldsymbol{A})^{-1} - (x\boldsymbol{I} - (\boldsymbol{A} + \boldsymbol{P}))^{-1} = (x\boldsymbol{I} - \boldsymbol{A})^{-1} \boldsymbol{P} (x\boldsymbol{I} - (\boldsymbol{A} + \boldsymbol{P}))^{-1}.$$

By substituting back into the integral, we have:

$$\boldsymbol{A}^\alpha - (\boldsymbol{A} + \boldsymbol{P})^\alpha = \frac{1}{2\pi i} \int_\Gamma x^\alpha (x\boldsymbol{I} - \boldsymbol{A})^{-1} \boldsymbol{P} (x\boldsymbol{I} - (\boldsymbol{A} + \boldsymbol{P}))^{-1} dx.$$

Take the operator norm and apply submultiplicativity:

$$||\boldsymbol{A}^\alpha - (\boldsymbol{A} + \boldsymbol{P})^\alpha|| \leq \frac{1}{2\pi} \int_\Gamma |x^\alpha| ||(x\boldsymbol{I} - \boldsymbol{A})^{-1}|| ||\boldsymbol{P}|| ||(x\boldsymbol{I} - (\boldsymbol{A} + \boldsymbol{P}))^{-1}|| dx|.$$

If we choose $\Gamma$ to be a contour at distance $d > 0$ from the spectra of $\boldsymbol{A}$, we can have:

$$(x\boldsymbol{I} - \boldsymbol{A})^{-1} = \boldsymbol{U}(x\boldsymbol{I} - \boldsymbol{\Lambda})^{-1}\boldsymbol{U}^T,$$

where $\boldsymbol{A} = \boldsymbol{U}\boldsymbol{\Lambda}\boldsymbol{U}^T$ is the eigendecomposition of $\boldsymbol{A}$ and the corresponding norm is:

$$||(x\boldsymbol{I} - \boldsymbol{A})^{-1}|| = ||(x\boldsymbol{I} - \boldsymbol{\Lambda})^{-1}|| = \max_{\lambda \in \sigma(\boldsymbol{A})} \frac{1}{|x - \lambda|} = \frac{1}{\text{dist}(x, \sigma(\boldsymbol{A}))} = \frac{1}{d},$$

where $\sigma(\boldsymbol{A})$ is the spectrum of $\boldsymbol{A}$ and $\text{dist}(\cdot)$ is the distance function.

For a small perturbation $||\boldsymbol{P}||$, the spectrum of $\boldsymbol{A} + \boldsymbol{P}$ will lie in a neighborhood of the spectrum of $\boldsymbol{A}$. Specifically, for any eigenvalue $\lambda'$ of $\boldsymbol{A} + \boldsymbol{P}$, there exists an eigenvalue $\lambda$ of $\boldsymbol{A}$ such that $|\lambda' - \lambda| \leq ||\boldsymbol{P}||$, which implies:

$$\text{dist}(x, \sigma(\boldsymbol{A} + \boldsymbol{P})) \geq \text{dist}(x, \sigma(\boldsymbol{A})) - ||\boldsymbol{P}||.$$

Then for $x \notin \sigma(\boldsymbol{A} + \boldsymbol{P})$, we use the Neumann series:

$$(x\boldsymbol{I} - (\boldsymbol{A} + \boldsymbol{P}))^{-1} = (x\boldsymbol{I} - \boldsymbol{A})^{-1} \sum_{i=0}^{+\infty} [\boldsymbol{P}(x\boldsymbol{I} - \boldsymbol{A})^{-1}]^i,$$

which converges if $||\boldsymbol{P}(x\boldsymbol{I} - \boldsymbol{A})^{-1}|| < 1$.

Afterward, we take its norm and apply submultiplicativity:

$$||(x\boldsymbol{I} - (\boldsymbol{A} + \boldsymbol{P}))^{-1}|| \leq \frac{||(x\boldsymbol{I} - \boldsymbol{A})^{-1}||}{1 - ||\boldsymbol{P}|| ||(x\boldsymbol{I} - \boldsymbol{A})^{-1}||}$$

$$= \frac{1}{\text{dist}(x, \sigma(\boldsymbol{A})) - ||\boldsymbol{P}||}$$

$$\leq \frac{1}{d}.$$

Then let $M = \max_{x \in \Gamma} |x^\alpha|$ and $\text{L}(\cdot)$ be the length function, we can have:

$$||\boldsymbol{A}^\alpha - (\boldsymbol{A} + \boldsymbol{P})^\alpha|| \leq \frac{1}{2\pi d^2} M ||\boldsymbol{P}|| \text{L}(\Gamma)$$

Table 6: Statistics of 12 real-world datasets, where $n_n$ is the number of normal graphs, $n_a$ is the number of anomalous graphs, $h = \frac{n_a}{n_n+n_a}$ is the anomalous ratio, $\bar{n}$ is the average number of nodes, $\bar{m}$ is the average number of edges, and $F$ is the number of attributes.

| Dataset | MCF-7 | MOLT-4 | PC-3 | SW-620 | NCI-H23 | OVCAR-8 | P388 | SF-295 | SN12C | UACC257 | PROTEINS_full | DBLP_v1 |
|---------|-------|--------|------|--------|---------|---------|------|--------|-------|---------|---------------|---------|
| $n_n$ | 25476 | 36625 | 25941 | 38122 | 38296 | 38437 | 39174 | 38246 | 38049 | 38345 | 663 | 9926 |
| $n_a$ | 2294 | 3140 | 1568 | 2410 | 2057 | 2079 | 2298 | 2025 | 1955 | 1643 | 450 | 9530 |
| $h$ | 0.0826 | 0.079 | 0.057 | 0.0595 | 0.051 | 0.0513 | 0.0554 | 0.0503 | 0.0489 | 0.0411 | 0.4043 | 0.4898 |
| $\bar{n}$ | 26.4 | 26.1 | 26.36 | 26.06 | 26.07 | 26.08 | 22.11 | 26.06 | 26.08 | 262.09 | 39.06 | 10.48 |
| $\bar{m}$ | 28.53 | 28.14 | 28.49 | 28.09 | 28.1 | 28.11 | 23.56 | 28.09 | 28.11 | 28.13 | 72.82 | 19.65 |
| $F$ | 46 | 64 | 45 | 65 | 65 | 65 | 72 | 65 | 65 | 64 | 3 | 41325 |

Define $c = \frac{ML(\Gamma)}{2\pi d^2}$, yielding $||\boldsymbol{A}^\alpha - (\boldsymbol{A} + \boldsymbol{P})^\alpha|| \leq c||\boldsymbol{P}||$.

Besides, for a generated adjacency matrix from the perturbation method, it can be diagonalized, so we can have:

$$(\boldsymbol{A} + \boldsymbol{P}) - (\boldsymbol{A} + \boldsymbol{P})^\alpha = \boldsymbol{V}(\boldsymbol{\Sigma} - \boldsymbol{\Sigma}^\alpha)\boldsymbol{V}^T,$$

where $\boldsymbol{A} + \boldsymbol{P} = \boldsymbol{V}\boldsymbol{\Sigma}\boldsymbol{V}^T$ and $\boldsymbol{\Sigma}$ is a diagonal matrix composed of $(\lambda_1, \lambda_2, \cdots, \lambda_n)$. Take the norm, we can get:

$$||(\boldsymbol{A} + \boldsymbol{P}) - (\boldsymbol{A} + \boldsymbol{P})^\alpha|| = \max_i |\lambda_i - \lambda_i^\alpha|.$$

Finally, by applying the submultiplicativity, we can conclude:

$$||\boldsymbol{A}^\alpha - (\boldsymbol{A} + \boldsymbol{P})|| \leq c||\boldsymbol{P}|| + \max_i |\lambda_i - \lambda_i^\alpha|,$$

where $c$ depends on $\alpha$ and the spectral gap of $\boldsymbol{P}$, and $\lambda_i$ is the $i$-th eigenvalue of $\boldsymbol{A} + \boldsymbol{P}$. $\qquad\square$

**Proof of Proposition 1.** Assume the two prediction error rates for original graphs and corresponding fractional graphs are two Bernoulli variables with mean $\delta$, and the correlation of the errors is $\rho$. Then we have the joint error rate:

$$\mathbb{P}(\text{Both wrong}) = \delta^2 + \rho\delta(1 - \delta).$$

Since the original error rate is $\delta$, the mutual verification will lower the error with the reduction factor $\delta + \rho\delta(1 - \delta)$.

According to the above analysis, the error rate of mutual verification is $p = \delta^2 + \rho\delta(1 - \delta)$. Assuming it is also a Bernoulli variable, the variance can be calculated as:

$$v = (\delta^2 + \rho\delta(1 - \delta))(1 - \delta^2 - \rho\delta(1 - \delta)).$$

For a small error rate $\delta$, we can approximate it as $v = \rho\delta(1 - \rho\delta)$. Therefore, the reduction factor of variance is close to $\rho$. $\qquad\square$

## B  DATASETS AND BASELINES

**Datasets.** The datasets used in our experiments are collected by TUDataset (Morris et al., 2020). Specifically, MCF-7, MOLT-4, PC-3, SW-620, NCI-H23, OVCAR-8, P388, SF-295, SN12C, and UACC257 are small-molecule datasets from PubChem [2], which provide information on the biological activities of small molecules. In these datasets, nodes represent atoms within chemical compounds, while edges indicate the chemical bonds connecting pairs of atoms. Each dataset corresponds to a specific type of cancer screening, with outcomes classified as either active or inactive. We consider inactive chemical compounds as normal graphs and active compounds as anomalous graphs. Furthermore, the attributes are derived from node labels using one-hot encoding.

Besides, PROTEINS_full is a typical bioinformatics-related dataset (Dobson & Doig, 2003), which processes several proteins represented as graphs. In this dataset, nodes and edges are formulated in a similar way to small-molecule datasets from PubChem. This dataset aims to classify enzymes and non-enzymes, which are denoted as normal and anomalous graphs, respectively.

---

[2]https://pubchem.ncbi.nlm.nih.gov/

Beyond the above datasets, we also conduct experiments on DBLP_v1 (Pan et al., 2013), which consists of bibliography data in computer science. Each record in DBLP_v1 is associated with a number of attributes such as abstract, authors, year, venue, title, and reference ID. Since the dimension of the attributes is high, we first utilize EVD to lower the dimension to 16. In this dataset, nodes denote papers, while edges represent reference relations between papers. The classification task is to predict whether a paper belongs to the CVPR (computer vision and pattern recognition) or DBDM (database and data mining) conferences, which are seen as normal and anomalous, respectively.

**Baselines.** The first group is graph classification models:

- GCN (Kipf & Welling, 2017): a GNN that uses a convolution function on a graph to propagate information within the neighborhood of nodes;
- GraphSAGE (Hamilton et al., 2017): a GNN that leverages a sampling technique to aggregate features from the neighborhood.
- GAT (Velickovic et al., 2018): a GNN that adopts an attention mechanism within the neighborhood of each node;
- GIN (Xu et al., 2019): a GNN that follows graph isomorphism to capture the properties of a graph.
- LRGNN (Wei et al., 2023): a GNN stacking multiple GNNs to extract the long-range dependencies;
- GRDL (Wang & Fan, 2024): a GNN treating node embeddings as a discrete distribution, enabling direct classification without global pooling.

The second group is GAD models:

- iGAD (Zhang et al., 2022): a GNN with a substructure-aware component to capture properties of anomalous graphs.
- GmapAD (Ma et al., 2023a): a GNN mapping graphs into a latent space where anomalies can be effectively detected;
- RQGNN (Dong et al., 2024): a GNN using Rayleigh Quotient to obtain information from both spectral and spatial spaces.
- UniGAD (Lin et al., 2024): a GNN that unifies different levels of graph-related tasks.

The third group is graph-level augmentation frameworks:

- MAA (Yoo et al., 2022): a framework using node split and merge, and subgraph mix to augment graphs heuristically;
- GLA (Yue et al., 2022): a framework augmenting data in the representation space from the most difficult direction while keeping the label of augmented data the same as the original samples;
- GMixup (Han et al., 2022): a framework that interpolates graphons of different classes in the Euclidean space to get mixed graphons;
- FGWMixup (Ma et al., 2023b): a framework that seeks a midpoint of source graphs in the Fused Gromov-Wasserstein metric space to interpolate graphons of different classes.

## C  ALGORITHM

---
**Algorithm 1:** Preprocess

---
   **Input:** $\mathcal{D}, k_l, k_s$
1 **for** $G$ *in* $\mathcal{D}$ **do**
2     $G.\boldsymbol{A} \leftarrow \frac{1}{2}(\boldsymbol{I} + G.\boldsymbol{D}^{-\frac{1}{2}} * G.\boldsymbol{A} * G.\boldsymbol{D}^{-\frac{1}{2}})$;
3     $G.\boldsymbol{U}_l, G.\boldsymbol{\Lambda}_l \leftarrow \text{EVD}(G.\boldsymbol{A}, k_l)$;
4     $G.\boldsymbol{U}_s, G.\boldsymbol{\Lambda}_s \leftarrow \text{EVD}(G.\boldsymbol{A}, k_s)$;

---

## D  TIME COMPLEXITY ANALYSIS

For the Preprocess function, we first analyze the time complexity of the matrix multiplication. Since we utilize sparse matrices to conduct the experiment, the time complexity of the multiplication is

---

**Algorithm 2:** FGG

---

**Input:** $\mathcal{D}, H_l, H_s$
**Output:** $\mathcal{D}'$

1 **for** $G$ *in* $\mathcal{D}$ **do**
2    **for** $i = 0$ *to* $H_l$ **do**
3       $\lfloor$ $G_l \leftarrow G_l + \boldsymbol{\omega}_l[i] * G.\boldsymbol{U}_l * G.\boldsymbol{\Lambda}_l^{\boldsymbol{\alpha}_l[i]} * G.\boldsymbol{U}_l^T$;
4    **for** $i = 0$ *to* $H_l$ **do**
5       $\lfloor$ $G_s \leftarrow G_s + \boldsymbol{\omega}_s[i] * G.\boldsymbol{U}_s * G.\boldsymbol{\Lambda}_s^{\boldsymbol{\alpha}_s[i]} * G.\boldsymbol{U}_s^T$;
6    $G' \leftarrow \omega * G_l + (1 - \omega) * G_s$;
7    $\mathcal{D}' \leftarrow \mathcal{D}' \cup G'$;

8 Return $\mathcal{D}'$;

---

**Algorithm 3:** WDML

---

**Input:** $f, \mathcal{D}, \mathcal{D}'$

1 **for** $G, G'$ *in* $\mathcal{D}, \mathcal{D}'$ **do**
2    $\boldsymbol{s}, \boldsymbol{o} \leftarrow f(G)$;
3    $\boldsymbol{s}', \boldsymbol{o}' \leftarrow f(G')$;
4    $m \leftarrow \frac{1 - \cos(\boldsymbol{o}, \boldsymbol{o}')}{2}$;
5    $L_{\text{WDML}} \leftarrow L_{\text{WDML}} + -\frac{1}{N_{G.y}} \log \frac{e^{\boldsymbol{s}[G.y]-m}}{e^{\boldsymbol{s}[G.y]-m}+e^{\boldsymbol{s}[1-G.y]}}$;

6 $L_{\text{WDML}}$.backward();

---

**Algorithm 4:** MVP

---

**Input:** $f, \mathcal{D}, \mathcal{D}', \tau_n, \tau_a$
**Output:** $\mathcal{D}''$

1 **for** $G, G'$ *in* $\mathcal{D}, \mathcal{D}'$ **do**
2    $\boldsymbol{s}, \boldsymbol{o} \leftarrow f(G)$;
3    $\boldsymbol{s}', \boldsymbol{o}' \leftarrow f(G')$;
4    **if** $\boldsymbol{s}[0] < \tau_n \wedge \boldsymbol{s}'[0] \leq \tau_n$ **then**
5       $G.y \leftarrow 0$;
6       $\mathcal{D}'' \leftarrow \mathcal{D}'' \cup G$;
7    **else if** $\boldsymbol{s}[1] < \tau_a \wedge \boldsymbol{s}'[1] \leq \tau_a$ **then**
8       $G.y \leftarrow 1$;
9       $\mathcal{D}'' \leftarrow \mathcal{D}'' \cup G$;

10 Return $\mathcal{D}''$;

---

**Algorithm 5:** FracAug

---

**Input:** $f, \mathcal{D}_{train}, \mathcal{D}_{val}, \mathcal{D}_{test}, H_l, H_s, k_l, k_s, e_{warmup}, e_{aug}, \tau_n, \tau_a$

1 Preprocess($\mathcal{D}_{train} \cup \mathcal{D}_{val} \cup \mathcal{D}_{test}, k_l, k_s$);
2 $D'_{train} \leftarrow D_{train}$;
3 **for** $e = 0$ *to* $e_f$ **do**
4    **if** $e > e_{warmup} \wedge e\%e_{aug} == 0$ **then**
5       **for** $e' = 0$ *to* $e_{FGG}$ **do**
6          $\mathcal{D}_{temp} \leftarrow \text{FGG}(\mathcal{D}_{train}, H_l, H_s)$;
7          $\lfloor$ WDML($f, \mathcal{D}, \mathcal{D}_{temp}$);
8       $\mathcal{D}_{temp} \leftarrow \text{FGG}(\mathcal{D}_{val} \cup \mathcal{D}_{test}, H_l, H_s)$;
9       $\mathcal{D}_{temp} \leftarrow \text{MVP}(f, \mathcal{D}_{val} \cup \mathcal{D}_{test}, \mathcal{D}_{temp}, \tau_n, \tau_a)$;
10       $D'_{train} \leftarrow D_{train} \cup \mathcal{D}_{temp}$;
11    train($f, D'_{train}$));

---

Table 7: Comparison of average running time (s).

| Datasets | NSv+FA | NodeSam | SubMix | GLAv+FA | GLA | GMixupv+FA | GMixup | FGWMixupv+FA | FGWMixup |
|---|---|---|---|---|---|---|---|---|---|
| MCF-7 | 92.18+58.56 | 1282.84 | 876.39 | 92.00+73.57 | 1493.55 | 90.54+107.51 | 51.64 | 92.29+106.76 | 553.90 |
| MOLT-4 | 133.10+83.70 | 1352.07 | 881.00 | 128.18+113.70 | 1924.35 | 131.30+150.27 | 64.55 | 129.84+176.65 | 855.30 |
| PC-3 | 91.26+70.05 | 1249.30 | 879.76 | 90.55+64.84 | 1484.25 | 90.46+110.26 | 51.52 | 90.70+98.88 | 613.19 |
| SW-620 | 133.80+102.96 | 1248.39 | 875.52 | 136.75+99.17 | 2076.36 | 133.42+158.47 | 75.68 | 133.50+153.30 | 873.41 |
| NCI-H23 | 130.20+108.73 | 1301.09 | 883.09 | 130.63+93.86 | 2091.18 | 131.54+134.20 | 63.68 | 131.07+143.53 | 896.41 |
| OVCAR-8 | 131.48+98.46 | 1351.17 | 890.68 | 132.62+102.36 | 2077.41 | 132.07+168.57 | 61.41 | 131.23+130.47 | 986.01 |
| P388 | 120.73+103.09 | 1258.91 | 884.15 | 120.43+90.74 | 2140.75 | 122.53+145.56 | 60.47 | 121.12+135.97 | 965.61 |
| SF-295 | 130.92+84.81 | 1359.48 | 879.52 | 131.26+110.89 | 2063.71 | 130.77+168.45 | 66.25 | 130.43+150.38 | 915.04 |
| SN12C | 129.79+102.16 | 1334.82 | 914.87 | 129.62+87.23 | 2044.62 | 130.21+148.75 | 68.58 | 128.95+134.45 | 933.55 |
| UACC257 | 128.63+84.19 | 1387.70 | 930.43 | 128.39+88.05 | 2053.46 | 129.04+138.45 | 58.83 | 129.65+142.53 | 891.95 |
| PROTEINS_full | 9.03+6.67 | 206.50 | 202.65 | 8.50+8.86 | 69.30 | 8.62+7.98 | 4.58 | 8.84+7.63 | 30.25 |
| DBLP_v1 | 38.86+62.06 | 1335.42 | 897.90 | 36.99+61.50 | 1123.57 | 39.22+99.74 | 30.51 | 39.15+105.83 | 155.67 |

Table 8: Comparison of average wall-clock time (s).

| Datasets | NSv+FA | NodeSam | SubMix | GLAv+FA | GLA | GMixupv+FA | GMixup | FGWMixupv+FA | FGWMixup |
|---|---|---|---|---|---|---|---|---|---|
| MCF-7 | 135.73+69.63 | 1410.86 | 912.23 | 136.19+92.94 | 1543.64 | 131.46+107.54 | 70.60 | 132.19+106.70 | 949.81 |
| MOLT-4 | 191.15+93.22 | 1433.10 | 936.19 | 194.52+127.01 | 1993.29 | 189.00+155.83 | 98.79 | 188.37+164.47 | 1487.41 |
| PC-3 | 133.72+74.67 | 1423.05 | 973.39 | 132.97+86.91 | 1524.84 | 131.72+116.66 | 68.41 | 131.70+103.64 | 968.09 |
| SW-620 | 193.47+104.12 | 1411.45 | 945.87 | 191.01+130.31 | 2138.65 | 194.62+162.24 | 102.90 | 191.97+162.04 | 1321.40 |
| NCI-H23 | 191.622+113.72 | 1417.68 | 896.13 | 191.50+118.83 | 2220.64 | 192.70+139.89 | 102.48 | 194.30+150.65 | 1438.07 |
| OVCAR-8 | 194.69+102.93 | 1459.6 | 891.68 | 192.11+127.31 | 2521.36 | 190.40+160.44 | 102.89 | 193.93+171.23 | 1430.77 |
| P388 | 171.61+103.73 | 1573.25 | 933.75 | 172.97+130.55 | 2146.63 | 171.64+146.75 | 104.16 | 175.47+195.82 | 1754.24 |
| SF-295 | 193.68+93.42 | 1640.00 | 889.18 | 195.20+129.01 | 2065.14 | 190.63+165.59 | 102.69 | 194.14+185.80 | 1469.35 |
| SN12C | 191.82+102.57 | 1429.26 | 972.91 | 192.14+125.52 | 2116.96 | 192.33+145.00 | 101.46 | 192.01+185.39 | 1378.80 |
| UACC257 | 191.56+100.38 | 1429.63 | 977.20 | 193.70+122.99 | 2110.75 | 192.43+159.39 | 102.32 | 189.07+164.62 | 1320.10 |
| PROTEINS_full | 12.90+7.600 | 213.95 | 216.55 | 12.28+8.87 | 80.33 | 12.91+9.25 | 6.65 | 12.04+8.55 | 42.86 |
| DBLP_v1 | 69.62+67.53 | 2238.69 | 1866.97 | 66.97+74.43 | 3155.02 | 69.79+99.96 | 61.35 | 66.85+104.41 | 227.15 |

$O\big(\text{nnz}(G.\boldsymbol{D}^{-\frac{1}{2}}) * \text{nnz}(G.\boldsymbol{A}) + \text{nnz}(G.\boldsymbol{D}^{-\frac{1}{2}} * G.\boldsymbol{A}) * \text{nnz}(G.\boldsymbol{D}^{-\frac{1}{2}})\big)$, where nnz means non-zero entries of the matrix. Then, by adopting EVD to only keep the top-$k_l$ largest and top-$k_s$ smallest eigenvalues, the time complexity can be $O(n * (k_l^2 + k_s^2) + m * (k_l + k_s))$, where $n, m$ is the number of nodes/edges. As shown in Algorithm 5, Preprocess can be called before the training process, and thus it won't burden the training or inference of our FracAug.

Then, we analyze the time complexity of FGG for each graph. As presented in Algorithm 2, we perform sparse matrix multiplication for every sample $H_l$ and $H_s$ times. Besides, since $G.\boldsymbol{\Lambda}$ is a diagonal matrix, the time complexity of multiplying $G.\boldsymbol{\Lambda}$ is the same as that of multiplying $G.\boldsymbol{\Lambda}^\alpha$. Therefore, the total time complexity of FGG is $O(\text{FGG}) = O(\text{nnz}(G.\boldsymbol{U}_l) * k_l + \text{nnz}(G.\boldsymbol{U}_l * G.\boldsymbol{\Lambda}_l) * \text{nnz}(G.\boldsymbol{U}_l^T) + \text{nnz}(G.\boldsymbol{U}_s) * k_s + \text{nnz}(G.\boldsymbol{U}_s * G.\boldsymbol{\Lambda}_s) * \text{nnz}(G.\boldsymbol{U}_s^T))$.

Next, we analyze the time complexity of WDML in Algorithm 3. Assuming that we only have one sample in $\mathcal{D}_{train}$, then the time complexity of WDML is $O(\text{WDML}) = O(d)$, where $d$ is the dimension of the generated graph embedding $\boldsymbol{o}$.

Moreover, as shown in Algorithm 4, in MVP, we only need to see if the probability predicted by the given GNN satisfies the criterion, so the time complexity for MVP is $O(\text{MVP}) = O(1)$.

Finally, in Algorithm 5, we combine all the time complexities together within one training epoch of the given GNN $f$, assuming the time complexity of $f$ for each sample is $O(f)$, then we have the total complexity as $O(e_{\text{FGG}} * (O(\text{FGG}) + O(\text{WDML}) + O(f)) * N_{train} + (O(\text{FGG}) + O(\text{MVP}) + O(f)) * (N_{val} + N_{test})$, where $N_{val}, N_{test}$ represent the number of samples in $\mathcal{D}_{val}$ and $\mathcal{D}_{test}$, respectively.

In practice, we set $e_{FGG}$ to 10 and $e_{aug}$ to 25, which can reduce the computational cost, and FGG can still converge. According to the final complexity, we can see the dominant factor within each epoch is $O(e_{FGG} * O(f) * N_{train} + O(f) * (N_{val} + N_{test}))$. For such a factor, we need to calculate it in total $\frac{e_f - e_{warmup}}{e_{aug}} * e_{FGG}$ times, which is much less than the original training epoch of $f$. Hence, the increase in time complexity will not be the limitation of our FracAug in real applications.

In Tables 7 and 8, we present a detailed runtime comparison and wall-clock time comparison between FracAug and several leading graph augmentation methods, respectively. For each technique, we decompose the total computational cost into a one-time preprocessing phase, performed once per

Table 9: Comparison of average consumed memory (MB).

| Datasets | NSv+FA | NodeSam | SubMix | GLAv+FA | GLA | GMixupv+FA | GMixup | FGWMixupv+FA | FGWMixup |
|---|---|---|---|---|---|---|---|---|---|
| MCF-7 | 171.99+530.47 | 655.12 | 689.79 | 171.88+687.56 | 1125.73 | 172.20+538.38 | 675.25 | 172.32+552.62 | 598.73 |
| MOLT-4 | 240.71+534.24 | 763.71 | 784.25 | 239.91+672.44 | 1262.11 | 239.92+538.74 | 770.23 | 239.85+573.34 | 817.38 |
| PC-3 | 170.76+530.65 | 652.03 | 687.16 | 170.68+661.78 | 1118.31 | 170.50+533.72 | 671.04 | 170.77+505.36 | 602.24 |
| SW-620 | 243.48+530.75 | 764.33 | 801.05 | 241.11+666.00 | 1276.12 | 243.46+531.57 | 769.41 | 249.02+546.39 | 838.00 |
| NCI-H23 | 248.02+531.50 | 761.09 | 798.24 | 247.96+678.73 | 1300.57 | 250.67+534.30 | 772.79 | 247.66+549.04 | 836.02 |
| OVCAR-8 | 248.82+526.89 | 764.66 | 800.23 | 248.93+669.54 | 1247.54 | 243.27+536.46 | 771.64 | 243.43+550.80 | 805.35 |
| P388 | 249.69+531.71 | 762.02 | 800.58 | 249.85+678.52 | 1292.48 | 249.46+534.69 | 765.74 | 249.85+573.38 | 807.48 |
| SF-295 | 243.00+527.81 | 761.71 | 799.39 | 247.26+669.17 | 1273.55 | 243.53+529.14 | 774.14 | 242.94+548.67 | 793.34 |
| SN12C | 240.86+535.26 | 765.17 | 787.12 | 240.62+689.41 | 1267.07 | 240.81+534.07 | 783.81 | 240.57+550.13 | 788.09 |
| UACC257 | 245.70+536.63 | 765.68 | 786.74 | 240.70+690.96 | 1263.72 | 240.79+542.29 | 787.33 | 240.82+501.84 | 825.58 |
| PROTEINS_full | 36.85+553.28 | 455.68 | 491.23 | 36.88+707.05 | 746.64 | 36.93+549.06 | 462.45 | 36.87+561.18 | 79.47 |
| DBLP_v1 | 129.30+526.47 | 810.82 | 865.18 | 129.12+697.11 | 990.82 | 129.04+539.86 | 604.73 | 129.20+552.64 | 417.28 |

Table 10: Comparison of average peak memory (MB).

| Datasets | NSv+FA | NodeSam | SubMix | GLAv+FA | GLA | GMixupv+FA | GMixup | FGWMixupv+FA | FGWMixup |
|---|---|---|---|---|---|---|---|---|---|
| MCF-7 | 62.53 | 55.84 | 52.07 | 73.49 | 143.10 | 60.55 | 51.83 | 65.46 | 137.32 |
| MOLT-4 | 89.45 | 78.66 | 73.17 | 95.55 | 192.97 | 86.34 | 79.51 | 90.78 | 194.61 |
| PC-3 | 61.93 | 54.51 | 51.37 | 71.91 | 142.55 | 60.67 | 51.40 | 72.49 | 137.59 |
| SW-620 | 91.19 | 80.08 | 74.80 | 85.14 | 196.15 | 93.25 | 81.22 | 93.69 | 197.01 |
| NCI-H23 | 90.79 | 79.15 | 74.23 | 99.11 | 195.42 | 87.77 | 81.13 | 96.70 | 197.96 |
| OVCAR-8 | 91.16 | 80.31 | 74.50 | 97.86 | 196.09 | 92.59 | 81.50 | 94.43 | 195.91 |
| P388 | 93.28 | 82.13 | 76.83 | 103.88 | 200.72 | 95.81 | 90.00 | 96.29 | 208.33 |
| SF-295 | 90.61 | 78.46 | 74.34 | 94.73 | 195.08 | 86.58 | 81.05 | 95.56 | 196.82 |
| SN12C | 90.02 | 78.43 | 73.81 | 96.41 | 193.95 | 90.44 | 80.36 | 94.84 | 199.04 |
| UACC257 | 89.98 | 79.26 | 74.08 | 96.52 | 193.98 | 88.65 | 80.81 | 90.62 | 194.10 |
| PROTEINS_full | 4.89 | 5.83 | 5.63 | 4.97 | 6.51 | 5.01 | 7.02 | 4.95 | 7.35 |
| DBLP_v1 | 45.35 | 38.40 | 37.67 | 48.54 | 95.30 | 43.37 | 29.05 | 46.59 | 93.41 |

dataset, and the subsequent training time measured over multiple epochs. While some baselines require repeated feature perturbations or costly online sampling at every iteration, FracAug's eigenvalue decomposition is only performed during preprocessing. As a result, the per-epoch training overhead of FracAug remains on par with, or even below, that of competing approaches, despite leveraging additional spectral information to boost anomaly detection performance.

Crucially, these efficiency gains do not come at the expense of detection performance. Across all datasets and baseline comparisons, FracAug consistently delivers state-of-the-art AUROC, AUPRC, and F1-score results while maintaining competitive total runtimes. By amortizing the heavier spectral computations over the entire training cycle and by implementing optimized matrix operations, FracAug strikes an effective balance between computational tractability and augmentation quality. This combination of speed and performance underscores the practical value of our method: practitioners can readily adopt FracAug for real graph applications without incurring prohibitive time costs.

# E  MEMORY COMPLEXITY ANALYSIS

In Tables 9 and 10, we report detailed comparisons of total and peak memory consumption for FracAug against several leading graph augmentation methods. For each method in Table 9, we further break down the total memory usage into a one-time preprocessing stage, executed once per dataset, and the recurring memory cost measured across training epochs. For each method in Table 10, we present the peak memory during the training procedure. Unlike baselines that repeatedly perturb features or perform expensive online sampling at every iteration, FracAug requires eigenvalue decomposition only during preprocessing. Consequently, its per-epoch training overhead remains comparable to, or even lower than, that of competing methods, despite incorporating additional spectral information to enhance anomaly detection.

Importantly, these memory efficiency benefits do not compromise detection quality. Across all datasets and baselines, FracAug consistently achieves state-of-the-art AUROC, AUPRC, and F1-scores while maintaining competitive end-to-end memory cost. By amortizing the heavier spectral computations across the full training process and employing optimized matrix operations, FracAug achieves a strong balance between memory efficiency and augmentation quality. This blend of

Table 11: Hyperparameters of 12 datasets based on GIN.

| Datasets | MCF-7 | MOLT-4 | PC-3 | SW-620 | NCI-H23 | OVCAR-8 | P388 | SF-295 | SN12C | UACC257 | PROTEINS_full | DBLP_v1 |
|---|---|---|---|---|---|---|---|---|---|---|---|---|
| $k_l$ | 4 | 4 | 4 | 4 | 3 | 3 | 4 | 3 | 4 | 4 | 3 | 4 |
| $H_l$ | 4 | 3 | 3 | 3 | 3 | 3 | 4 | 4 | 3 | 4 | 4 | 4 |
| $k_s$ | 3 | 4 | 3 | 3 | 4 | 3 | 4 | 3 | 3 | 4 | 3 | 3 |
| $H_s$ | 4 | 4 | 3 | 3 | 3 | 3 | 4 | 3 | 3 | 4 | 4 | 3 |
| $e_{warmup}$ | 50 | 25 | 50 | 50 | 25 | 25 | 50 | 50 | 25 | 50 | 50 | 25 |

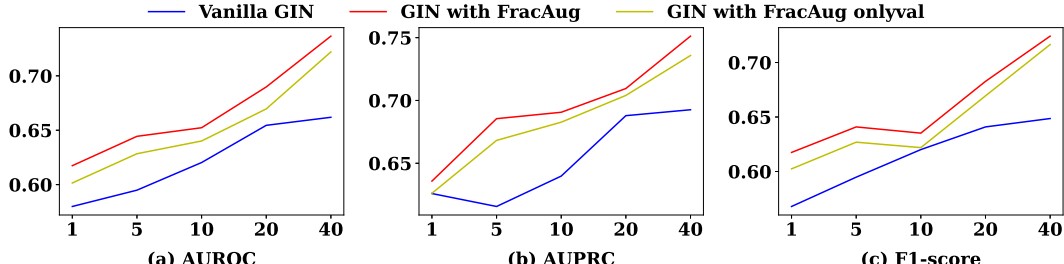

Figure 5: Varying training size (%) for PROTEINS_full.

memory usage and performance highlights the method's practical value: practitioners can deploy FracAug in real-world graph applications without incurring prohibitive memory costs.

## F  EXPERIMENTAL SETTINGS

Table 11 provides a comprehensive summary of the hyperparameter configurations used in our experiments based on the GIN backbone. To obtain the final settings for FracAug, we perform a grid search and select the configuration that achieves the highest combined AUROC, AUPRC, and F1-score on the validation set, after which we report the corresponding test performance. Concretely, the parameters $k_l, H_l, k_s, H_s$ are each chosen from the set $3, 4$, and the number of GNN warmup epochs is selected from $25, 50$, which helps keep the overall search cost manageable while still ensuring sufficient coverage of plausible configurations. For fairness and reproducibility, all experiments are conducted on an NVIDIA Quadro RTX 8000, ensuring a consistent computational environment across all evaluations.

## G  MARGIN LOSS COMPARISON

Next, we examine the performance of FracAug under different margin losses, as discussed in Section 4.3, by conducting comparisons across 12 datasets using the GIN backbone. As reported in Table 12, FracAug consistently surpasses its variants that employ Softmax, LMCL, or LDAM as alternative margin formulations. This consistent advantage highlights the effectiveness of our design and reinforces the idea that sample-specific decision boundaries provide a more suitable and flexible mechanism than fixed margins for graph-level anomaly detection. The results align closely with our theoretical analysis in Section 4.3, further validating the motivation and significance of adopting adaptive, distance-aware margins within the FracAug framework.

## H  PERFORMANCE WITH MORE TRAINING DATA

To further demonstrate the superior capability of our proposed FracAug framework, we conduct additional experiments on PROTEINS_full and DBLP_v1 under varying training sizes (%), as illustrated in Figures 5 and 6. Across all training-size configurations, the red curves, representing the performance of GIN equipped with FracAug, consistently appear at the top of the plots. This clear and repeated trend shows that as more training data becomes available, FracAug continues to deliver steady performance gains over the baseline in terms of AUROC, AUPRC, and F1-score. Overall, these results provide additional evidence that FracAug offers a robust, broadly applicable,

Table 12: Comparison of different margin loss.

| Datasets | Metrics | GIN | +FA | Softmax | LMCL | LDAM |
|---|---|---|---|---|---|---|
| MCF-7 | AUC | 0.5867 | 0.5976 | 0.5844 | 0.5880 | 0.5844 |
| | AUPRC | 0.2830 | 0.2971 | 0.2781 | 0.2847 | 0.2796 |
| | MF1 | 0.5366 | 0.5421 | 0.5378 | 0.5372 | 0.5360 |
| MOLT-4 | AUC | 0.5733 | 0.5854 | 0.5760 | 0.5797 | 0.5751 |
| | AUPRC | 0.2830 | 0.3001 | 0.2969 | 0.2952 | 0.2857 |
| | MF1 | 0.5072 | 0.5103 | 0.4991 | 0.5059 | 0.5076 |
| PC-3 | AUC | 0.5969 | 0.6119 | 0.6021 | 0.6037 | 0.6001 |
| | AUPRC | 0.2797 | 0.2893 | 0.2809 | 0.2778 | 0.2769 |
| | MF1 | 0.5063 | 0.5205 | 0.5134 | 0.5195 | 0.5144 |
| SW-620 | AUC | 0.5938 | 0.6004 | 0.5947 | 0.5936 | 0.5931 |
| | AUPRC | 0.2776 | 0.2813 | 0.2779 | 0.2768 | 0.2720 |
| | MF1 | 0.5090 | 0.5155 | 0.5100 | 0.5092 | 0.5133 |
| NCI-H23 | AUC | 0.5897 | 0.5968 | 0.5913 | 0.5896 | 0.5887 |
| | AUPRC | 0.2566 | 0.2659 | 0.2650 | 0.2612 | 0.2622 |
| | MF1 | 0.5059 | 0.5073 | 0.5001 | 0.5012 | 0.4990 |
| OVCAR-8 | AUC | 0.5935 | 0.5963 | 0.5905 | 0.5890 | 0.5932 |
| | AUPRC | 0.2573 | 0.2612 | 0.2557 | 0.2570 | 0.2579 |
| | MF1 | 0.5118 | 0.5123 | 0.5087 | 0.5051 | 0.5107 |
| P388 | AUC | 0.5565 | 0.5913 | 0.5864 | 0.5653 | 0.5602 |
| | AUPRC | 0.2850 | 0.3309 | 0.3265 | 0.3080 | 0.2901 |
| | MF1 | 0.4468 | 0.4491 | 0.4465 | 0.4377 | 0.4469 |
| SF-295 | AUC | 0.5844 | 0.6076 | 0.5943 | 0.5898 | 0.5955 |
| | AUPRC | 0.2766 | 0.2832 | 0.2664 | 0.2712 | 0.2715 |
| | MF1 | 0.4803 | 0.5047 | 0.5017 | 0.4909 | 0.4985 |
| SN12C | AUC | 0.5995 | 0.6079 | 0.5914 | 0.5955 | 0.6004 |
| | AUPRC | 0.2696 | 0.2746 | 0.2661 | 0.2601 | 0.2707 |
| | MF1 | 0.5030 | 0.5110 | 0.4950 | 0.5068 | 0.5034 |
| UACC257 | AUC | 0.5877 | 0.6015 | 0.5905 | 0.5899 | 0.5935 |
| | AUPRC | 0.2480 | 0.2598 | 0.2584 | 0.2581 | 0.2557 |
| | MF1 | 0.4906 | 0.4983 | 0.4849 | 0.4844 | 0.4912 |
| PROTEINS_full | AUC | 0.5799 | 0.6174 | 0.6002 | 0.5937 | 0.6051 |
| | AUPRC | 0.6259 | 0.6358 | 0.6235 | 0.6078 | 0.6231 |
| | MF1 | 0.5679 | 0.6175 | 0.5987 | 0.5945 | 0.6051 |
| DBLP_v1 | AUC | 0.6231 | 0.8044 | 0.7776 | 0.7834 | 0.7874 |
| | AUPRC | 0.7201 | 0.8626 | 0.8383 | 0.8463 | 0.8502 |
| | MF1 | 0.5996 | 0.8028 | 0.7774 | 0.7817 | 0.7855 |

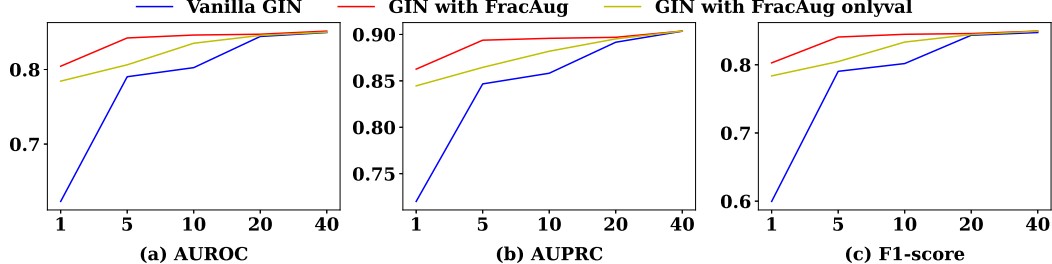

Figure 6: Varying training size (%) for DBLP_v1.

and practical enhancement to existing models, maintaining its advantages even as the amount of supervised data varies.

In addition, we examine the inductive setting, where only the validation set is used for pseudo-labeling, to evaluate the generalization capabilities of our proposed framework, FracAug. This analysis is represented by the yellow curves in Figures 5 and 6. Across all training-size configurations, the yellow curves, which depict the performance of the GIN model enhanced with FracAug under this setting, consistently outperform the vanilla model. While the yellow curve falls slightly below the red curve due to the reduced amount of data available, the performance gap is minimal. This further

Table 13: Learned parameters

| Datasets | | MCF-7 | MOLT-4 | PC-3 | SW-620 | NCI-H23 | OVCAR-8 | P388 | SF-295 | SN12C | UACC257 | PROTEINS_full | DBLP_v1 |
|---|---|---|---|---|---|---|---|---|---|---|---|---|---|
| | $\boldsymbol{\alpha}_l$ | 0.9062 | 1.4187 | 1.0862 | 2.186 | 0.4991 | 0.8986 | 2.8806 | 0.8845 | 2.7404 | 1.6178 | 1.1407 | 1.8663 |
| | | 2.0063 | 2.2813 | 1.9282 | 1.5108 | 0.3535 | 0.1262 | 2.9620 | 2.9038 | 0.5877 | 1.9339 | 0.6885 | 1.1374 |
| | | 1.4770 | 1.7911 | 1.9928 | 1.8073 | 2.3672 | 1.1480 | 2.3129 | 2.3939 | 1.0219 | 2.4874 | 2.5432 | 1.1843 |
| | | 1.4269 | - | - | - | - | - | 1.2674 | 2.4786 | - | 2.2634 | 1.4364 | 2.8254 |
| | $\boldsymbol{\omega}_l$ | 0.2921 | 0.2655 | 0.3450 | 0.2667 | 0.2793 | 0.3476 | 0.1639 | 0.3051 | 0.2385 | 0.1165 | 0.3842 | 0.1685 |
| | | 0.3108 | 0.3101 | 0.3349 | 0.5068 | 0.4612 | 0.3615 | 0.3654 | 0.1747 | 0.3833 | 0.2806 | 0.1535 | 0.1714 |
| | | 0.1549 | 0.4244 | 0.3202 | 0.2266 | 0.2595 | 0.2909 | 0.2728 | 0.2145 | 0.3782 | 0.3028 | 0.2927 | 0.4004 |
| | | 0.2422 | - | - | - | - | - | 0.1979 | 0.3057 | - | 0.3000 | 0.1696 | 0.2597 |
| | $\boldsymbol{\alpha}_s$ | 2.3128 | 1.9621 | 2.8048 | 1.5579 | 2.4828 | 0.5030 | 1.5344 | 1.6912 | 2.9331 | 0.8959 | 1.1290 | 1.7201 |
| | | 2.1539 | 0.4087 | 1.1722 | 1.8088 | 2.1366 | 1.4235 | 1.552 | 1.9896 | 1.5150 | 1.9589 | 0.1467 | 2.8101 |
| | | 0.7485 | 2.9168 | 1.6098 | 0.4474 | 2.2668 | 2.6830 | 2.0318 | 1.8741 | 2.9352 | 2.8844 | 0.5492 | 2.0935 |
| | | 2.5362 | 0.2449 | - | - | - | - | 2.7135 | - | - | 0.1148 | 0.0124 | - |
| | $\boldsymbol{\omega}_s$ | 0.3042 | 0.2821 | 0.3910 | 0.3756 | 0.3292 | 0.3384 | 0.1696 | 0.4052 | 0.4554 | 0.2916 | 0.2407 | 0.2517 |
| | | 0.2569 | 0.2112 | 0.2750 | 0.4245 | 0.3630 | 0.3436 | 0.4001 | 0.2386 | 0.2343 | 0.2539 | 0.3285 | 0.4758 |
| | | 0.2764 | 0.1519 | 0.3340 | 0.1999 | 0.3078 | 0.3180 | 0.2027 | 0.3562 | 0.3103 | 0.3122 | 0.2524 | 0.2725 |
| | | 0.1625 | 0.3548 | - | - | - | - | 0.2277 | - | - | 0.1423 | 0.1785 | - |
| | $\boldsymbol{\omega}$ | 0.4634 | 0.6795 | 0.6481 | 0.5579 | 0.5263 | 0.4541 | 0.4408 | 0.5589 | 0.5724 | 0.5309 | 0.7134 | 0.5174 |
| | | 0.5366 | 0.3205 | 0.3519 | 0.4421 | 0.4737 | 0.5459 | 0.5592 | 0.4411 | 0.4276 | 0.4691 | 0.2866 | 0.4826 |

underscores the viability of our proposed framework in strictly inductive scenarios. Overall, these findings highlight that FracAug provides a robust, versatile, and practical improvement to existing models, maintaining its effectiveness even when limited to the validation set for pseudo-labeling.

## I  LEARNED PARAMETERS

Here, we present the learned parameters based on GIN, $\boldsymbol{\alpha}_l$, $\boldsymbol{\omega}_l$, $\boldsymbol{\alpha}_s$, $\boldsymbol{\omega}_s$, and $\boldsymbol{\omega}$, as summarized in Table 13. Specifically, $\boldsymbol{\alpha}_l$ and $\boldsymbol{\alpha}_s$ correspond to the learned fractional powers associated with the largest and smallest eigenvalue groups, respectively, while $\boldsymbol{\omega}_l$ and $\boldsymbol{\omega}_s$ denote the learned coefficients applied to the fractional graphs generated from these eigenvalues. The vector $\boldsymbol{\omega}$ represents the balancing coefficient between the two groups of fractional graphs derived from the largest and smallest eigenvalues.

As described in Section 4.2, the hyperparameters $H_l$ and $H_s$ determine the number of fractional graphs combined within each eigenvalue group. Consequently, the dimensionality of $\boldsymbol{\alpha}_l$ and $\boldsymbol{\omega}_l$ equals $H_l$, and that of $\boldsymbol{\alpha}_s$ and $\boldsymbol{\omega}_s$ equals $H_s$. Since we only consider two eigenvalue groups, the largest and the smallest, the vector $\boldsymbol{\omega}$ always contains two entries. Moreover, as indicated by Table 11, the optimal values of $H_l$ and $H_s$ vary across datasets, leading to corresponding variations in the number of entries reported in Table 13. In cases where a particular entry does not exist due to the selection of $H_l$ or $H_s$, we mark it with "–" for clarity.

From Table 13, we observe that the learned fractional powers fall within the range $(0, 3)$, which is consistent with common practices in GNN design: excessively large effective depths can cause over-smoothing, whereas overly small depths may lead to under-fitting. This observation further validates the rationality of our proposed FracAug, demonstrating that the learned parameters remain well-aligned with established principles while effectively capturing dataset-specific structural information.

## J  ADDITIONAL EXPERIMENTAL RESULTS

We present the additional experimental results for MCF-7, MOLT-4, PC-3, SW-620, NCI-H23, and OVCAR-8 to further demonstrate our superiority over other baselines. On one hand, as shown in Tables 14 and 15, FracAug consistently boosts the performance of a variety of backbone models in both the graph classification and graph-level anomaly detection methods. This pattern holds across all the newly included datasets, reinforcing the generalization ability of our proposed framework and showing that its benefits are not restricted to specific architectures or data distributions. On the other hand, as shown in Table 16, FracAug also outperforms other existing graph augmentation techniques, even when those techniques are evaluated alongside their own native backbones. This result highlights the effectiveness and robustness of our framework, indicating that FracAug can

Table 14: Average AUROC, AUPRC, and F1-score on 6 datasets with multiple runs, using graph classification models as baselines, where the white columns represent vanilla models and the "+FA" represent models augmented by FracAug.

| Datasets | Metrics | GCN | +FA | SAGE | +FA | GAT | +FA | GIN | +FA | LRGNN | +FA | GRDL | +FA |
|---|---|---|---|---|---|---|---|---|---|---|---|---|---|
| MCF-7 | AUROC | 0.5753 | **0.5840** | 0.5801 | **0.5930** | 0.5885 | **0.6058** | 0.5867 | **0.5976** | 0.5467 | **0.6117** | 0.5867 | **0.6197** |
| | AUPRC | 0.3035 | **0.3073** | 0.3316 | **0.3340** | 0.3678 | **0.3814** | 0.2830 | **0.2971** | 0.2951 | **0.3309** | 0.3038 | **0.3469** |
| | F1-score | 0.4982 | **0.5074** | 0.4798 | **0.4963** | 0.4573 | **0.4687** | 0.5366 | **0.5421** | 0.4660 | **0.5290** | 0.5147 | **0.5254** |
| MOLT-4 | AUROC | 0.5531 | **0.5650** | 0.5326 | **0.5727** | 0.5403 | **0.5721** | 0.5733 | **0.5854** | 0.5495 | **0.5851** | 0.5858 | **0.5924** |
| | AUPRC | 0.3183 | **0.3217** | 0.1907 | **0.2602** | 0.3456 | **0.3667** | 0.2830 | **0.3001** | 0.3047 | **0.3175** | 0.3018 | **0.3101** |
| | F1-score | 0.4501 | **0.4626** | 0.5166 | **0.5277** | 0.4096 | **0.4313** | 0.5072 | **0.5103** | 0.4570 | **0.4939** | 0.5091 | **0.5117** |
| PC-3 | AUROC | 0.5697 | **0.5863** | 0.5986 | **0.6119** | 0.5707 | **0.5865** | 0.5969 | **0.6119** | 0.5690 | **0.6102** | 0.6044 | **0.6202** |
| | AUPRC | 0.2740 | **0.2837** | 0.3154 | **0.3248** | 0.3562 | **0.3626** | 0.2797 | **0.2893** | 0.2320 | **0.3038** | 0.3381 | **0.3459** |
| | F1-score | 0.4745 | **0.4876** | 0.4751 | **0.4841** | 0.4036 | **0.4166** | 0.5063 | **0.5205** | **0.5103** | 0.5024 | 0.4615 | **0.4750** |
| SW-620 | AUROC | 0.5662 | **0.5839** | 0.5800 | **0.5968** | 0.5633 | **0.5870** | 0.5938 | **0.6004** | 0.5758 | **0.5946** | 0.6005 | **0.6046** |
| | AUPRC | 0.3134 | **0.3229** | **0.3401** | 0.3260 | 0.2481 | **0.2587** | 0.2776 | **0.2813** | **0.3506** | 0.3386 | **0.2908** | 0.2901 |
| | F1-score | 0.4406 | **0.4541** | 0.4339 | **0.4678** | 0.4923 | **0.5187** | 0.5090 | **0.5155** | 0.4190 | **0.4536** | 0.5059 | **0.5135** |
| NCI-H23 | AUROC | 0.5811 | **0.5864** | 0.5765 | **0.6105** | 0.5777 | **0.6084** | 0.5897 | **0.5968** | 0.6002 | **0.6315** | 0.6161 | **0.6271** |
| | AUPRC | 0.2777 | **0.2777** | 0.3197 | **0.3207** | **0.2966** | 0.2945 | 0.2566 | **0.2659** | 0.2830 | **0.3205** | 0.3034 | **0.3069** |
| | F1-score | 0.4751 | **0.4819** | 0.4333 | **0.4740** | 0.4548 | **0.4961** | 0.5059 | **0.5073** | 0.4987 | **0.5055** | 0.4983 | **0.5112** |
| OVCAR-8 | AUROC | 0.5692 | **0.5809** | 0.5763 | **0.5836** | 0.5396 | **0.5784** | 0.5935 | **0.5963** | 0.5628 | **0.5984** | 0.6230 | **0.6311** |
| | AUPRC | 0.3240 | **0.3263** | 0.3391 | **0.3423** | 0.2010 | **0.2401** | 0.2573 | **0.2612** | 0.3106 | **0.3290** | 0.3042 | **0.3129** |
| | F1-score | 0.4216 | **0.4334** | 0.4163 | **0.4221** | 0.4855 | **0.5060** | 0.5118 | **0.5123** | 0.4260 | **0.4518** | 0.5087 | **0.5119** |

Table 15: Average AUROC, AUPRC, and F1-score on 6 datasets with multiple runs, using GAD models as baselines, where the white columns represent vanilla models and the "+FA" represent models augmented by FracAug.

| Datasets | Metrics | iGAD | +FA | GmapAD | +FA | RQGNN | +FA | UniGAD | +FA |
|---|---|---|---|---|---|---|---|---|---|
| MCF-7 | AUROC | 0.5670 | **0.5756** | 0.5159 | **0.5342** | 0.5522 | **0.5709** | 0.5380 | **0.5480** |
| | AUPRC | 0.3402 | **0.3525** | 0.3521 | **0.3577** | 0.2379 | **0.2648** | 0.2405 | **0.2527** |
| | F1-score | 0.4546 | **0.4549** | 0.3776 | **0.3961** | 0.5609 | **0.5804** | 0.4959 | **0.5008** |
| MOLT-4 | AUROC | 0.5562 | **0.5573** | 0.5100 | **0.5374** | 0.5576 | **0.5696** | 0.5353 | **0.5445** |
| | AUPRC | 0.3088 | **0.3092** | 0.2704 | **0.2827** | 0.2302 | **0.2468** | 0.2111 | **0.2241** |
| | F1-score | 0.4621 | **0.4636** | 0.4354 | **0.4592** | 0.5644 | **0.5718** | 0.5079 | **0.5125** |
| PC-3 | AUROC | 0.5526 | **0.5674** | 0.5112 | **0.5266** | 0.5618 | **0.6043** | 0.5496 | **0.5559** |
| | AUPRC | 0.1887 | **0.2159** | 0.2999 | **0.3094** | 0.2370 | **0.2663** | 0.3875 | **0.3889** |
| | F1-score | 0.5217 | **0.5229** | 0.3845 | **0.3938** | 0.5721 | **0.5972** | 0.3461 | **0.3542** |
| SW-620 | AUROC | 0.5641 | **0.5776** | 0.5279 | **0.5362** | 0.5428 | **0.5692** | 0.5427 | **0.5688** |
| | AUPRC | 0.3332 | **0.3701** | **0.3506** | 0.3479 | 0.1936 | **0.2219** | 0.2527 | **0.2585** |
| | F1-score | **0.4207** | 0.4029 | 0.3605 | **0.3734** | 0.5530 | **0.5654** | 0.4600 | **0.4903** |
| NCI-H23 | AUROC | 0.5689 | **0.5721** | 0.5289 | **0.5489** | 0.5704 | **0.6061** | 0.5694 | **0.5860** |
| | AUPRC | 0.2267 | **0.2288** | 0.3274 | **0.3401** | 0.2166 | **0.2500** | 0.2733 | **0.2756** |
| | F1-score | 0.5039 | **0.5066** | 0.3710 | **0.3827** | 0.5770 | **0.5820** | 0.4645 | **0.4831** |
| OVCAR-8 | AUROC | 0.5609 | **0.5685** | 0.5209 | **0.5243** | 0.5549 | **0.5773** | 0.5360 | **0.5445** |
| | AUPRC | 0.2319 | **0.2325** | **0.2863** | 0.2836 | 0.1933 | **0.2216** | **0.3196** | 0.3112 |
| | F1-score | 0.4880 | **0.4991** | 0.3992 | **0.4054** | 0.5618 | **0.5794** | 0.3873 | **0.4043** |

provide complementary advantages regardless of the underlying model design or augmentation strategy used.

## K  STANDARD DEVIATION FOR EXPERIMENTAL RESULTS

Due to the limited space, we present the standard deviation for our experimental results here in Tables 17, 18, and 19. Across the three tables reporting standard deviations, for graph classification, graph-level anomaly detection, and data augmentation, we observe a consistent stabilizing effect introduced by FracAug. Although FracAug serves as a plug-in framework that can be seamlessly integrated with diverse baselines, its impact is both clear and substantial. In most cases, the incorporation of FracAug not only improves AUROC, AUPRC, and F1-score, as shown in Tables 1, 2, 3, 14, 15, and 16, but also reduces the variance of these metrics, as shown in Tables 17, 18, and 19. This dual benefit indicates that FracAug enhances both the effectiveness and the reliability of downstream models. By pseudo-labeling the semantic-preserving graphs, FracAug helps models converge to more stable

Table 16: Average AUROC, AUPRC, and F1-score on 6 datasets with multiple runs, using graph-level augmentation models as baselines, where the white columns represent vanilla models and their own augmentation method, while the "+FA" represent vanilla models augmented by FracAug.

| Datasets | Metrics | MAAv | NodeSam | SubMix | +FA | GLAv | GLA | +FA | GMixupv | GMixup | +FA | FGWMixupv | FGWMixup | +FA |
|---|---|---|---|---|---|---|---|---|---|---|---|---|---|---|
| MCF-7 | AUROC | 0.5496 | **0.5727** | 0.5428 | 0.5695 | 0.5797 | 0.5735 | **0.6068** | 0.5730 | 0.5581 | **0.5935** | 0.5731 | 0.5502 | **0.5828** |
| | AUPRC | 0.2346 | 0.2445 | 0.2224 | **0.2455** | 0.2558 | 0.2456 | **0.2944** | 0.2767 | 0.2864 | **0.3081** | 0.2720 | 0.2130 | **0.2945** |
| | F1-score | 0.5179 | 0.5635 | 0.5512 | **0.5721** | 0.5607 | 0.5609 | **0.5793** | 0.5186 | 0.4879 | **0.5220** | 0.5585 | 0.5283 | **0.5971** |
| MOLT-4 | AUROC | 0.5506 | 0.5391 | 0.5113 | **0.5663** | 0.5585 | 0.5578 | **0.5792** | 0.5637 | 0.5547 | **0.5771** | 0.5477 | 0.5308 | **0.6067** |
| | AUPRC | 0.2203 | 0.1914 | 0.1796 | **0.2356** | 0.2177 | 0.2159 | **0.2511** | 0.2411 | 0.2176 | **0.2559** | 0.1994 | 0.1939 | **0.3104** |
| | F1-score | 0.5595 | 0.5392 | 0.5048 | **0.5658** | 0.5540 | 0.5519 | **0.5733** | 0.5306 | 0.5368 | **0.5416** | **0.5446** | 0.5115 | 0.5366 |
| PC-3 | AUROC | 0.5688 | 0.5805 | 0.5284 | **0.5821** | 0.6207 | 0.5938 | **0.6221** | 0.5705 | 0.5646 | **0.5782** | 0.5490 | 0.5442 | **0.6107** |
| | AUPRC | 0.2112 | 0.2233 | 0.1854 | **0.2385** | 0.2770 | 0.2386 | **0.2790** | 0.3506 | 0.3456 | **0.3587** | 0.2028 | 0.2108 | **0.2623** |
| | F1-score | 0.5698 | 0.5685 | 0.5321 | **0.5848** | 0.5650 | 0.5569 | **0.5686** | 0.4085 | 0.4058 | **0.4101** | 0.5027 | 0.4914 | **0.5633** |
| SW-620 | AUROC | 0.5577 | 0.5776 | 0.5232 | **0.5834** | 0.5822 | 0.5799 | **0.5936** | 0.5839 | 0.5722 | **0.5987** | 0.5265 | 0.5395 | **0.6183** |
| | AUPRC | 0.2026 | 0.2205 | 0.1958 | **0.2279** | 0.2315 | 0.2258 | **0.2455** | 0.2599 | 0.2511 | **0.2728** | 0.1340 | 0.1777 | **0.2759** |
| | F1-score | 0.5641 | 0.5477 | 0.5273 | **0.5676** | 0.5422 | 0.5473 | **0.5786** | 0.5111 | 0.5015 | **0.5220** | 0.5283 | 0.5113 | **0.5729** |
| NCI-H23 | AUROC | 0.5792 | 0.5634 | 0.5323 | **0.5979** | 0.5773 | 0.5762 | **0.6194** | 0.5912 | 0.5647 | **0.6014** | 0.5658 | 0.5760 | **0.6422** |
| | AUPRC | 0.2273 | 0.1983 | 0.2017 | **0.2407** | 0.2082 | 0.2188 | **0.2680** | 0.2587 | 0.2139 | **0.2672** | 0.1986 | 0.2345 | **0.3105** |
| | F1-score | 0.5821 | 0.5625 | 0.5445 | **0.5835** | 0.5659 | 0.5779 | **0.5914** | 0.5058 | 0.5087 | **0.5134** | **0.5561** | 0.5066 | 0.5358 |
| OVCAR-8 | AUROC | 0.5507 | 0.5494 | 0.5278 | **0.5726** | 0.5911 | 0.5859 | **0.6049** | 0.5786 | 0.5725 | **0.6024** | 0.5696 | 0.5713 | **0.6317** |
| | AUPRC | 0.1775 | 0.1633 | 0.1665 | **0.2132** | 0.2346 | 0.2196 | **0.2447** | 0.2764 | 0.2994 | **0.3072** | 0.2696 | 0.2401 | **0.3060** |
| | F1-score | 0.5461 | 0.5408 | 0.5337 | **0.5749** | 0.5350 | 0.5548 | **0.5728** | 0.4733 | 0.4469 | **0.4766** | 0.4831 | 0.4950 | **0.5211** |

Table 17: Standard Deviation of AUROC, AUPRC, and F1-score for Tables 1 and 14.

| Datasets | Metrics | GCN | +FA | SAGE | +FA | GAT | +FA | GIN | +FA | LRGNN | +FA | GRDL | +FA |
|---|---|---|---|---|---|---|---|---|---|---|---|---|---|
| MCF-7 | AUROC | 0.0027 | 0.0016 | 0.0011 | 0.0123 | 0.0144 | 0.0011 | 0.0015 | 0.0010 | 0.0011 | 0.0069 | 0.0128 | 0.0050 |
| | AUPRC | 0.0110 | 0.0016 | 0.0024 | 0.0050 | 0.0436 | 0.0465 | 0.0036 | 0.0030 | 0.0245 | 0.0135 | 0.0256 | 0.0007 |
| | F1-score | 0.0136 | 0.0038 | 0.0006 | 0.0134 | 0.0215 | 0.0442 | 0.0012 | 0.0017 | 0.0182 | 0.0030 | 0.0041 | 0.0081 |
| MOLT-4 | AUROC | 0.0080 | 0.0015 | 0.0233 | 0.0035 | 0.0028 | 0.0101 | 0.0043 | 0.0006 | 0.0122 | 0.0055 | 0.0076 | 0.0035 |
| | AUPRC | 0.0093 | 0.0032 | 0.0391 | 0.0059 | 0.0217 | 0.0024 | 0.0126 | 0.0052 | 0.0018 | 0.0107 | 0.0076 | 0.0037 |
| | F1-score | 0.0023 | 0.0047 | 0.0116 | 0.0003 | 0.0233 | 0.0155 | 0.0047 | 0.0058 | 0.0171 | 0.0177 | 0.0045 | 0.0092 |
| PC-3 | AUROC | 0.0132 | 0.0047 | 0.0093 | 0.0018 | 0.0284 | 0.0150 | 0.0015 | 0.0042 | 0.0117 | 0.0015 | 0.0163 | 0.0045 |
| | AUPRC | 0.0067 | 0.0027 | 0.0092 | 0.0043 | 0.0235 | 0.0269 | 0.0017 | 0.0049 | 0.0465 | 0.0093 | 0.0332 | 0.0004 |
| | F1-score | 0.0109 | 0.0038 | 0.0037 | 0.0016 | 0.0124 | 0.0071 | 0.0040 | 0.0017 | 0.0258 | 0.0071 | 0.0098 | 0.0064 |
| SW-620 | AUROC | 0.0063 | 0.0010 | 0.0008 | 0.0014 | 0.0216 | 0.0112 | 0.0038 | 0.0026 | 0.0054 | 0.0153 | 0.0087 | 0.0077 |
| | AUPRC | 0.0190 | 0.0035 | 0.0078 | 0.0045 | 0.0115 | 0.0040 | 0.0027 | 0.0001 | 0.0068 | 0.0106 | 0.0064 | 0.0004 |
| | F1-score | 0.0088 | 0.0045 | 0.0082 | 0.0022 | 0.0196 | 0.0237 | 0.0086 | 0.0045 | 0.0130 | 0.0292 | 0.0068 | 0.0129 |
| NCI-H23 | AUROC | 0.0030 | 0.0033 | 0.0013 | 0.0016 | 0.0062 | 0.0020 | 0.0016 | 0.0045 | 0.0306 | 0.0085 | 0.0075 | 0.0087 |
| | AUPRC | 0.0074 | 0.0015 | 0.0031 | 0.0006 | 0.0098 | 0.0074 | 0.0014 | 0.0071 | 0.0038 | 0.0197 | 0.0061 | 0.0136 |
| | F1-score | 0.0026 | 0.0029 | 0.0043 | 0.0028 | 0.0009 | 0.0100 | 0.0037 | 0.0004 | 0.0411 | 0.0349 | 0.0048 | 0.0011 |
| OVCAR-8 | AUROC | 0.0047 | 0.0008 | 0.0026 | 0.0010 | 0.0201 | 0.0061 | 0.0018 | 0.0014 | 0.0130 | 0.0008 | 0.0013 | 0.0009 |
| | AUPRC | 0.0017 | 0.0009 | 0.0037 | 0.0045 | 0.0220 | 0.0062 | 0.0033 | 0.0016 | 0.0061 | 0.0122 | 0.0052 | 0.0018 |
| | F1-score | 0.0070 | 0.0002 | 0.0003 | 0.0030 | 0.0086 | 0.0153 | 0.0005 | 0.0006 | 0.0203 | 0.0101 | 0.0037 | 0.0036 |
| P388 | AUROC | 0.0519 | 0.0009 | 0.0107 | 0.0030 | 0.1071 | 0.0002 | 0.0261 | 0.0150 | 0.0392 | 0.0008 | 0.0063 | 0.0104 |
| | AUPRC | 0.0506 | 0.0066 | 0.0127 | 0.0047 | 0.0323 | 0.0051 | 0.0249 | 0.0089 | 0.0023 | 0.0025 | 0.0640 | 0.0011 |
| | F1-score | 0.0129 | 0.0078 | 0.0011 | 0.0005 | 0.0966 | 0.0172 | 0.0100 | 0.0104 | 0.0442 | 0.0069 | 0.0422 | 0.0126 |
| SF-295 | AUROC | 0.0086 | 0.0026 | 0.0055 | 0.0011 | 0.0059 | 0.0124 | 0.0274 | 0.0067 | 0.0235 | 0.0043 | 0.0169 | 0.0026 |
| | AUPRC | 0.0007 | 0.0059 | 0.0019 | 0.0005 | 0.0336 | 0.0060 | 0.0022 | 0.0045 | 0.0116 | 0.0011 | 0.0381 | 0.0081 |
| | F1-score | 0.0106 | 0.0083 | 0.0048 | 0.0019 | 0.0254 | 0.0203 | 0.0334 | 0.0056 | 0.0399 | 0.0078 | 0.0161 | 0.0052 |
| SN12C | AUROC | 0.0100 | 0.0011 | 0.0153 | 0.0054 | 0.0083 | 0.0025 | 0.0053 | 0.0010 | 0.0137 | 0.0083 | 0.0094 | 0.0035 |
| | AUPRC | 0.0039 | 0.0053 | 0.0325 | 0.0044 | 0.2283 | 0.0106 | 0.0099 | 0.0051 | 0.2345 | 0.0136 | 0.2344 | 0.0019 |
| | F1-score | 0.0082 | 0.0033 | 0.0093 | 0.0027 | 0.0022 | 0.0081 | 0.0020 | 0.0039 | 0.0231 | 0.0264 | 0.0153 | 0.0038 |
| UACC257 | AUROC | 0.0040 | 0.0033 | 0.0040 | 0.0172 | 0.0032 | 0.0145 | 0.0112 | 0.0043 | 0.0156 | 0.0116 | 0.0031 | 0.0069 |
| | AUPRC | 0.0035 | 0.0083 | 0.0047 | 0.0031 | 0.0325 | 0.0245 | 0.0146 | 0.0071 | 0.0049 | 0.0249 | 0.0096 | 0.0014 |
| | F1-score | 0.0013 | 0.0040 | 0.0002 | 0.0230 | 0.0335 | 0.0053 | 0.0016 | 0.0008 | 0.0228 | 0.0087 | 0.0050 | 0.0115 |
| PROTEINS_full | AUROC | 0.0057 | 0.0013 | 0.0202 | 0.0182 | 0.0411 | 0.0039 | 0.0410 | 0.0052 | 0.0331 | 0.0265 | 0.0028 | 0.0066 |
| | AUPRC | 0.0053 | 0.0001 | 0.0346 | 0.0248 | 0.0367 | 0.0060 | 0.0010 | 0.0044 | 0.0345 | 0.0218 | 0.0078 | 0.0077 |
| | F1-score | 0.0059 | 0.0017 | 0.0219 | 0.0139 | 0.0420 | 0.0060 | 0.0593 | 0.0054 | 0.0312 | 0.0275 | 0.0093 | 0.0086 |
| DBLP_v1 | AUROC | 0.0141 | 0.0007 | 0.0278 | 0.0008 | 0.0400 | 0.0053 | 0.0124 | 0.0004 | 0.0031 | 0.0004 | 0.0125 | 0.0024 |
| | AUPRC | 0.0065 | 0.0004 | 0.0277 | 0.0040 | 0.0267 | 0.0023 | 0.0185 | 0.0004 | 0.0016 | 0.0002 | 0.0034 | 0.0021 |
| | F1-score | 0.0161 | 0.0008 | 0.0337 | 0.0006 | 0.0851 | 0.0075 | 0.0081 | 0.0003 | 0.0035 | 0.0005 | 0.0143 | 0.0023 |

decision boundaries. Consequently, the tables collectively demonstrate that FracAug is not just a performance booster but also a variance reducer, offering practitioners a method that delivers stronger and more consistent results across a wide range of graph-level anomaly detection tasks.

Table 18: Standard Deviation of AUROC, AUPRC, and F1-score for Tables 2 and 15.

| Datasets | Metrics | iGAD | +FA | GmapAD | +FA | RQGNN | +FA | UniGAD | +FA |
|---|---|---|---|---|---|---|---|---|---|
| MCF-7 | AUROC | 0.0020 | 0.0022 | 0.0006 | 0.0035 | 0.0167 | 0.0136 | 0.0069 | 0.0055 |
| | AUPRC | 0.0118 | 0.0016 | 0.0135 | 0.0086 | 0.0047 | 0.0042 | 0.0100 | 0.0025 |
| | F1-score | 0.0078 | 0.0015 | 0.0138 | 0.0037 | 0.0136 | 0.0067 | 0.0165 | 0.0097 |
| MOLT-4 | AUROC | 0.0033 | 0.0020 | 0.0021 | 0.0038 | 0.0041 | 0.0132 | 0.0011 | 0.0001 |
| | AUPRC | 0.0075 | 0.0009 | 0.0168 | 0.0082 | 0.0247 | 0.0047 | 0.0226 | 0.0009 |
| | F1-score | 0.0105 | 0.0018 | 0.0100 | 0.0016 | 0.0122 | 0.0001 | 0.0139 | 0.0009 |
| PC-3 | AUROC | 0.0246 | 0.0037 | 0.0096 | 0.0021 | 0.0264 | 0.0017 | 0.0040 | 0.0009 |
| | AUPRC | 0.0459 | 0.0074 | 0.0069 | 0.0049 | 0.0017 | 0.0018 | 0.0011 | 0.0005 |
| | F1-score | 0.0006 | 0.0011 | 0.0172 | 0.0068 | 0.0182 | 0.0028 | 0.0040 | 0.0015 |
| SW-620 | AUROC | 0.0033 | 0.0008 | 0.0020 | 0.0033 | 0.0088 | 0.0266 | 0.0002 | 0.0059 |
| | AUPRC | 0.0023 | 0.0023 | 0.0100 | 0.0025 | 0.0010 | 0.0172 | 0.0052 | 0.0036 |
| | F1-score | 0.0020 | 0.0033 | 0.0075 | 0.0064 | 0.0075 | 0.0033 | 0.0011 | 0.0049 |
| NCI-H23 | AUROC | 0.0081 | 0.0101 | 0.0102 | 0.0040 | 0.0068 | 0.0205 | 0.0015 | 0.0034 |
| | AUPRC | 0.0413 | 0.0409 | 0.0221 | 0.0075 | 0.0064 | 0.0223 | 0.0022 | 0.0018 |
| | F1-score | 0.0260 | 0.0235 | 0.0089 | 0.0023 | 0.0038 | 0.0010 | 0.0001 | 0.0028 |
| OVCAR-8 | AUROC | 0.0117 | 0.0044 | 0.0016 | 0.0051 | 0.0110 | 0.0013 | 0.0017 | 0.0093 |
| | AUPRC | 0.0375 | 0.0405 | 0.0028 | 0.0016 | 0.0007 | 0.0042 | 0.0035 | 0.0153 |
| | F1-score | 0.0155 | 0.0301 | 0.0006 | 0.0070 | 0.0037 | 0.0024 | 0.0012 | 0.0028 |
| P388 | AUROC | 0.0016 | 0.0021 | 0.0015 | 0.0270 | 0.0155 | 0.0146 | 0.0104 | 0.0150 |
| | AUPRC | 0.0235 | 0.0063 | 0.0030 | 0.0140 | 0.0175 | 0.0127 | 0.0026 | 0.0093 |
| | F1-score | 0.0133 | 0.0018 | 0.0040 | 0.0186 | 0.0055 | 0.0056 | 0.0101 | 0.0115 |
| SF-295 | AUROC | 0.0022 | 0.0013 | 0.0094 | 0.0156 | 0.0004 | 0.0110 | 0.0069 | 0.0027 |
| | AUPRC | 0.0264 | 0.0202 | 0.0303 | 0.0088 | 0.0046 | 0.0181 | 0.0182 | 0.0121 |
| | F1-score | 0.0203 | 0.0158 | 0.0158 | 0.0103 | 0.0008 | 0.0094 | 0.0078 | 0.0070 |
| SN12C | AUROC | 0.0004 | 0.0007 | 0.0057 | 0.0054 | 0.0025 | 0.0112 | 0.0066 | 0.0069 |
| | AUPRC | 0.2670 | 0.0159 | 0.0001 | 0.0015 | 0.0091 | 0.0100 | 0.0056 | 0.0039 |
| | F1-score | 0.0107 | 0.0167 | 0.0065 | 0.0076 | 0.0046 | 0.0031 | 0.0040 | 0.0062 |
| UACC257 | AUROC | 0.0085 | 0.0013 | 0.0105 | 0.0001 | 0.0027 | 0.0069 | 0.0016 | 0.0081 |
| | AUPRC | 0.0143 | 0.0053 | 0.0196 | 0.0025 | 0.0014 | 0.0099 | 0.0037 | 0.0006 |
| | F1-score | 0.0008 | 0.0040 | 0.0282 | 0.0019 | 0.0028 | 0.0040 | 0.0050 | 0.0091 |
| PROTEINS_full | AUROC | 0.0206 | 0.0052 | 0.0163 | 0.0069 | 0.0287 | 0.0508 | 0.0020 | 0.0008 |
| | AUPRC | 0.0052 | 0.0061 | 0.0045 | 0.0093 | 0.0428 | 0.0617 | 0.0020 | 0.0009 |
| | F1-score | 0.0239 | 0.0054 | 0.0132 | 0.0066 | 0.0354 | 0.0419 | 0.0029 | 0.0008 |
| DBLP_v1 | AUROC | 0.0004 | 0.0003 | 0.0072 | 0.0002 | 0.0029 | 0.0028 | 0.0123 | 0.0003 |
| | AUPRC | 0.0002 | 0.0002 | 0.0032 | 0.0027 | 0.0010 | 0.0012 | 0.0042 | 0.0001 |
| | F1-score | 0.0004 | 0.0004 | 0.0069 | 0.0010 | 0.0036 | 0.0026 | 0.0151 | 0.0004 |

## L  ADDITIONAL ABLATION STUDY

Beyond the 6 datasets examined in Section 5.3, we further conduct an ablation study on the remaining 6 datasets, as reported in Table 20. The notations used follow the same definitions as those in Table 4. In this extended analysis, we again observe that FracAug consistently outperforms all four of its variants across the newly included datasets. This repeated pattern reinforces the importance of the individual components that constitute our framework and provides additional evidence that each design choice contributes meaningfully to the overall performance improvements achieved by FracAug.

## M  MVP FOR VALIDATION SET ONLY

Although we follow the standard data augmentation setting—using both the validation and test sets for pseudo-labeling—as adopted in related works such as ConsisGAD (Chen et al., 2024), we additionally evaluate a more restrictive setting in which only the validation set is used for pseudo-labeling. This complementary experiment allows us to examine the robustness of our framework under an extremely limited auxiliary data scenario.

As shown in Table 21, pseudo-labeling only the validation set leads to a reduction in performance gains compared to our original setting. This behavior indicates that having access to more data for pseudo-labeling can more fully reveal the strength of our framework. At the same time, the results also demonstrate that even when supplied with very limited auxiliary data, our method can still

Table 19: Standard Deviation of AUROC, AUPRC, and F1-score for Tables 3 and 16.

| Datasets | Metrics | MAAv | NodeSam | SubMix | +FA | GLAv | GLA | +FA | GMixupv | GMixup | +FA | FGWMixupv | FGWMixup | +FA |
|---|---|---|---|---|---|---|---|---|---|---|---|---|---|---|
| MCF-7 | AUROC | 0.0069 | 0.0062 | 0.0154 | 0.0003 | 0.0069 | 0.0208 | 0.0091 | 0.0216 | 0.0173 | 0.0015 | 0.0091 | 0.0427 | 0.0014 |
| | AUPRC | 0.0070 | 0.0094 | 0.0107 | 0.0016 | 0.0112 | 0.0320 | 0.0126 | 0.0226 | 0.0272 | 0.0018 | 0.0004 | 0.0854 | 0.0152 |
| | F1-score | 0.0169 | 0.0018 | 0.0168 | 0.0014 | 0.0003 | 0.0070 | 0.0082 | 0.0144 | 0.0016 | 0.0006 | 0.0330 | 0.0096 | 0.0055 |
| MOLT-4 | AUROC | 0.0080 | 0.0225 | 0.0059 | 0.0065 | 0.0135 | 0.0029 | 0.0004 | 0.0038 | 0.0058 | 0.0001 | 0.0057 | 0.0083 | 0.0146 |
| | AUPRC | 0.0110 | 0.0284 | 0.0313 | 0.0016 | 0.0206 | 0.0047 | 0.0013 | 0.0010 | 0.0105 | 0.0003 | 0.0096 | 0.0409 | 0.0116 |
| | F1-score | 0.0076 | 0.0174 | 0.0110 | 0.0046 | 0.0042 | 0.0030 | 0.0017 | 0.0065 | 0.0013 | 0.0001 | 0.0041 | 0.0136 | 0.0148 |
| PC-3 | AUROC | 0.0062 | 0.0181 | 0.0296 | 0.0104 | 0.0046 | 0.0195 | 0.0053 | 0.0144 | 0.0179 | 0.0009 | 0.0246 | 0.0141 | 0.0011 |
| | AUPRC | 0.0078 | 0.0201 | 0.0329 | 0.0120 | 0.0091 | 0.0277 | 0.0099 | 0.0065 | 0.0290 | 0.0174 | 0.0651 | 0.0373 | 0.0017 |
| | F1-score | 0.0033 | 0.0003 | 0.0430 | 0.0058 | 0.0111 | 0.0072 | 0.0153 | 0.0235 | 0.0053 | 0.0153 | 0.0163 | 0.0095 | 0.0001 |
| SW-620 | AUROC | 0.0034 | 0.0043 | 0.0172 | 0.0020 | 0.0061 | 0.0135 | 0.0057 | 0.0029 | 0.0038 | 0.0008 | 0.0302 | 0.0241 | 0.0069 |
| | AUPRC | 0.0001 | 0.0059 | 0.0093 | 0.0031 | 0.0151 | 0.0141 | 0.0035 | 0.0191 | 0.0136 | 0.0002 | 0.0645 | 0.0704 | 0.0091 |
| | F1-score | 0.0004 | 0.0027 | 0.0276 | 0.0019 | 0.0091 | 0.0182 | 0.0052 | 0.0142 | 0.0066 | 0.0013 | 0.0335 | 0.0169 | 0.0071 |
| NCI-H23 | AUROC | 0.0139 | 0.0330 | 0.0020 | 0.0013 | 0.0057 | 0.0077 | 0.0061 | 0.0317 | 0.0018 | 0.0002 | 0.0551 | 0.0073 | 0.0045 |
| | AUPRC | 0.0125 | 0.0355 | 0.0049 | 0.0008 | 0.0118 | 0.0076 | 0.0085 | 0.0401 | 0.0070 | 0.0069 | 0.0661 | 0.0037 | 0.0071 |
| | F1-score | 0.0059 | 0.0175 | 0.0029 | 0.0006 | 0.0100 | 0.0032 | 0.0037 | 0.0083 | 0.0036 | 0.0071 | 0.0305 | 0.0074 | 0.0205 |
| OVCAR-8 | AUROC | 0.0031 | 0.0111 | 0.0193 | 0.0036 | 0.0023 | 0.0230 | 0.0007 | 0.0169 | 0.0099 | 0.0006 | 0.0194 | 0.0144 | 0.0063 |
| | AUPRC | 0.0171 | 0.0175 | 0.0139 | 0.0031 | 0.0083 | 0.0335 | 0.0008 | 0.0361 | 0.0016 | 0.0047 | 0.0783 | 0.0284 | 0.0034 |
| | F1-score | 0.0221 | 0.0028 | 0.0255 | 0.0013 | 0.0072 | 0.0062 | 0.0004 | 0.0098 | 0.0131 | 0.0035 | 0.1100 | 0.0049 | 0.0072 |
| P388 | AUROC | 0.0003 | 0.0067 | 0.0066 | 0.0095 | 0.0715 | 0.0220 | 0.0122 | 0.0001 | 0.0033 | 0.0007 | 0.0023 | 0.0028 | 0.0071 |
| | AUPRC | 0.0346 | 0.0438 | 0.0308 | 0.0058 | 0.0660 | 0.0287 | 0.0058 | 0.0049 | 0.0207 | 0.0009 | 0.0330 | 0.0086 | 0.0228 |
| | F1-score | 0.0187 | 0.0136 | 0.0138 | 0.0045 | 0.0382 | 0.0114 | 0.0057 | 0.0081 | 0.0194 | 0.0050 | 0.0292 | 0.0334 | 0.0084 |
| SF-295 | AUROC | 0.0100 | 0.0087 | 0.0035 | 0.0038 | 0.0208 | 0.0065 | 0.0023 | 0.0013 | 0.0303 | 0.0081 | 0.0059 | 0.0018 | 0.0059 |
| | AUPRC | 0.0061 | 0.0030 | 0.0016 | 0.0002 | 0.0268 | 0.0076 | 0.0023 | 0.0046 | 0.0460 | 0.0093 | 0.0073 | 0.0069 | 0.0013 |
| | F1-score | 0.0052 | 0.0030 | 0.0052 | 0.0002 | 0.0072 | 0.0087 | 0.0015 | 0.0081 | 0.0057 | 0.0049 | 0.0028 | 0.0107 | 0.0224 |
| SN12C | AUROC | 0.0052 | 0.0506 | 0.0081 | 0.0043 | 0.0308 | 0.0187 | 0.0073 | 0.0125 | 0.0117 | 0.0105 | 0.0281 | 0.0243 | 0.0116 |
| | AUPRC | 0.0206 | 0.0733 | 0.0223 | 0.0076 | 0.0287 | 0.0268 | 0.0103 | 0.0030 | 0.0247 | 0.0147 | 0.0489 | 0.0646 | 0.0317 |
| | F1-score | 0.0124 | 0.0098 | 0.0144 | 0.0040 | 0.0067 | 0.0027 | 0.0030 | 0.0177 | 0.0241 | 0.0016 | 0.0048 | 0.0232 | 0.0221 |
| UACC257 | AUROC | 0.0335 | 0.0052 | 0.0302 | 0.0041 | 0.0132 | 0.0038 | 0.0101 | 0.0137 | 0.0013 | 0.0083 | 0.0489 | 0.0366 | 0.0115 |
| | AUPRC | 0.0295 | 0.0185 | 0.1041 | 0.0001 | 0.0200 | 0.0199 | 0.0025 | 0.0547 | 0.0088 | 0.0054 | 0.0804 | 0.0959 | 0.0195 |
| | F1-score | 0.0071 | 0.0083 | 0.0454 | 0.0051 | 0.0013 | 0.0299 | 0.0127 | 0.0334 | 0.0068 | 0.0165 | 0.0378 | 0.0324 | 0.0155 |
| PROTEINS_full | AUROC | 0.0124 | 0.0013 | 0.0957 | 0.0035 | 0.0013 | 0.0355 | 0.0048 | 0.0301 | 0.0090 | 0.0008 | 0.0194 | 0.0052 | 0.0023 |
| | AUPRC | 0.0106 | 0.0071 | 0.0088 | 0.0011 | 0.0239 | 0.0339 | 0.0033 | 0.0077 | 0.0141 | 0.0059 | 0.0909 | 0.0098 | 0.0175 |
| | F1-score | 0.0129 | 0.0041 | 0.1812 | 0.0062 | 0.0116 | 0.0346 | 0.0088 | 0.0392 | 0.0042 | 0.0006 | 0.0199 | 0.0093 | 0.0087 |
| DBLP_v1 | AUROC | 0.0464 | 0.0047 | 0.0289 | 0.0012 | 0.0043 | 0.0284 | 0.0069 | 0.0064 | 0.0052 | 0.0033 | 0.0170 | 0.0180 | 0.0061 |
| | AUPRC | 0.0030 | 0.0007 | 0.0560 | 0.0026 | 0.0027 | 0.0032 | 0.0029 | 0.0037 | 0.0031 | 0.0015 | 0.0104 | 0.0132 | 0.0075 |
| | F1-score | 0.0739 | 0.0070 | 0.0711 | 0.0032 | 0.0059 | 0.0545 | 0.0078 | 0.0068 | 0.0059 | 0.0045 | 0.0181 | 0.0177 | 0.0057 |

Table 20: Ablation study.

| Datasets | Metrics | GIN | +FA | w/o largest | w/o smallest | w/o WDML | w/o MVP |
|---|---|---|---|---|---|---|---|
| MCF-7 | AUROC | 0.5867 | 0.5976 | 0.5848 | 0.5889 | 0.5860 | 0.5835 |
| | AUPRC | 0.2830 | 0.2971 | 0.2842 | 0.2882 | 0.2813 | 0.2790 |
| | F1-score | 0.5366 | 0.5421 | 0.5317 | 0.5351 | 0.5372 | 0.5350 |
| MOLT-4 | AUROC | 0.5733 | 0.5854 | 0.5770 | 0.5760 | 0.5772 | 0.5754 |
| | AUPRC | 0.2830 | 0.3001 | 0.2974 | 0.2862 | 0.2981 | 0.2881 |
| | F1-score | 0.5072 | 0.5103 | 0.5001 | 0.5085 | 0.4998 | 0.5060 |
| PC-3 | AUROC | 0.5969 | 0.6119 | 0.5963 | 0.6018 | 0.6026 | 0.5975 |
| | AUPRC | 0.2797 | 0.2893 | 0.2797 | 0.2815 | 0.2806 | 0.2780 |
| | F1-score | 0.5063 | 0.5205 | 0.5055 | 0.5124 | 0.5145 | 0.5092 |
| SW-620 | AUROC | 0.5938 | 0.6004 | 0.5941 | 0.5938 | 0.5949 | 0.5930 |
| | AUPRC | 0.2776 | 0.2813 | 0.2737 | 0.2778 | 0.2800 | 0.2697 |
| | F1-score | 0.5090 | 0.5155 | 0.5132 | 0.5089 | 0.5082 | 0.5155 |
| NCI-H23 | AUROC | 0.5897 | 0.5968 | 0.5893 | 0.5911 | 0.5866 | 0.5925 |
| | AUPRC | 0.2566 | 0.2659 | 0.2564 | 0.2633 | 0.2614 | 0.2656 |
| | F1-score | 0.5059 | 0.5073 | 0.5054 | 0.5013 | 0.4968 | 0.5013 |
| OVCAR-8 | AUROC | 0.5935 | 0.5963 | 0.5911 | 0.5918 | 0.5911 | 0.5904 |
| | AUPRC | 0.2573 | 0.2612 | 0.2579 | 0.2565 | 0.2573 | 0.2605 |
| | F1-score | 0.5118 | 0.5123 | 0.5074 | 0.5100 | 0.5081 | 0.5038 |

extract meaningful supervisory signals and deliver performance improvements. This resilience under constrained conditions highlights the superiority of our framework, further confirming that FracAug remains effective even in extreme data augmentation settings.

# N  ORIGINAL LOSS FOR BASELINES

Table 21: Average AUROC, AUPRC, and F1-score on 3 datasets with multiple runs, where the "+FA" represents our original setting, and onlyval represents pseudo-labeling only the validation set.

| Datasets | Metrics | GAT | +FA | onlyval | GIN | +FA | onlyval | iGAD | +FA | onlyval | NSv | +FA | onlyval |
|---|---|---|---|---|---|---|---|---|---|---|---|---|---|
| UACC257 | AUROC | 0.5890 | 0.6174 | 0.6021 | 0.5877 | 0.6015 | 0.5946 | 0.5697 | 0.5748 | 0.5721 | 0.5623 | 0.5947 | 0.5731 |
| | AUPRC | 0.3493 | 0.3389 | 0.3250 | 0.2480 | 0.2598 | 0.2631 | 0.1906 | 0.1970 | 0.1956 | 0.1805 | 0.2210 | 0.1847 |
| | F1-score | 0.4031 | 0.4455 | 0.4491 | 0.4906 | 0.4983 | 0.4908 | 0.5201 | 0.5224 | 0.5214 | 0.5541 | 0.5784 | 0.5582 |
| PROTEINS_full | AUROC | 0.6157 | 0.6836 | 0.6633 | 0.5799 | 0.6174 | 0.6015 | 0.5976 | 0.6206 | 0.6125 | 0.6009 | 0.6217 | 0.6147 |
| | AUPRC | 0.6350 | 0.7005 | 0.6933 | 0.6259 | 0.6358 | 0.6261 | 0.6200 | 0.6333 | 0.6271 | 0.6183 | 0.6366 | 0.6357 |
| | F1-score | 0.6158 | 0.6859 | 0.6508 | 0.5679 | 0.6175 | 0.6024 | 0.5960 | 0.6211 | 0.6137 | 0.6015 | 0.6214 | 0.614 |
| DBLP_v1 | AUROC | 0.6119 | 0.6885 | 0.6489 | 0.6231 | 0.8044 | 0.7844 | 0.7755 | 0.7909 | 0.7901 | 0.6446 | 0.6822 | 0.6712 |
| | AUPRC | 0.7507 | 0.7796 | 0.7768 | 0.7201 | 0.8626 | 0.8446 | 0.8377 | 0.8473 | 0.8466 | 0.7689 | 0.7816 | 0.7814 |
| | F1-score | 0.5782 | 0.6868 | 0.6311 | 0.5996 | 0.8028 | 0.7837 | 0.7749 | 0.7910 | 0.7902 | 0.6252 | 0.6778 | 0.6621 |

Table 22: Average AUROC, AUPRC, and F1-score on 3 datasets with multiple runs, where the "+FA" represents our original setting, and oriloss represents the original setting of baselines.

| Datasets | Metrics | GAT | +FA | oriloss | GIN | +FA | oriloss | NodeSam | +FA | oriloss | GMixup | +FA | oriloss |
|---|---|---|---|---|---|---|---|---|---|---|---|---|---|
| UACC257 | AUROC | 0.5890 | 0.6174 | 0.5583 | 0.5877 | 0.6015 | 0.5749 | 0.5023 | 0.5947 | 0.4994 | 0.5843 | 0.6209 | 0.5280 |
| | AUPRC | 0.3493 | 0.3389 | 0.3106 | 0.2480 | 0.2598 | 0.2448 | 0.0559 | 0.2210 | 0.0206 | 0.2285 | 0.2963 | 0.1185 |
| | F1-score | 0.4031 | 0.4455 | 0.4039 | 0.4906 | 0.4983 | 0.4772 | 0.5005 | 0.5784 | 0.4892 | 0.5042 | 0.4898 | 0.5165 |
| PROTEINS_full | AUROC | 0.6157 | 0.6836 | 0.5739 | 0.5799 | 0.6174 | 0.5517 | 0.6083 | 0.6217 | 0.5546 | 0.5132 | 0.6097 | 0.4979 |
| | AUPRC | 0.6350 | 0.7005 | 0.6097 | 0.6259 | 0.6358 | 0.5824 | 0.6294 | 0.6366 | 0.5703 | 0.5057 | 0.6244 | 0.5259 |
| | F1-score | 0.6158 | 0.6859 | 0.5695 | 0.5679 | 0.6175 | 0.5503 | 0.6039 | 0.6214 | 0.5548 | 0.5025 | 0.6102 | 0.4978 |
| DBLP_v1 | AUROC | 0.6119 | 0.6885 | 0.6263 | 0.6231 | 0.8044 | 0.6151 | 0.6608 | 0.6822 | 0.5000 | 0.7885 | 0.7994 | 0.7903 |
| | AUPRC | 0.7507 | 0.7796 | 0.7168 | 0.7201 | 0.8626 | 0.7154 | 0.7868 | 0.7816 | 0.7551 | 0.8471 | 0.8563 | 0.8467 |
| | F1-score | 0.5782 | 0.6868 | 0.6199 | 0.5996 | 0.8028 | 0.6147 | 0.6408 | 0.6778 | 0.3288 | 0.7878 | 0.7985 | 0.7904 |

We conduct an additional set of experiments in which baseline models were evaluated under the original, unweighted loss functions, without applying any rebalancing techniques or class frequency adjustments. The results are summarized in Table 22. As shown, most baseline models experience a substantial decline in performance when trained with their original loss in the imbalanced setting. This degradation is consistent across metrics and datasets, confirming that the performance of these models is sensitive to skewed class distributions. Importantly, this finding validates our experimental choice in the main paper, i.e., the weighted loss was introduced precisely to prevent the baselines from collapsing under severe imbalance, thereby ensuring a fair and meaningful comparison with our proposed method. Without such weighting, the baselines fail to capture minority class patterns effectively, leading to inflated majority class predictions and weakened anomaly or minority detection capability. These newly added results therefore reinforce the significance of using balanced training strategies and further highlight the robustness advantage of our approach.

## O  RUNNING TIME WHEN VARYING HYPERPARAMETERS

Because FGG involves four hyperparameters, $k_l, k_s, H_l, H_s$, that might influence the running time, we evaluate the computational overhead under different configurations. We present two 3D plots that report the runtime under varying hyperparameters, as shown in Figures 7. The first plot sweeps $k_l$ and $k_s$ from 1 to 8 and records the training time. The second plot similarly varies $H_l$ and $H_s$ from 1 to 8 and measures the runtime during training.

Across both experiments, we observe that the runtime does increase as the hyperparameters grow, but the increase is small and progresses gradually. Larger $k_l$ and $k_s$ require computing additional eigenvalues, and larger $H_l$ and $H_s$ apply more repeated or deeper fractional operations. Despite this, the overall time remains low and exhibits no sharp growth. These results confirm that FGG's computational cost scales smoothly with its hyperparameters, and even at higher settings, it maintains an efficient and manageable runtime suitable for practical graph-level anomaly detection.

## P  PSEUDO LABEL CONSISTENCY

An important aspect of evaluating FracAug is understanding how reliably the synthetic samples align with the labels of their corresponding original graphs to keep label invariance from semantic

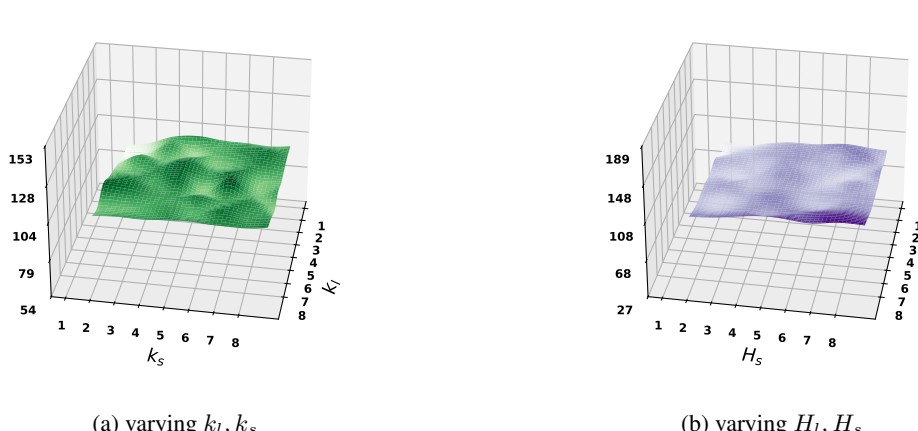

(a) varying $k_l, k_s$                      (b) varying $H_l, H_s$

Figure 7: Training time when varying $k_l, k_s, H_l, H_s$ for UACC257 based on GIN.

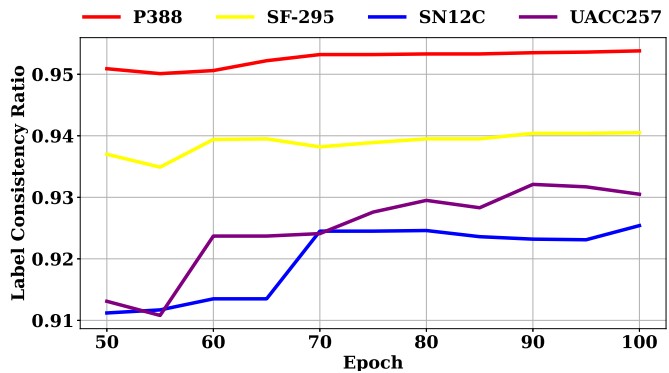

Figure 8: Label consistency ratio.

preservation. To measure this, we track label consistency ratio, defined as the proportion of synthetic samples whose pseudo labels match the predicted label of the original sample from which they are generated. This metric reflects how faithfully the synthetic data preserves the class semantics and whether the pseudo-labeling process remains stable throughout iterative training.

Our experimental results in Figure 8 reveal a clear and consistent trend: label consistency ratio is high after the warmup and continues to increase across iterations. This high label consistency ratio can be attributed to the positive feedback loop established by the MVP pseudo-labeling mechanism and semantic-preserving generation by FGG. Since only synthetic samples that match the prediction of their associated original graph are admitted into training, the model is repeatedly reinforced with high-quality, label-aligned synthetic instances. Over time, this encourages the baseline model to develop more robust feature representations. Simultaneously, the synthetic graphs generated by FGG become increasingly aligned with the semantic structure of the original samples, further boosting consistency.

Across all evaluated datasets, we observe the same pattern: label consistency ratio remains high throughout training and increases over iterations. This behavior confirms that the mutual verification pseudo-labeling strategy effectively suppresses noisy labels and enhances the reliability of synthetic augmentation. The rising label consistency ratio also indicates that FracAug drives the model toward a stable equilibrium, where both original and synthetic samples converge to consistent and semantically coherent predictions.

In summary, the empirical evidence demonstrates that FracAug maintains strong label fidelity and progressively strengthens the mutual agreement between original and synthetic samples during training. This increasing label consistency ratio is a key factor contributing to the overall robustness and performance gains enabled by FracAug.

Table 23: Ablation study by replacing FGG by Node Drop(ND) or Edge Drop (ED), and replacing MVP by Soft Ensemble (SE).

| Datasets | Metrics | GIN | +FA | ND | ED | SE | NS | +FA | ND | ED | SE | GMixup | +FA | ND | ED | SE |
|---|---|---|---|---|---|---|---|---|---|---|---|---|---|---|---|---|
| MCF-7 | AUROC | 0.5867 | 0.5976 | 0.5878 | 0.5838 | 0.5917 | 0.5496 | 0.5695 | 0.5575 | 0.5442 | 0.5594 | 0.5730 | 0.5935 | 0.5758 | 0.5720 | 0.5738 |
| | AUPRC | 0.2830 | 0.2971 | 0.2834 | 0.2832 | 0.2959 | 0.2346 | 0.2455 | 0.2224 | 0.2054 | 0.2248 | 0.2767 | 0.3081 | 0.2941 | 0.2947 | 0.2762 |
| | F1-score | 0.5366 | 0.5421 | 0.5384 | 0.5309 | 0.5317 | 0.5179 | 0.5721 | 0.5460 | 0.5506 | 0.5589 | 0.5186 | 0.5220 | 0.5070 | 0.5009 | 0.5203 |
| MOLT-4 | AUROC | 0.5733 | 0.5854 | 0.5790 | 0.5774 | 0.5789 | 0.5506 | 0.5663 | 0.5496 | 0.5521 | 0.5557 | 0.5637 | 0.5771 | 0.5737 | 0.5707 | 0.5714 |
| | AUPRC | 0.2830 | 0.3001 | 0.2967 | 0.2929 | 0.2949 | 0.2203 | 0.2356 | 0.2271 | 0.2261 | 0.2240 | 0.2411 | 0.2559 | 0.2414 | 0.2545 | 0.2459 |
| | F1-score | 0.5072 | 0.5103 | 0.5024 | 0.5047 | 0.5101 | 0.5595 | 0.5658 | 0.5606 | 0.5622 | 0.5632 | 0.5306 | 0.5416 | 0.5378 | 0.5300 | 0.5414 |
| PC-3 | AUROC | 0.5969 | 0.6119 | 0.6036 | 0.5991 | 0.6015 | 0.5688 | 0.5821 | 0.5664 | 0.5639 | 0.5679 | 0.5705 | 0.5782 | 0.5677 | 0.5700 | 0.5731 |
| | AUPRC | 0.2797 | 0.2893 | 0.2855 | 0.2837 | 0.2778 | 0.2112 | 0.2385 | 0.2132 | 0.2103 | 0.2149 | 0.3506 | 0.3587 | 0.3473 | 0.3527 | 0.3546 |
| | F1-score | 0.5063 | 0.5205 | 0.5110 | 0.5058 | 0.5156 | 0.5698 | 0.5848 | 0.5716 | 0.5700 | 0.5725 | 0.4085 | 0.4101 | 0.4083 | 0.4038 | 0.4096 |
| SW-620 | AUROC | 0.5938 | 0.6004 | 0.5919 | 0.5958 | 0.5932 | 0.5577 | 0.5834 | 0.5634 | 0.5610 | 0.5581 | 0.5839 | 0.5987 | 0.5855 | 0.5834 | 0.5872 |
| | AUPRC | 0.2776 | 0.2813 | 0.2721 | 0.2793 | 0.2746 | 0.2026 | 0.2279 | 0.2072 | 0.2041 | 0.1991 | 0.2599 | 0.2728 | 0.2574 | 0.2609 | 0.2658 |
| | F1-score | 0.5090 | 0.5155 | 0.5112 | 0.5101 | 0.5109 | 0.5641 | 0.5676 | 0.5666 | 0.5651 | 0.5625 | 0.5111 | 0.5220 | 0.5169 | 0.5090 | 0.5110 |
| NCI-H23 | AUROC | 0.5897 | 0.5968 | 0.5847 | 0.5850 | 0.5900 | 0.5792 | 0.5979 | 0.5729 | 0.5780 | 0.5909 | 0.5912 | 0.6014 | 0.5911 | 0.5947 | 0.5984 |
| | AUPRC | 0.2566 | 0.2659 | 0.2543 | 0.2522 | 0.2626 | 0.2273 | 0.2407 | 0.2255 | 0.2258 | 0.2295 | 0.2587 | 0.2672 | 0.2522 | 0.2665 | 0.2610 |
| | F1-score | 0.5059 | 0.5073 | 0.5007 | 0.5031 | 0.5004 | 0.5821 | 0.5835 | 0.5816 | 0.5821 | 0.5776 | 0.5058 | 0.5134 | 0.5127 | 0.5036 | 0.5121 |
| OVCAR-8 | AUROC | 0.5935 | 0.5963 | 0.5926 | 0.5918 | 0.5933 | 0.5507 | 0.5726 | 0.5603 | 0.5561 | 0.5549 | 0.5786 | 0.6024 | 0.5846 | 0.5840 | 0.5827 |
| | AUPRC | 0.2573 | 0.2612 | 0.2527 | 0.2584 | 0.2604 | 0.1775 | 0.2132 | 0.1904 | 0.1758 | 0.1740 | 0.2764 | 0.3072 | 0.2916 | 0.2851 | 0.2883 |
| | F1-score | 0.5118 | 0.5123 | 0.5068 | 0.5081 | 0.5082 | 0.5461 | 0.5749 | 0.5628 | 0.5518 | 0.5359 | 0.4733 | 0.4766 | 0.4680 | 0.4729 | 0.4740 |
| P388 | AUROC | 0.5565 | 0.5913 | 0.5662 | 0.5749 | 0.5645 | 0.5500 | 0.5720 | 0.5507 | 0.5700 | 0.5507 | 0.5469 | 0.5647 | 0.5467 | 0.5486 | 0.5484 |
| | AUPRC | 0.2850 | 0.3309 | 0.2933 | 0.3026 | 0.2948 | 0.1958 | 0.2229 | 0.1919 | 0.2077 | 0.2015 | 0.1694 | 0.1957 | 0.1821 | 0.1795 | 0.1789 |
| | F1-score | 0.4468 | 0.4491 | 0.4445 | 0.4439 | 0.4480 | 0.5520 | 0.5746 | 0.5599 | 0.5678 | 0.5625 | 0.5315 | 0.5373 | 0.5313 | 0.5222 | 0.5224 |
| SF-295 | AUROC | 0.5844 | 0.6076 | 0.5948 | 0.5972 | 0.5998 | 0.5649 | 0.5753 | 0.5636 | 0.5721 | 0.5624 | 0.5665 | 0.6040 | 0.5862 | 0.5841 | 0.5694 |
| | AUPRC | 0.2766 | 0.2832 | 0.2673 | 0.2796 | 0.2780 | 0.2114 | 0.2252 | 0.2154 | 0.2072 | 0.2149 | 0.2004 | 0.2509 | 0.2374 | 0.2278 | 0.2090 |
| | F1-score | 0.4803 | 0.5047 | 0.5016 | 0.4932 | 0.4984 | 0.5736 | 0.5820 | 0.5752 | 0.5717 | 0.5745 | 0.5245 | 0.5371 | 0.5186 | 0.5260 | 0.5198 |
| SN12C | AUROC | 0.5995 | 0.6079 | 0.5972 | 0.5959 | 0.6017 | 0.5509 | 0.5795 | 0.5561 | 0.5538 | 0.5690 | 0.5713 | 0.5984 | 0.5882 | 0.5740 | 0.5817 |
| | AUPRC | 0.2696 | 0.2746 | 0.2677 | 0.2682 | 0.2695 | 0.2164 | 0.2164 | 0.1710 | 0.1704 | 0.1855 | 0.2163 | 0.2524 | 0.2386 | 0.2150 | 0.2328 |
| | F1-score | 0.5030 | 0.5110 | 0.5017 | 0.4991 | 0.5065 | 0.5538 | 0.5770 | 0.5490 | 0.5523 | 0.5650 | 0.5136 | 0.5195 | 0.5175 | 0.5185 | 0.5127 |
| UACC257 | AUROC | 0.5877 | 0.6015 | 0.5905 | 0.5859 | 0.5952 | 0.5623 | 0.5947 | 0.5768 | 0.5679 | 0.5689 | 0.5853 | 0.6209 | 0.5989 | 0.5945 | 0.5951 |
| | AUPRC | 0.2480 | 0.2598 | 0.2442 | 0.2553 | 0.2573 | 0.1805 | 0.2210 | 0.1932 | 0.1813 | 0.2112 | 0.2365 | 0.2963 | 0.2405 | 0.2548 | 0.2727 |
| | F1-score | 0.4906 | 0.4983 | 0.4978 | 0.4818 | 0.4919 | 0.5541 | 0.5784 | 0.5657 | 0.5615 | 0.5794 | 0.4993 | 0.4898 | 0.4883 | 0.4933 | 0.4780 |
| PROTEINS_full | AUROC | 0.5799 | 0.6174 | 0.5808 | 0.5815 | 0.5979 | 0.6009 | 0.6217 | 0.6054 | 0.6039 | 0.6157 | 0.5411 | 0.6097 | 0.5759 | 0.5523 | 0.5554 |
| | AUPRC | 0.6259 | 0.6358 | 0.6139 | 0.6075 | 0.6297 | 0.6183 | 0.6366 | 0.6271 | 0.6353 | 0.6304 | 0.5810 | 0.6244 | 0.5997 | 0.5770 | 0.5992 |
| | F1-score | 0.5679 | 0.6175 | 0.5770 | 0.5801 | 0.5933 | 0.6137 | 0.6214 | 0.6003 | 0.5991 | 0.6137 | 0.5348 | 0.6102 | 0.5753 | 0.5520 | 0.5494 |
| DBLP_v1 | AUROC | 0.6231 | 0.8044 | 0.7034 | 0.6791 | 0.7849 | 0.6446 | 0.6822 | 0.6601 | 0.6414 | 0.6712 | 0.7939 | 0.7994 | 0.7891 | 0.7803 | 0.7812 |
| | AUPRC | 0.7201 | 0.8626 | 0.8003 | 0.7738 | 0.8411 | 0.7689 | 0.7816 | 0.7794 | 0.7329 | 0.7614 | 0.8503 | 0.8563 | 0.8479 | 0.8394 | 0.8403 |
| | F1-score | 0.5996 | 0.8028 | 0.7009 | 0.6775 | 0.7847 | 0.6252 | 0.6778 | 0.6467 | 0.6399 | 0.6712 | 0.7937 | 0.7985 | 0.7885 | 0.7805 | 0.7812 |

## Q  REPLACEMENT FOR FGG/MVP

To further evaluate the contribution of FracAug's main components, we conduct an additional ablation study by replacing FGG. We substitute FGG with two standard perturbation-based augmentations, Node Drop and Edge Drop. As shown in Table 23, these replacements often result in degraded performance compared to our original setting, and sometimes even lower than their baselines. Although our MVP can rule out the noise from non-semantic-preserving generation by mutual verification to some extent, the fail to leverage guaranteed semantic-preserving properties prevents existing data augmentation methods from achieving optimal results, which further demonstrates the effectiveness of our proposed components.

We further replace MVP with a soft ensemble pseudo-labeling strategy that averages the predictions of the original and synthetic samples. This relaxation produces noisier and less reliable pseudo-labels, since it lacks the strict consistency constraint enforced by mutual verification. Consequently, as shown in Table 23, performance becomes unstable or deteriorates across datasets. In contrast, the full FracAug framework, using FGG for high-fidelity synthetic generation and MVP for robust pseudo-label filtering, consistently delivers superior results, underscoring the critical importance of both design components.

Instead of conducting experiments on GIN, we also conduct the same experiments on different backbones to prove that the effectiveness of our framework is a general advantage.

In conclusion, the additional ablation study, where FGG is replaced with either Node Drop or Edge Drop, and MVP is replaced with Soft Ensemble, further reinforces the effectiveness of each individual component. These controlled substitutions consistently lead to inferior performance, thereby validating that all the components contribute uniquely and substantially to the overall design.

## R  INFLUENCE OF MIDDLE EIGENVALUES

Table 24: Ablation study for middle eigenvalues.

| Datasets | Metrics | GIN | +FA | w/ middle |
|---|---|---|---|---|
| MCF-7 | AUROC | 0.5867 | 0.5976 | 0.5958 |
| | AUPRC | 0.2830 | 0.2971 | 0.2913 |
| | F1-score | 0.5366 | 0.5421 | 0.5452 |
| MOLT-4 | AUROC | 0.5733 | 0.5854 | 0.5851 |
| | AUPRC | 0.2830 | 0.3001 | 0.3034 |
| | F1-score | 0.5072 | 0.5103 | 0.5066 |
| PC-3 | AUROC | 0.5969 | 0.6119 | 0.6182 |
| | AUPRC | 0.2797 | 0.2893 | 0.2898 |
| | F1-score | 0.5063 | 0.5205 | 0.5283 |
| SW-620 | AUROC | 0.5938 | 0.6004 | 0.5980 |
| | AUPRC | 0.2776 | 0.2813 | 0.2820 |
| | F1-score | 0.5090 | 0.5155 | 0.5110 |
| NCI-H23 | AUROC | 0.5897 | 0.5968 | 0.5923 |
| | AUPRC | 0.2566 | 0.2659 | 0.2614 |
| | F1-score | 0.5059 | 0.5073 | 0.5063 |
| OVCAR-8 | AUROC | 0.5935 | 0.5963 | 0.6026 |
| | AUPRC | 0.2573 | 0.2612 | 0.2617 |
| | F1-score | 0.5118 | 0.5123 | 0.5119 |
| P388 | AUROC | 0.5565 | 0.5913 | 0.6031 |
| | AUPRC | 0.2850 | 0.3309 | 0.3376 |
| | F1-score | 0.4468 | 0.4491 | 0.4483 |
| SF-295 | AUROC | 0.5844 | 0.6076 | 0.6089 |
| | AUPRC | 0.2766 | 0.2832 | 0.2873 |
| | F1-score | 0.4803 | 0.5047 | 0.4983 |
| SN12C | AUROC | 0.5995 | 0.6079 | 0.6082 |
| | AUPRC | 0.2696 | 0.2746 | 0.2756 |
| | F1-score | 0.5030 | 0.5110 | 0.5103 |
| UACC257 | AUROC | 0.5877 | 0.6015 | 0.5980 |
| | AUPRC | 0.2480 | 0.2598 | 0.2525 |
| | F1-score | 0.4906 | 0.4983 | 0.5006 |
| PROTEINS_full | AUROC | 0.5799 | 0.6174 | 0.6105 |
| | AUPRC | 0.6259 | 0.6358 | 0.6321 |
| | F1-score | 0.5679 | 0.6175 | 0.6096 |
| DBLP_v1 | AUROC | 0.6231 | 0.8044 | 0.8028 |
| | AUPRC | 0.7201 | 0.8626 | 0.8607 |
| | F1-score | 0.5996 | 0.8028 | 0.8015 |

Originally, FGG is intentionally constructed around the largest top-$k_l$ and smallest top-$k_s$ eigenvalues, as these parts of the spectrum encode global structural patterns and localized irregularities, both of which are crucial for effectively distinguishing anomalous graphs. To further examine this design choice, we conduct an analysis to determine whether incorporating the middle region of the eigenvalue spectrum offers any meaningful contribution to our fractional augmentation process.

As shown in Table 24, our results indicate that including all eigenvalues does not lead to any noticeable performance improvement across the evaluated datasets. Moreover, extracting these intermediate spectral components requires performing a full eigendecomposition, which incurs a computational complexity of $O(n^3)$, where $n$ denotes the number of nodes in the graph. This stands in sharp contrast to the top-$k$ approximate eigensolvers used in our original design, whose complexity is only $O(k \cdot nnz \cdot t)$, with $nnz$ representing the number of non-zero entries in the adjacency matrix and $t$ denoting the iteration count of the eigendecomposition. Given the minimal performance gains and the substantially higher computational burden, these findings confirm that focusing solely on the largest top-$k_l$ and smallest top-$k_s$ eigenvalues is both an effective and efficient choice for FGG's augmentation strategy.

## S    COMPARISON WITH WARMUP MODELS

Table 25: Average AUROC, AUPRC, and F1-score on 3 datasets with multiple runs, where the "+FA" represents our original setting, and warmup represents the baseline mode after the warmup phase.

| Datasets | Metrics | GAT | +FA | warmup | GIN | +FA | warmup | iGAD | +FA | warmup | NSv | +FA | warmup |
|---|---|---|---|---|---|---|---|---|---|---|---|---|---|
| | AUROC | 0.5890 | 0.6174 | 0.5568 | 0.5877 | 0.6015 | 0.5606 | 0.5697 | 0.5748 | 0.5590 | 0.5623 | 0.5947 | 0.5499 |
| UACC257 | AUPRC | 0.3493 | 0.3389 | 0.3139 | 0.2480 | 0.2598 | 0.1994 | 0.1906 | 0.1970 | 0.1780 | 0.1805 | 0.2210 | 0.1515 |
| | F1-score | 0.4031 | 0.4455 | 0.3993 | 0.4906 | 0.4983 | 0.4972 | 0.5201 | 0.5224 | 0.5145 | 0.5541 | 0.5784 | 0.5280 |
| | AUROC | 0.6157 | 0.6836 | 0.6012 | 0.5799 | 0.6174 | 0.5723 | 0.5976 | 0.6206 | 0.5631 | 0.6009 | 0.6217 | 0.5875 |
| PROTEINS_full | AUPRC | 0.6350 | 0.7005 | 0.6112 | 0.6259 | 0.6358 | 0.5987 | 0.6200 | 0.6333 | 0.6234 | 0.6183 | 0.6366 | 0.6287 |
| | F1-score | 0.6158 | 0.6859 | 0.6014 | 0.5679 | 0.6175 | 0.5713 | 0.5960 | 0.6211 | 0.5669 | 0.6015 | 0.6214 | 0.5619 |
| | AUROC | 0.6119 | 0.6885 | 0.5436 | 0.6231 | 0.8044 | 0.6098 | 0.7755 | 0.7909 | 0.7482 | 0.6446 | 0.6822 | 0.6414 |
| DBLP_v1 | AUPRC | 0.7507 | 0.7796 | 0.7207 | 0.7201 | 0.8626 | 0.7014 | 0.8377 | 0.8473 | 0.8272 | 0.7689 | 0.7816 | 0.7329 |
| | F1-score | 0.5782 | 0.6868 | 0.5425 | 0.5996 | 0.8028 | 0.5910 | 0.7749 | 0.7910 | 0.7447 | 0.6252 | 0.6778 | 0.6239 |

Table 26: Comparison with unsupervised and finetuned models. "+FA" represents FracAug-enhanced GIN. "OOM" represents the out-of-memory issue.

| Datasets | Metrics | Unsupervised | | | | | | Mole-BERT | | ours | |
|---|---|---|---|---|---|---|---|---|---|---|---|
| | | OCGIN | GLADC | GLocaKD | OCGTL | SIGNET | CVTGAD | pretrain | finetune | GIN | +FA |
| | AUROC | 0.3435 | 0.3206 | 0.5596 | 0.5971 | 0.5841 | 0.2485 | 0.4414 | 0.5124 | 0.5799 | 0.6174 |
| PROTEINS_full | AUPRC | 0.3132 | 0.3032 | 0.4941 | 0.4302 | 0.4962 | 0.2815 | 0.5968 | 0.5988 | 0.6259 | 0.6358 |
| | F1-score | 0.2880 | 0.2879 | 0.3733 | 0.2879 | 0.2879 | 0.2879 | 0.3697 | 0.4882 | 0.5679 | 0.6175 |
| | AUROC | 0.4536 | 0.5340 | 0.4370 | 0.5425 | 0.5131 | OOM | 0.5077 | 0.6677 | 0.6231 | 0.8044 |
| DBLP_v1 | AUPRC | 0.4651 | 0.5470 | 0.4522 | 0.5552 | 0.5332 | OOM | 0.6506 | 0.7305 | 0.7201 | 0.8626 |
| | F1-score | 0.3290 | 0.3288 | 0.3288 | 0.3378 | 0.3378 | OOM | 0.5069 | 0.6678 | 0.5996 | 0.8028 |

We conduct an additional experiment to further demonstrate the effectiveness of our proposed model. As described in Section 4, before integrating FracAug into the baseline models, we ensure that the baselines first acquire a foundational understanding of the graph-level anomaly detection dataset by warmup. This step is crucial to stabilize their predictions and enhance the reliability of pseudo-labeling.

As shown in Table 25, the baseline models that undergo only a brief warmup phase, without sufficient training epochs, are outperformed by their fully trained counterparts. However, once FracAug is integrated into the training process of warmup baselines and then trained for an equivalent number of epochs as the fully-trained baselines, it becomes evident that the warmup models with FracAug achieve substantial performance improvements over the baselines. Ultimately, these models reach state-of-the-art performance, providing strong evidence of the effectiveness of our proposed FracAug framework.

# T    COMPARISON WITH UNSUERVISED AND FINETUNED MODELS

We first conduct comparisons between unsupervised graph-level anomaly detection models from a novel benchmark paper (Wang et al., 2025) and the FracAug-enhanced GIN. The results in Table 26 show that unsupervised training falls significantly behind our method, even though we only utilize 1% examples as the training set, indicating that label information plays a crucial role in achieving strong anomaly detection performance. Consistent with our earlier experiments in Appendix H, increasing the amount of available labeled data leads to further improvements, reinforcing the importance of label supervision. These findings highlight the importance and effectiveness of our proposed plug-in data augmentation framework, since in real deployment, anomaly detection works usually suffer from the limited label supervision issue.

Furthermore, we compare our performance with Mole-BERT (Xia et al., 2023), which is a model pretrained in an unsupervised manner and subsequently finetuned with labeled data for a specific domain. As shown in Table 26, the unsupervised pretraining alone produces suboptimal results, consistent with the behavior of the unsupervised baselines. Even after finetuning with the same labeled data budget as our framework, the performance remains inferior to ours to a large extent. This observation suggests two key points: (1) finetuning may inherit a suboptimal parameter space induced by unsupervised pretraining, limiting its adaptability, and (2) sufficient and properly utilized label information is essential for achieving robust performance. Together, these findings further validate

the novelty and effectiveness of our plug-in data augmentation framework, which can effectively generate more fidelity label information, even under extremely limited supervision.

