# OpenReview forum: "FracAug: Fractional Augmentation boost Graph-level Anomaly Detection under Limited Supervision"
_ICLR.cc/2026/Conference — Submitted to ICLR 2026_

### Official Review · Reviewer_rSBY · 2025-10-25

**Soundness:** 2
**Presentation:** 2
**Contribution:** 2
**Rating:** 2
**Confidence:** 4

**Summary:**

This paper proposes FracAug, a plug-in augmentation framework for graph-level anomaly detection (GAD) under limited supervision. FracAug has three parts: (i) a Fractional Graph Generator (FGG) that forms augmented graphs via fractional powers of a transformed adjacency (spectral) operator; (ii) a Weighted Distance-aware Margin Loss (WDML) to handle class imbalance with reweight and keep synthetic samples near their originals; and (iii) a Mutual Verification Pseudo-labeler (MVP) that accepts pseudo-labels only when predictions on original and augmented graphs agree. Experiments cover 12 datasets and multiple GNN backbones.

**Strengths:**

- The mutual-verification pseudo-labeler is a sensible way to reduce pseudo-label noise in low-label regimes; the formalization of agreement criteria is clear.

- This paper uses simple thorem to help readers better understand the motivation of the method, which is good.

**Weaknesses:**

- Scalability and resources. The graph generation relies on eigenvalue decomposition of the adjacency matrix, which is computationally expensive. Please report how time and memory grow with graph size, and include wall-clock time and peak memory for representative datasets.

- Spectral choices. During graph generation, this work retains a small set of the smallest eigenvalues and drop the middle of the spectrum. Why are the smallest ones kept, given that they often align with noise, and why is the middle removed? Please provide either a clear rationale or an ablation that compares different keep/drop bands.

- Organization of experiments. Important analyses (such as ablations and hyperparameter studies) are in the appendix, while three long tables are put in the main text. Please shorten the main tables and move core analyses into the main text so readers do not need to hunt for critical evidence in the Appendix.

- Variance and significance. Tables 1–3 omit standard deviations and the number of runs; the phrase 'multiple runs' is not informative. In a limited-supervision setting—especially when mean gaps are small in the result—significance testing is necessary to establish that the observed gains are statistically meaningful rather than noise. Please report mean and standard deviation with the number of runs, and include significance tests against the baselines. Also clarify what “+FA” denotes in Table 3.

**Questions:**

See Weakness.

---

> ### Author Response · Authors · 2025-11-20
> **Response to Reviewer rSBY (1/1)**
>
> We appreciate your comprehensive and constructive review. Your crucial comments on experiments are exceedingly helpful for us to improve our manuscript. Our point-to-point responses to your comments are given below.
>
> ---
>
> **W1**: Please note that the eigenvalue decomposition of the adjacency matrix can be preprocessed without causing additional computational cost during training. For each dataset, we only need to conduct the eigenvalue decomposition once for different backbones. And we have already provided time analysis and reported the preprocess+training time of our FracAug in Appendix D of our original manuscript. To further address the concern of time usage and memory usage, we additionally provide wall-clock time usage, the total memory usage, and peak memory usage in Appendix D and E in our revised manuscript. Besides, as required, we also provide the corresponding time cost when varying $k_l, k_s, H_l, H_s$ in Appendix O in our revised manuscript. As shown in the comprehensive cost comparison experiments, we observe that our FracAug achieves a relatively low cost in terms of both memory and time, while boosting the performance of the baselines, demonstrating the superiority of our framework.
>
> ---
>
> **W2**: We will provide both perspectives to address the reviewer’s concern.
>
> - Rationale: several works, such as [1-3], present that using high-pass and low-pass filters to preserve large and small eigenvalues will provide enough information for the spectral domain of graphs, where large eigenvalues will preserve effective information from heterophilic graphs, and small eigenvalues will retain effective information from homophilic graphs. Thus, the combination of the large and small eigenvalues technique can preserve enough key information when applied to different graph-level anomaly detection datasets. Furthermore, related works, such as [4-5], also consider the balance of effectiveness and cost. We can utilize the fast implementation of eigendecomposition to get the top-k largest and smallest eigenvalues with $O(k·nnz·t)$, where $nnz$ is the non-zero entry in the matrix, and $t$ is the iteration of the decomposition, whereas we have to do full eigendecomposition with $O(n^3)$, where $n$ is the number of nodes in the graph, to obtain the middle eigenvalues, which can be a huge burden even with preprocessing.
>
> - Ablation: To further address the reviewer’s concern, we utilize a full eigendecomposition to get all the eigenvalues to show the performance of our framework with all the eigenvalues. As shown in Appendix R in our revised manuscript, although in some cases, including additional information (middle eigenvalues) will increase the final performance to a small extent, in most cases, the gain can be negligible, which indicates that including the top-k largest and smallest eigenvalues can already preserve the needed semantics contained in the graph.
>
> [1] Jingyu Chen, Runlin Lei, Zhewei Wei. PolyGCL: GRAPH CONTRASTIVE LEARNING via Learnable Spectral Polynomial Filters. ICLR 2024.
>
> [2] Guancheng Wan, Yijun Tian, Wenke Huang, Nitesh V Chawla, Mang Ye. S3GCL: Spectral, Swift, Spatial Graph Contrastive Learning. ICML 2024.
>
> [3] Ziyun Zou, Yinghui Jiang, Lian Shen, Juan Liu, Xiangrong Liu. LOHA: Direct Graph Spectral Contrastive Learning Between Low-Pass and High-Pass Views. AAAI 2025.
>
> [4] Yuchen Yan, Yuzhong Chen, Huiyuan Chen, Minghua Xu, Mahashweta Das, Hao Yang, Hanghang Tong. From Trainable Negative Depth to Edge Heterophily in Graphs. NeurIPS 2023.
>
> [5] Richard B Lehoucq, Danny C Sorensen, and Chao Yang. ARPACK users’ guide: solution of large-scale eigenvalue problems with implicitly restarted Arnoldi methods. SIAM, 1998.
>
> ---
>
>
> **W3**: We thank the reviewer for providing valuable suggestions in terms of the paper writing, and we will follow the instructions to revise our main manuscript.
>
> ---
>
> **W4**: To address the reviewer’s concern, we present the corresponding standard deviation (5 runs) of Tables 1, 2, 3, 14, 15, and 16 in Appendix K of our revised manuscript. As shown in the results, our results indicate that the observed gains are statistically meaningful rather than noise. Besides, the results also suggest that our FracAug can reduce the variance of the backbones. As for the meaning of “+FA” in Table 3, we have already explained it in the title of the table by illustrating that the gray column is the combination of the backbone and our proposed FracAug. However, we also thank the reviewer for providing valuable suggestions, and we will revise our manuscript based on the instructions.
>
> ---
>
> We sincerely appreciate your time, and we are glad to answer any additional questions you may have.

---

### Official Review · Reviewer_iYum · 2025-10-29

**Soundness:** 3
**Presentation:** 2
**Contribution:** 2
**Rating:** 4
**Confidence:** 4

**Summary:**

This paper focus on the issues of class imbalance and lack of lables in graph-level anomaly detection. To handle the problems, the paper  proposes to augment graphs using fractional graphs that preserve semantics, a dynamic marginal loss, and a verification strategy to pseudo-label graphs. Extensive experiments demonstrate the effectiveness of the proposed method.

**Strengths:**

1. Lable scarcity and class imbalance are crucial problems in graph anomaly detection.
2. The proposed method is model-agonstic, improving performance of SOTA detection methods.
3. The motivation of each technique design is clear.

**Weaknesses:**

1. The novelty of each component is limited, only a few improvments on existing frameworks.
2. The effectivenss of each component is doubtable, in Table 8, w/o WDML w/o MVP sometimes fall behind GIN.
3. The experiment setting (1% labeled training, 98% unlabeled for testing) brings advantage to pseudo-labeling methods, I think other unsupervised GAD method should be compared [1,2].

[1] GOOD-D: On Unsupervised Graph Out-Of-Distribution Detection
[2] Towards self-interpretable graph-level anomaly detection

**Questions:**

1. The exp of Table 3 is not convincing, I'd like to see the results of other augmentations + WDML+MVP, to demonstrate the superiority of FGG.
2. What is performance of other pseudo-labeling methods like ConsisGAD?

---

> ### Author Response · Authors · 2025-11-20
> **Response to Reviewer iYum (1/2)**
>
> We appreciate your comprehensive and constructive review. Your crucial comments on experiments are exceedingly helpful for us to improve our manuscript. Our point-to-point responses to your comments are given below.
>
> ---
>
> **W1**: We respectively disagree with the statement that the novelty of our component is limited. Our method represents a fundamentally novel approach in graph augmentation, specifically designed for graph-level anomaly detection, rather than an incremental improvement. To be specific, three main aspects distinguish FracAug from previous work:
>
> - Distinct Design: We present the first data augmentation approach specifically designed for graph-level anomaly detection, simultaneously tackling the limited supervision and distribution imbalance problems. Under such an extreme scenario, previous augmentation methods that neglect the semantics of samples often generate suboptimal or counterproductive samples, as shown in our experiments in Section 5. Our spectral-based design advances graph augmentation by considering semantics within the spaces.
>
> - Solid Theory: We are the first to prove the semantic preservation nature of spectral graph augmentation. As shown in Section 4, we first prove that our fractional augmentation can preserve semantics within the corresponding continuous semantic space, and then we demonstrate that under a certain approximation, our techniques can approximate any structural modification methods, which means useful variants generated by related work can be a subset of our augmentation. The latter experiments show the significance of preserving semantics for data augmentation in graph-level anomaly detection areas.
>
> - Extensive Experiments: We are the most state-of-the-art methods in the graph augmentation area for graph-level anomaly detection as well. As presented in Section 5, our framework substantially enhances the given GNN baselines compared to related works. Under the extreme scenario of graph-level anomaly detection, previous data augmentation methods fall way behind FracAug due to the lack of possessing semantics and even degrade the performance of the augmented baselines, further demonstrating the effectiveness and novelty of our approach.
>
> ---
>
> **W2**: We respectively disagree with the statement that the effectiveness of our component is doubtful. Specifically, WDML guides FGG during the training procedure, which allows an instance-wise distance adjustment from the decision boundary for each generated sample, assisting the generation of semantic-preserving graphs from FGG, while MVP utilizes mutual verifications to prevent false positives/negatives in the generated training datasets, and thus ensures little noise in later training. The replacement of our design by a simple Cross-Entropy Loss or single verification might hurt the semantics of the generated examples, and include noisy examples. Besides, without our high-quality examples ensured by the three components together, the generalization issue of the backbone will cause loss of information under such conditions, which can be the possible reason that in mere cases, the results fall behind the backbone. Therefore, such an observation should further demonstrate the effectiveness of the combination of our components rather than the contrary.
>
> Other than the ablation studies in Table 8, we also provide additional ablation studies in Appendix Q to investigate the replacement of FGG and MVP by standard data augmentation, such as Node Drop/Edge Drop, and soft ensemble prediction for pseudo-labeling, respectively, which further suggests the effectiveness of our components.

---

> > ### Author Response · Authors · 2025-11-20
> > **Response to Reviewer iYum (2/2)**
> >
> > ---
> >
> > **W3**: Please note that unsupervised GAD methods are out of our scope, as our target is to propose a framework that can boost the performance of graph-level anomaly detection methods under limited supervision. In our proposed framework, we first require warm-up supervised learning for the backbones, so that they can capture the basic knowledge of examples. After the warm-up, we need to train our plug-in framework iteratively with the backbone, guided by supervised loss WDML. Afterwards, MVP will pseudo-label the generated samples so that backbones can gain an additional training set for supervised learning. However, unsupervised GAD methods usually do not require label information in their training procedure. Thus, integrating our proposed framework with unsupervised GAD methods can be a totally different work, which we can explore in the future.
> >
> > ---
> >
> > **Q1**: Following the suggestions of the reviewer, we include several additional ablation studies with different baselines. Specifically, we first change the FGG component to Node Drop/Edge Drop to investigate the influence of our proposed sample generation component. As shown in Appendix Q in our revised manuscript, after replacing with Node Drop/Edge Drop, the final performance will improve to a small extent or even degrade compared to the original baseline. Such a phenomenon suggests that the standard graph augmentation methods might not provide enough useful generated graphs, due to their failure to understand the underlying semantics. In contrast, our proposed FGG component can provide semantic-preserving synthetic samples for pseudo-labeling and thus improve the performance to a larger extent.
> >
> > ---
> >
> > **Q2**: Please note that ConsisGAD is a node-level research, which is orthogonal to our proposed graph-level framework. And the main contribution of ConsisGAD is not in the pseudo-labeling part; instead, their main component is to design a pseudo node representation generation component to generate synthetic samples for pseudo-labeling. Thus, their pseudo-labeling component only utilizes a simple single verification, as shown in Section 3.2.2 in their paper. We have included such a technique in the ablation study in our original manuscript to show the effectiveness of our proposed MVP. To further address the reviewer’s concern, we also replace MVP with a soft ensemble prediction, i.e., averaging the predicted probability of the generated sample and the original sample for pseudo-labeling. As presented in Appendix Q in our revised manuscript, the replacement will also show a negative impact on the performance, compared to the original setting, which aligns with Proposition 1 in our original manuscript. Such a phenomenon suggests that even with additional information from synthetic graphs, a single verification pseudo-labeling will still underperform a mutual verification one. The possible reason lies in that mutual verification pseudo-labeling tends to reject rather than mislabel the synthetic samples.
> >
> > ---
> >
> > We sincerely appreciate your time, and we are glad to answer any additional questions you may have.

---

> > > ### Comment · Reviewer_iYum · 2025-11-26
> > >
> > > Thank you for your timely response and the additional experiments. I still have some concerns about the experimental setting with 1% labeled training data and 98% unlabeled data used for testing.
> > > 1) With only 1% labeled data, the supervised GNN baseline is likely under-trained, whereas your method can leverage the entire dataset (even without using test-set ground truth). I guess substantial portion of the improvement may simply come from training the GNN on more graphs, akin to self-supervised pretraining.
> > > 2) Table 21 further confuses me: even when only 1% additional validation data is used, the results are similar to the case where all 99% of the remaining data is used for pseudo-labeling. This seems to suggest that if we enlarge the labeled training split to, say, 2% or 10%, the marginal gain of FracAug over the baseline might become minimal.
> > >
> > > To address these concerns, I would like to see the following:
> > > 1) Comparisons with unsupervised methods. I do not see why unsupervised approaches cannot be included in this setting: since you already use the entire dataset (labeled training set plus unlabeled test set), unsupervised methods could also use the whole dataset, but without any labels at all, which is arguably a more challenging scenario. I'd like to see whether the improvement simply comes from pretraining.
> > > 2) Results where the labeled training split varies from 1% to 10%, comparing (a) a GNN trained only on the labeled training set and (b) FracAug using the validation set for pseudo-labeling under each split.

---

> > > > ### Author Response · Authors · 2025-11-27
> > > > **Response to Reviewer iYum (1/1)**
> > > >
> > > > Thank you for your insightful comments and suggestions. Our point-to-point responses to your comments are given below.
> > > >
> > > > ---
> > > >
> > > > **C1:** Please note that our model is specifically designed for **supervised learning**, where labeled data plays a central role in training. There are **two key problems** when integrating FracAug and unsupervised models, to be explained below:
> > > >
> > > > 1. Without **supervised** $D_{train}$, our WDML can not be used for training FGG to generate appropriate synthetic examples.
> > > > 2. MVP aims to pseudo-label more data so that the **supervised** model can benefit from the additional label information in later training, but unsupervised models **do not** need label information for training.
> > > >
> > > > To be specific, we clarify the components and training process of FracAug in case there is a misunderstanding.
> > > >
> > > > Three novel components:
> > > >
> > > > 1. FGG: a learnable component to generate synthetic samples for each graph based on semantic consistency, guided by the **supervised** loss WDML.
> > > > 2. WDML: a **supervised** loss, designed for tackling the imbalance issue and adjusting the decision boundary of samples.
> > > > 3. MVP: a pseudo-labeler, using mutual verification to pseudo-label auxiliary data so that **supervised** models can leverage additional label information.
> > > >
> > > > The entire training process:
> > > >
> > > > 1. Warm up by training the baseline using its default setting for $e_{warmup}$ epochs, so that the baseline can obtain a basic understanding of the dataset.
> > > > 2. Use FGG to generate synthetic counterparts for data in $D_{train}$, and then use WDML to train the FGG for $e_{FGG}$ epochs. **Please note that in this step, we must use the ground truth information so that our proposed WDML can guide the parameter update in FGG. If we integrate FracAug with unsupervised methods, to ensure fair comparison, we can not leverage any ground truth information for updating FGG with WDML. Hence, FGG and WDML are not compatible with an unsupervised setting.**
> > > > 3. For data in $D_{val}$ and $D_{test}$, we first use FGG to generate their synthetic counterparts, and then use MVP to pseudo-label the counterparts based on mutual verification between the original and synthetic samples. **Please note that in this step, the key insight is to label additional data to enlarge the $D_{train}$, which is the typical usage of pseudo-labeling methods. Since unsupervised methods can not leverage label information during training, the pseudo-labeled data is meaningless for them. Hence, MVP is not compatible with an unsupervised setting.**
> > > > 4. Use the enlarged $D_{train}$ to train the baseline model with its own setting, and repeat steps 2-4, until the training epoch reaches $e_{default}$, where $e_{default}$ means the training epoch of the baseline in their default setting, and $e_{default}>e_{warmup}$.
> > > >
> > > > As illustrated above, we can see from both our proposed model and the unsupervised model sides, integrating FracAug with the unsupervised model is a difficult but fruitless task.
> > > >
> > > > Besides, as for the concern made by the reviewer, “I guess substantial portion of the improvement may simply come from training the GNN on more graphs”, we need to emphasize that it is the target for plug-in data augmentation works, such as [1-2]. **The key point of such data augmentation is how to generate high-quality synthetic samples or pseudo-labels, so that the model can use additional label information for training.**
> > > >
> > > > To further address the reviewer’s concern, we provide additional experiments in Appendix S. These experiments compare the original baselines (fully trained), FracAug-enhanced baselines, and baselines after warmup. Results show that while warmup helps the baseline capture basic dataset knowledge, it still lags behind the fully trained version. FracAug-enhanced baselines, which integrate FracAug with the warmup, significantly outperform the original baselines. This demonstrates that FracAug leverages the warmup baseline’s knowledge to generate high-quality pseudo-labels, boosting performance compared to the original baseline.
> > > >
> > > > [1] Nan Chen, Zemin Liu, Bryan Hooi, Bingsheng He, Rizal Fathony, Jun Hu, Jia Chen. Consistency Training with Learnable Data Augmentation for Graph Anomaly Detection with Limited Supervision. ICLR 2024.
> > > >
> > > > [2] Botao Wang, Jia Li, Yang Liu, Jiashun Cheng, Yu Rong, Wenjia Wang, Fugee Tsung. Deep Insights into Noisy Pseudo Labeling on Graph Data. NeurIPS 2023.
> > > >
> > > > ---
> > > >
> > > > **C2:** To further address the reviewer’s concern, we include a comparison using only the validation set for pseudo-labeling in Figures 5 and 6 in Appendix H, where we vary the training and validation size from 1% to 40%. While this new setting lags behind our original setup, it still significantly outperforms the baseline across varying training sizes, demonstrating the viability of our framework in strictly inductive scenarios.
> > > >
> > > > ---
> > > >
> > > > We sincerely hope that we have addressed your concerns satisfactorily, and we will gladly answer any additional questions you may have.

---

> ### Comment · Reviewer_iYum · 2025-11-27
>
> C1. I mean compare to results of unsupervised methods, not implementing FragAug over them. Experiments in Appendix S like fully-trained, warm-up are still on 1% training set? I'm not clear with their differences.
>
> C2. Fig.5, Fig6 further validate my point that the main improvment is simply based on ill-trained GNN, In Fig 6 on large dataset  DBLP_v1, the advantage of FragAug is minimal when training set is 20%. The improvment on Fig 5 is because PROTEINS_full is too small. I guess simply pretraining GNN [1] and finetune on GAD task can achieve better performance.
>
> [1] Mole-BERT: Rethinking Pre-training Graph Neural Networks for Molecules

---

> > ### Author Response · Authors · 2025-11-28
> > **Response to Reviewer iYum (1/1)**
> >
> > Thank you for your timely comments. Our point-to-point responses to them are given below.
> >
> >
> > ---
> >
> >
> > **C1:** Thanks for the clarification of the experiment setting. To further address the reviewer’s concern, we compare the unsupervised graph-level anomaly detection (GLAD) baselines provided in [1], including OCGIN, GLADC, GLocalKD, OCGTL, SIGNET, and CVTGAD. As reported in Table 26 in Appendix T of our revised paper, unsupervised training falls significantly behind our method, indicating that label information plays a crucial role in achieving strong anomaly detection performance. Consistent with our earlier experiments, increasing the amount of available labeled data leads to further improvements, reinforcing the importance of supervision. These findings highlight the importance and effectiveness of our proposed plug-in data augmentation framework, since in real deployment, anomaly detection works usually suffer from the limited label supervision issue.
> > ​
> >
> > As for the experiments in Appendix S, the difference between warm-up and fully-trained variants is only the number of training epochs (e.g., train warm-up with 50 epochs, while fully-trained variants with 200 epochs). In this experiment, we aim to demonstrate that, with a basic understanding of the dataset, our FracAug can generate fidelity pseudo-labels for further training to boost the performance to a large extent.
> > ​
> >
> > 1. Yili Wang, Yixin Liu, Xu Shen, Chenyu Li, Kaize Ding, Rui Miao, Ying Wang, Shirui Pan, Xin Wang. Unifying Unsupervised Graph-Level Out-of-Distribution Detection and Anomaly Detection: A Benchmark. ICLR 2025.
> > ​
> >
> > ---
> > ​
> >
> > **C2:** We need to clarify the misunderstanding of the reviewer first. The so-called ill-trained GNN is due to the limited label supervision. Limited label supervision is a typical and key challenge in real deployment for anomaly detection, as the labeling requires expert knowledge and a significant time cost. Therefore, we specifically design FracAug to address such an issue by generating fidelity pseudo-labels for unlabeled data, so that models can achieve higher performance with fewer labels available.
> > ​
> >
> > As shown in Figure 5, the model capacity can handle larger label information to boost its performance, and thus, under a limited supervision setting in the real world, our FracAug framework should be a life-saving solution to make the model less ill-trained. Also, please note that, from Figure 5, it is easy to observe that, with less data (e.g., 1% labels), FracAug-enhanced GIN can compete in performance with vanilla GIN using more data (e.g., 5% labels), which further demonstrates the significance of FracAug under the limited label setting in real life.
> > ​
> >
> > Similar observations can be seen in Figure 6, as we can still utilize FracAug-enhanced models trained on less data to compete with models trained on more data labels. We also notice that due to the model capacity, the model will converge when the data amount reaches 20%. However, please note that in real deployment for anomaly detection, the available label information is usually much less than 20%, which means that to reach the model capacity, we need to use a data augmentation technique for pseudo-labeling. Our FracAug successfully allows the model to reach much higher performance with much fewer labels, which proves the importance of our proposed method.
> > ​
> >
> > Therefore, the two Figures instead support the reviewer’s comment, but reduce the rationality of the reviewer‘s comment, and boost the importance of our proposed FracAug.
> > ​
> >
> > To further address the reviewer’s concern, we compare our performance (FracAug-enhanced GIN) with Mole-BERT (GIN as backbone, following the original setting) mentioned in the reviewer’s comment, which is a model pretrained in an unsupervised manner and subsequently finetuned with labeled data for a specific domain. As shown in Table 26 in Appendix T of our revised paper, the unsupervised pretraining alone produces suboptimal results, consistent with the behavior of the unsupervised baselines. Even after finetuning with the same labeled data budget as our framework, the performance remains inferior to ours to a large extent. This observation suggests two key points: (1) finetuning may inherit a suboptimal parameter space induced by unsupervised pretraining, limiting its adaptability, and (2) sufficient and properly utilized label information is essential for achieving robust performance. Together, these findings further validate the novelty and effectiveness of our plug-in data augmentation framework over the pretrained and finetuned framework, as we can effectively generate more fidelity label information for model training, even under extremely limited supervision.
> > ​
> >
> > ---
> >
> >
> > ​
> > We sincerely hope that we have addressed your concerns satisfactorily, and we will gladly answer any additional questions you may have.
> > ​

---

> > > ### Comment · Reviewer_iYum · 2025-11-28
> > >
> > > Thanks for the authors’ detailed response. Based on the new comparisons with unsupervised GAD methods and the finetuning performance of the self-supervised Mole-BERT model, my concern about the GNN being ill-trained has been resolved. I now get the point that the key to effective GAD lies in the quantity and quality of labeled data, and that FracAug provides an effective way to generate high-quality pseudo-labels under limited supervision, which can substantially improve GNN-based methods in real-world settings. I appreciate the extensive additional experiments and the contributions of this work, and therefore I have decided to increase my score from 4 to 8.

---

> > > > ### Author Response · Authors · 2025-11-29
> > > > **Response to Reviewer iYum (1/1)**
> > > >
> > > > Thank you very much for the time and effort you have put into reviewing our paper and providing insightful comments. Moreover, we are also delighted to know that you are satisfied with our responses and confirm our contribution to the graph anomaly detection area by increasing our rating score. We deeply appreciate your support for our work.

---

### Official Review · Reviewer_L6cQ · 2025-11-04

**Soundness:** 2
**Presentation:** 2
**Contribution:** 3
**Rating:** 6
**Confidence:** 3

**Summary:**

FracAug proposes an augmentation framework that enhances GNNs by generating semantically consistent graph variants through fractional powers of adjacency matrices and pseudo-labeling with mutual verification to address limited supervision and data imbalance in graph-level anomaly detection. The method consists of three components: (1) Fractional Graph Generator guided by weighted distance-aware margin loss, (2) a margin loss to handle class imbalance, and (3) Mutual Verification Pseudo-Labeler that uses predictions from both original and augmented graphs to iteratively expand the training set. Experiments across 14 GNNs on 12 real-world datasets show consistent gains, boosting average AUROC, AUPRC, and F1-score by up to 5.72%, 7.23%, and 4.18%, respectively.

**Strengths:**

1. The paper addresses an important and practical problem of graph-level anomaly detection under limited supervision and severe class imbalance, which is common in real-world applications like drug discovery.
2. The proposed fractional augmentation method provides theoretical guarantees (Theorems 1-2) showing that fractional powers of adjacency matrices can preserve semantic spaces while enabling continuous interpolation of graph structures.
3. Extensive experiments demonstrate consistent improvements across 14 different GNN architectures on 12 real-world datasets, showing the method's broad applicability and model-agnostic nature as a plug-in framework.

**Weaknesses:**

1. Important experiments such as ablation studies should be included in the main paper rather than appendix, as they are critical for validating the core contributions.
2. The variant design in ablation study is problematic. "w/o largest" and "w/o smallest" are not real ablations of the important FGG component but rather create incomplete FGG methods, making worse performance unsurprising. The real ablation should completely remove FGG and replace it with standard graph augmentation methods like edge or node dropout. Similarly, the ablation "w/o MVP" uses only single predictions from original graphs, which obviously performs worse. It should be replaced with widely-used methods like ensemble predictions with dropout enabled to provide fair comparison.
3. The ablation study only involves one backbone (GIN), which is insufficient to support the model-agnostic claim. At least two or three diverse backbones should be tested, especially given that Tables 1-2 show dramatically different improvement magnitudes across architectures.

**Questions:**

See Weaknesses.

---

> ### Author Response · Authors · 2025-11-20
> **Response to Reviewer L6cQ (1/1)**
>
> We appreciate your comprehensive and constructive review. Your crucial comments on experiments are exceedingly helpful for us to improve our manuscript. Our point-to-point responses to your comments are given below.
>
> ---
>
> **W1**: We thank the reviewer for providing valuable suggestions in terms of the paper writing, and we will follow the instructions to revise our main paper.
>
> ---
>
> **W2&W3**: Following the suggestions of the reviewer, we include several additional ablation studies with different baselines. Specifically, we first change the FGG component to Node Drop/Edge Drop to investigate the influence of our proposed sample generation component. As shown in Appendix Q in our revised manuscript, after replacing with Node Drop/Edge Drop, the final performance will improve to a small extent or even degrade compared to the original baseline. Such a phenomenon suggests that the standard graph augmentation methods might not provide enough useful generated graphs, due to their failure to understand the underlying semantics. In contrast, our proposed FGG component can provide semantic-preserving synthetic samples for pseudo-labeling and thus improve the performance to a larger extent.
>
> Besides, we also replace MVP with a soft ensemble prediction, i.e., averaging the predicted probability of the generated sample and the original sample for pseudo-labeling. As presented in Appendix Q in our revised manuscript, the replacement will also show a negative impact on the performance, compared to the original setting, which aligns with Proposition 1 in our original manuscript. Such a phenomenon suggests that even with additional information from synthetic graphs, a single verification pseudo-labeling will still underperform a mutual verification one. The possible reason lies in that mutual verification pseudo-labeling tends to reject rather than mislabel the synthetic samples. Lastly, we further provide the same ablation studies on other backbones than GIN, as shown in Appendix Q, which again indicates the effectiveness of our components.
>
> ---
>
> We sincerely appreciate your time, and we are glad to answer any additional questions you may have.

---

### Official Review · Reviewer_RNUQ · 2025-11-07

**Soundness:** 1
**Presentation:** 3
**Contribution:** 3
**Rating:** 6
**Confidence:** 4

**Summary:**

The paper tackles the challenges of graph-level anomaly detection (GAD) under extremely limited supervision and severely imbalanced class distributions. Existing graph-level augmentation techniques (e.g., MAA, GMixup, GLA) are often heuristic or rely on linear interpolation, making semantic preservation difficult—especially under scarce labels or extreme imbalance. The authors propose FracAug, a model-agnostic augmentation plugin designed to improve GAD performance without modifying the underlying GNN architectures.

**Strengths:**

1. Clear problem formulation.

    The paper accurately identifies two fundamental challenges in GAD: limited supervision and class imbalance. The proposed combination of augmentation and pseudo-labeling is well aligned with these difficulties.

2. Methodological contributions.
    - The use of fractional powers of the adjacency matrix to generate multi-scale structural variations is theoretically grounded and contrasts with discrete perturbation–based augmentations (e.g., MAA, GMixup). The paper argues that such fractional operators preserve semantic continuity in the spectral domain.
    - The proposed Weighted Distance-aware Margin Loss (WDML) integrates instance-specific alignment distance and class reweighting, making it more suitable for heterogeneous GAD samples than fixed-margin approaches such as LMCL or LDAM.
    - The introduction of instance-dependent dynamic margins further distinguishes WDML from fixed-boundary losses and offers better adaptability.
3. High model generality.

    FracAug functions as a plug-in module that can be attached to various GNN backbones without architectural modifications, demonstrating strong extensibility.

4. Comprehensive experimentation.
    - Evaluation across 12 datasets and 14 GNN architectures is extensive.
    - Ablation studies, hyperparameter analysis, and complexity analysis strengthen the empirical claims.

**Weaknesses:**

1. Gap between “semantic preservation” and “label preservation.”

    Theorem 1 shows that (A^{\alpha} x) can be approximated within the original “semantic space,” but this does not directly imply label invariance. For boundary cases or datasets with strong heterophily/feature heterogeneity, even small spectral perturbations may alter anomaly separability.

2. Potential issues in the evaluation protocol.

    The MVP procedure appears to incorporate samples from the validation/test sets back into the training loop, essentially creating a semi-supervised setting. However, final metrics are still reported on these same validation/test splits, raising concerns about possible data leakage.

3. Possible bias from baseline tuning and loss replacement.

    All baselines are replaced with reweighted losses, while FracAug is tuned via grid search to maximize validation performance. It is unclear whether baselines receive equally extensive hyperparameter search or adhere to their original recommended settings. This raises fairness concerns.

4. Marginal gains and lack of statistical significance.

    Many reported improvements are around 1% or less. In heavily imbalanced tasks, such small margins require confidence intervals and statistical significance tests, especially for AUPRC. Multi-seed averaging are recommended.

**Questions:**

1. Evaluation protocol.

    Does the pseudo-labeling process incorporate samples from the test set back into training and then evaluate on the same test set?

    If so, could the authors additionally report results under a strictly held-out test split or cross-split evaluation to eliminate potential leakage?

2. Computational complexity.

    Please provide explicit time/memory complexity formulas for FGG per iteration, as well as empirical scaling curves with respect to (($k_l, k_s, H_l, H_s$)).

3. Empirical justification of semantic preservation.

    Beyond Theorem 1, could the authors report:

    - label stability rates under continuous sampling of ($\alpha$),
    - and comparisons of feature-level or spectral energy distributions?

        These would more directly support the claim that augmentation does not alter label semantics.

4. Fairness of baseline tuning.

    Can the authors provide a comparison using baselines with their default losses and recommended settings, to rule out potential bias introduced by replacing the original loss functions?

5. Failure or degradation cases.

    Since several combinations in Tables 1–3 show negligible improvements or even degradation, can the authors provide typical failure-case analyses (including structural or spectral characteristics) to clarify the applicability range of FracAug?

---

> ### Author Response · Authors · 2025-11-20
> **Response to Reviewer RNUQ (1/3)**
>
> We appreciate your comprehensive and constructive review. Your crucial comments on experiments are exceedingly helpful for us to improve our manuscript. Our point-to-point responses to your comments are given below.
>
> ---
>
> **W1&Q3**: We agree that Theorem 1, which guarantees a high-fidelity approximation within the semantic space, does not directly imply perfect label invariance, particularly in the challenging edge cases the reviewer mentioned. This is one of the reasons why we introduced the WDML and MVP for processing the generated samples from FGG. To be specific, WDML, as illustrated in our manuscript, integrates instance-specific alignment distance to the decision boundary and class reweighting to mitigate the influence of edge cases, and in the meantime, MVP will further use mutual verifications to pseudo-label the generated graphs. As a result, the potential edge case issue will be mitigated to a large extent, ensuring the quality of the synthetic samples.
>
> Besides, to further address the reviewer’s concern, we present the label consistency experiment in Appendix P in the revised manuscript. In this additional experiment, we report the ratio of generated samples that retain the same labels as the original samples after MVP. The results indicate a high label preservation ratio, demonstrating the soundness of our design.
>
> ---
>
> **W2&Q1**: We follow the previous data augmentation setting in the related area, such as [1-2]. In their settings, they include all the validation and test sets for pseudo-labeling, which shows that such a protocol is a valid setting for work in the related area. However, we will further include an additional pseudo-labeling protocol in Appendix M in the revised manuscript to address the reviewer’s concern, where we use only the validation set for pseudo-labeling to mitigate the possible data leakage issue on the test set. We notice that only using the validation set for data augmentation, our FracAug can still boost the performance of the baseline models, which demonstrates that the data leakage should not be an issue for our framework.
>
> [1] Nan Chen, Zemin Liu, Bryan Hooi, Bingsheng He, Rizal Fathony, Jun Hu, Jia Chen. Consistency Training with Learnable Data Augmentation for Graph Anomaly Detection with Limited Supervision. ICLR 2024.
>
> [2] Botao Wang, Jia Li, Yang Liu, Jiashun Cheng, Yu Rong, Wenjia Wang, Fugee Tsung. Deep Insights into Noisy Pseudo Labeling on Graph Data. NeurIPS 2023.
>
> ---
>
> **W3&Q4**: We replace the original Cross-Entropy Loss within several baselines because these baselines are not specifically designed for graph anomaly detection, and thus, they might have problems in dealing with imbalanced data. We made such a change to ensure fair comparison in our setting. However, we understand the concern raised by the reviewer, so we include additional results with baselines using their original unweighted loss in Appendix N of the revised manuscript. The results suggest that when dealing with imbalanced data, the weighted loss can be significant for improving performance, as the performance drops after changing back to the original setting.

---

> > ### Author Response · Authors · 2025-11-20
> > **Response to Reviewer RNUQ (2/3)**
> >
> > ---
> >
> > **W4**: We respectively disagree with the statement of many marginal gains, as we face an extremely difficult but real setting. Yet, we still achieve competitive improvement compared to existing work in data augmentation for non-extreme settings. For instance:
> >
> > - As reported in Table 3 in MAA paper, the empirical gains for SubMix on MUTAG, PROT., PTC-MR, COLLAB, and Twitter are 0.0000, 0.0081, 0.0003, 0.0108, and 0.0039, respectively, and that for NodeSam on D&D, MUTAG, NCI1, PROT., COLLAB, and Twitter are 0.0017, 0.0102, 0.0065, 0.0072, 0.0016, and 0.0008, respectively.
> >
> > - As reported in Table 2 in GMixup paper, its empirical gains with baseline GIN on IMDB-B, REDD-B, REDD-M5, and REDD-M12 are 0.0039, 0.0031, 0.0030, and 0.0027, respectively. Besides, as reported in Table 7 in GMixup paper, its empirical gains with baseline GMT on PROTEINS, IMDB-B, and IMDB-M are 0.0054, 0.0110, and 0.0060, respectively. Moreover, as reported in Table 8 in GMixup paper, its empirical gains with baseline GCN on ogbg-molhiv and ogbg-molbace are 0.0005 and 0.0037, respectively.
> >
> > - As reported in Table 2 in GLA paper, its empirical gains compared to the worst baseline under 30% labels in the paper on DD and GITHUB are 0.0122 and 0.0045, respectively, while its empirical gains compared to the best baseline under 30% labels in the paper on PROTEINS, DD, COLLAB, RDT-B, RDT-M5K, and GITHUB are 0.0009, -0.0111, -0.0110, -0.006, -0.0022, and -0.0280, respectively.
> >
> > - As reported in Table 1 in FGWMixup paper, its empirical gains with baseline GIN on PROTEINS, NCI1, NCI109, and IMDB-M are 0.0027, 0.0029, 0.0094, and 0.0020, respectively, and its empirical gains with baseline GCN on PROTEINS, NCI109, and IMDB-M are 0.0045, 0.0090, and 0.0100, respectively.
> >
> > Please note that our FragAug targets to address the significantly more challenging problem of limited supervision under an imbalanced data setting for graph-level anomaly detection, which is inherently more difficult than standard graph classification tasks addressed by previous methods. As demonstrated in Section 5, existing graph augmentation methods can easily degrade the performance of baselines for graph-level anomaly detection tasks in our target scenario, highlighting the difficulty of achieving any improvement.
> >
> > Despite these challenging conditions, FracAug achieves remarkable improvements:
> > - PROTEINS_full, up to 0.1248, 0.1267, and 0.1279 gains in AUC, AUPRC, and F1, respectively.
> >
> > - DBLP_v1, up to 0.1813, 0.1425, and 0.2032 gains in AUC, AUPRC, and F1, respectively.
> >
> > - Consistent improvements across all datasets and baselines, with many gains exceeding 0.0100.
> >
> > - Average improvements of AUC, AUPRC, and F1 can be up to 0.0572, 0.0723, and 0.0418, respectively.
> >
> > These results demonstrate that our framework not only matches but substantially exceeds the improvements reported by previous graph augmentation methods, while operating under significantly more challenging conditions.
> >
> > However, to further address the reviewer’s concern, we present the corresponding standard deviation of Tables 1, 2, 3, 14, 15, and 16 in Appendix K of our revised manuscript. As shown in the results, our results indicate that FracAug can mitigate the instability of baselines to some extent while ensuring the improvements.

---

> > > ### Author Response · Authors · 2025-11-20
> > > **Response to Reviewer RNUQ (3/3)**
> > >
> > > ---
> > >
> > > **Q2**: Please note that we have provided the complexity analysis of each component in each iteration in Appendix D in the original manuscript. To further address the concern of time usage and memory usage, we additionally provide wall-clock time usage, the total memory usage, and peak memory usage in Appendix D and E in the revised manuscript. Besides, as required, we also provide the corresponding time cost when varying $k_l, k_s, H_l, H_s$ in Appendix O in the revised manuscript. As shown in the comprehensive cost comparison experiments, we observe that our FracAug achieves a relatively low cost in terms of both memory and time, while boosting the performance of the baselines, demonstrating the superiority of our framework.
> > >
> > > ---
> > >
> > > **Q5**: As shown in Appendix B in our original manuscript, small molecular datasets that are usually used in the related area, i.e., MCF-7, MOLT-4, PC- 3, SW-620, NCI-H23, OVCAR-8, P388, SF-295, SN12C, and UACC257, have one-hot encoding features, while the PROTEINS_full and DBLP_v1 datasets have meaningful features. Sparse features of small molecular datasets might negatively influence the downstream detection by making the synthetic structural information less significant. Therefore, in some cases, the improvements might be relatively small compared to other outstanding ones in our experiments. However, compared to previous related work, as mentioned in **W4**, we believe that our FracAug has a broader range of applications, even with extremely limited settings.
> > >
> > > ---
> > >
> > >
> > > We sincerely appreciate your time, and we are glad to answer any additional questions you may have.

---

> ### Comment · Reviewer_RNUQ · 2025-11-26
>
> The additional experiments in Appendices M, N, P and K have satisfactorily addressed my major concerns regarding the evaluation protocol, baseline fairness, and label consistency.
>
> However, I have one remaining suggestion for the final version: Since the main results utilize the test set for pseudo-labeling, I strongly recommend explicitly labeling this setting as "transductive" or "semi-supervised" in the paper. This distinction is critical to prevent confusion with standard inductive settings, even though the new Appendix M successfully demonstrates the method's viability in a strictly inductive scenario.

---

> > ### Author Response · Authors · 2025-11-27
> > **Response to Reviewer RNUQ (1/1)**
> >
> > Thank you very much for the time and effort you have put into reviewing our paper and providing insightful comments. We have revised our manuscript according to your constructive suggestions. Please kindly refer to Section 5.2 in our revised manuscript. Moreover, we are also delighted to know that you are satisfied with our responses and confirm our contribution to the graph anomaly detection area. We deeply appreciate your support for our work and welcome further discussion.

---

### Author Response · Authors · 2025-11-20
**Modifications in the revised version of our manuscript**

We express our gratitude to all the reviewers for their thorough and constructive feedback. Taking into consideration the valuable comments provided by the reviewers, we intend to incorporate the following modifications in the revised version of our manuscript.

---
Section 5
- shorten comparison Tables 1, 2, and 3 (Reviewer **L6cQ W1** and Reviewer **rSBY W3** )

- move part of Ablation Study section from Appendix to Section 5 (Reviewer **L6cQ W1** and Reviewer **rSBY W3** )

- move Hyperparameter Sensitivity section from Appendix to Section 5 (Reviewer **L6cQ W1** and Reviewer **rSBY W3** )

Appendix
- move the remaining comparison Tables 1, 2, and 3 to Appendix J (Reviewer **L6cQ W1** and Reviewer **rSBY W3** )

- add the wall-clock time comparison in Appendix D (Reviewer **RNUQ Q2** and Reviewer **rSBY W1** )

- add the consumed memory and peak memory comparison in Appendix E (Reviewer **RNUQ Q2** and Reviewer **rSBY W1** )

- add the standard deviation of Tables 1, 2, 3, 14, 15, and 16 in Appendix K (Reviewer **RNUQ W4** and Reviewer **rSBY W4** )

- add MVP only for validation set comparison in Appendix M (Reviewer **RNUQ W2&Q1**)

- add original baseline loss comparison in Appendix N (Reviewer **RNUQ W3&Q4**)

- add running time when varying hyperparameters in Appendix O (Reviewer **RNUQ Q2**)

- add pseudo labeling consistency ratio in Appendix P (Reviewer **RNUQ W1&Q3**)

- add further ablation study for FGG/MVP in Appendix Q  (Reviewer **L6cQ W2&W3** and Reviewer **iYum W2&Q1&Q2** )

- add influence of middle eigenvalues in Appendix R (Reviewer **rSBY W2**)

---

These modifications have been included in the revised version of our manuscript, which has been highlighted in blue to facilitate the reviewing process.

---

### Author Response · Authors · 2025-11-30
**Summarization of Discussion (1/2)**

Dear PCs, SACs, and ACs,

We sincerely thank you for your time, effort, and careful coordination of the discussion process. To facilitate the decision-making and make it less burdensome to value your work, below we summarize the full author–reviewer interaction to allow you to clearly assess how each reviewer’s concerns have been addressed and whether their rating improved after discussion. We follow a reviewer-by-reviewer structure to ensure clarity and completeness, where W stands for Weaknesses, Q stands for Questions, and C stands for Comments.

---

**Reviewer**: **RNUQ**

**Status**: **Confirm concerns are satisfactorily addressed. Keep the initial rating of 6.**

**Discussion**:

1. Concerns about label semantic consistency (**W1&Q3**): Addressed by additional experiment in Appendix P, i.e., we empirically demonstrate that the pseudo-labels have high semantic consistency during training.

2. Concerns about evaluation protocol (**W2&Q1**): Addressed by additional experiment in Appendix M, i.e., we empirically demonstrate that even under a strict inductive setting, our FracAug can still boost the performance of baselines to a large extent.

3. Concerns about baseline fairness (**W3&Q4**): Addressed by additional experiment in Appendix N, i.e., we empirically demonstrate that we instead reduce fairness, but improve the fairness during comparison with baselines.

4. Concerns about statistical significance (**W4**): Addressed by additional experiment in Appendix K, i.e., we provide the standard deviation of Tables 1, 2, 3, 14, 15, and 16.

5. Concerns about computational complexity (**Q2**): Addressed by additional experiments in Appendix D, E, and O, i.e., we provide comparisons of running time cost, wall-clock time cost, total memory usage, and peak memory usage.

6. Concerns about degradation cases. (**Q5**): Addressed by further explanation based on Appendix B, i.e., we provide the analysis that the degradation cases are caused by the one-hot encoding features of public datasets.

7. Concerns about experimental setting (**C1**): Addressed by further explanation in Section 5.2, i.e., we emphasize our experiments are conducted under semi-supervised settings.

---

**Reviewer**: **L6cQ**

**Status**: **Initial rating is 6. Although with no further discussion, we believe the reviewer’s concerns are satisfactorily addressed.**

**Discussion**:

1. Concerns about paper organization (**W1**): Addressed by adjustment of paper organization, i.e., we move the ablation study and hyperparameter analysis from Appendix to the main pages.

2. Concerns about ablation study (**W2&W3**): Addressed by additional experiment in Appendix Q, i.e., we conduct additional ablation study by replacing FGG with other data augmentation methods and MVP with other pseudo-labeling methods on different baselines.

---

**Reviewer**: **iYum**

**Status**: **Confirm concerns are satisfactorily addressed. Increase rating to 8.**

**Discussion**:

1. Concerns about novelty (**W1**): Addressed by further explanation in official comments, i.e., we emphasize our novelty lies in the distinct design, solid theory, and extensive experiments.

2. Concerns about the effectiveness of the proposed components (**W2**): Addressed by additional experiment in Appendix Q, i.e., we conduct additional ablation study by replacing FGG with other data augmentation methods and MVP with other pseudo-labeling methods on different baselines.

3. Concerns about the comparison with unsupervised models (**W3&C1**):  Addressed by additional experiment in Appendix T, i.e., we empirically demonstrate that FracAug-enhanced baselines can outperform unsupervised models to a large extent.

4. Concerns about comparison with other generation methods (**Q1**): Addressed by additional experiment in Appendix Q, i.e., we conduct additional ablation study by replacing FGG with other data augmentation methods on different baselines.

5. Concerns about comparison with other pseudo-labeling methods (**Q2**): Addressed by additional experiment in Appendix Q, i.e., we conduct additional ablation study by replacing MVP with other pseudo-labeling methods on different baselines.

6. Concerns about comparison with pretrain models (**C2**): Addressed by additional experiment in Appendix T, we empirically demonstrate that FracAug-enhanced baselines can outperform pretrain models to a large extent.

---

---

> ### Author Response · Authors · 2025-11-30
> **Summarization of Discussion (2/2)**
>
> **Reviewer**: **rSBY**
>
> **Status**: **Initial rating is 2. However, we believe the rating is unfair, given the positive feedback from other reviewers. Although without any discussion, we believe the reviewer’s concerns are satisfactorily addressed.**
>
> **Discussion**:
>
> 1. Concerns about scalability and resources (**W1**): Addressed by additional experiments in Appendix D, E, and O, i.e., we provide comparisons of running time cost, wall-clock time cost, total memory usage, and peak memory usage.
>
> 2. Concerns about spectral choices (**W2**): Addressed by additional experiment in Appendix R, we empirically demonstrate that middle eigenvalues have very limited influence on our experiments, and our design for including only large and small eigenvalues is supported by related papers.
>
> 3. Concerns about organization of experiments (**W3**): Addressed by adjustment of paper organization, i.e., we move the ablation study and hyperparameter analysis from Appendix to the main pages.
>
> 4. Concerns about variance and significance (**W4**): Addressed by additional experiment in Appendix K, i.e., we provide the standard deviation of Tables 1, 2, 3, 14, 15, and 16.
>
> ---
>
> We sincerely appreciate the time and effort that the reviewers have devoted to evaluating our submission. However, we would like to respectfully express a concern regarding the nature of the reviewer’s evaluation. Reviewer **rSBY** has never participated in the discussion, although we have provided our detailed experiments and explanations at an early stage, and all the other reviewers have provided positive ratings. Without discussion of the paper’s substance, it is difficult for us to understand the basis of the assigned rating or provide meaningful clarifications.
>
> Therefore, we briefly reemphasize the contribution of our work so that PCs, SACs, and ACs can clearly and quickly grasp our contribution:
>
> **Distinct Design:** We present the first data augmentation approach specifically designed for graph-level anomaly detection, simultaneously tackling the limited supervision and distribution imbalance problems. Under such an extreme scenario, previous augmentation methods that neglect the semantics of samples often generate suboptimal or counterproductive samples, as shown in our experiments in Section 5. Our spectral-based design advances graph augmentation by considering semantics within the spaces.
>
> **Solid Theory:** We are the first to prove the semantic preservation nature of spectral graph augmentation. As shown in Section 4, we first prove that our fractional augmentation can preserve semantics within the corresponding continuous semantic space, and then we demonstrate that under a certain approximation, our techniques can approximate any structural modification methods, which means useful variants generated by related work can be a subset of our augmentation. The latter experiments show the significance of preserving semantics for data augmentation in graph-level anomaly detection areas.
>
> **Extensive Experiments:** We are the most state-of-the-art methods in the graph augmentation area for graph-level anomaly detection as well. As presented in Section 5, our framework substantially enhances the given GNN baselines compared to related works. Under the extreme scenario of graph-level anomaly detection, previous data augmentation methods fall way behind FracAug due to the lack of possessing semantics and even degrade the performance of the augmented baselines, further demonstrating the effectiveness and novelty of our approach.
>
> ---
>
> If the PCs, SACs, and ACs have any further comments or advice regarding our manuscript, we welcome you to share them with us. Your guidance is highly valued, and we are looking forward to your feedback.
>
> Warm regards,
>
> The Authors of Submission12794

---

### Meta-Review · Area_Chair_bqvx · 2025-12-10

**Summary:**

This paper introduces FracAug for graph-level anomaly detection. The authors made a clear effort to address reviewer concerns through additional experiments and analyses, which improved the empirical presentation of the work. However, key issues remain, particularly regarding the conceptual motivation of fractional spectral augmentation, the lack of clear guarantees on semantic or label preservation, and the scalability of the spectral operations to realistically large graphs. As a result, while the empirical results are promising, the overall clarity and practicality of the approach remain insufficient for acceptance. I therefore recommend reject.

**Reviewer Concerns:**

Most concerns raised by RNUQ were addressed, including label preservation, potential data leakage, baseline fairness, variance reporting, and complexity analysis. L6cQ’s questions about ablation fairness and model-agnostic validation were also largely resolved through additional experiments. iYum’s concerns regarding novelty, component effectiveness, and the effect of limited supervision were fully addressed, leading to a significant score increase. The only issues that remain partly open are rSBY’s concerns about spectral choices and scalability, since this reviewer did not participate in the discussion and did not confirm whether the additional evidence fully resolved these points.

**Reviewer Scores:**

RNUQ began with a score of 6 and later indicated that all major concerns were satisfactorily resolved, so the score would likely have remained at 6.

L6cQ also started at 6 and did not rejoin the discussion, but the additional ablations and reorganized analyses directly addressed the reviewer’s requests, so the score would probably have stayed at 6.

iYum started at 4 and explicitly stated after the rebuttal and additional experiments that the concerns were fully addressed, raising the score to 8.

rSBY began at 2 and did not participate in the discussion. Although the authors provided detailed responses and new experiments, there is no indication that the reviewer would have updated the score, so it would likely have remained at 2.

---

### Decision · Program_Chairs · 2026-01-26

Reject